# Remodeling synaptic connections via engineered neuron-astrocyte interactions

Shin Heun Kim[1,8], Woojin Won [1,8], Gyu Hyun Kim [2,3,8], Yeon Hee Kook[1,4], Seungkyu Son[1], Songhee Choi [5], Dong Yeop Kang[1,6], Mingu Gordon Park [1], Young-Jin Choi[1,7], Seong Su Won[1], Juhee Shin[1], Yong Jeong [5], Kea Joo Lee [2] ✉, C. Justin Lee [1] ✉ & Sangkyu Lee [1,7] ✉

Information flow through synapses in the central nervous system is regulated by both rapid electrochemical activity and slower structural remodeling. While technological advances allow precise manipulation of synaptic activity, methods for structural remodeling remain limited. Here, we present SynTrogo (Synthetic Trogocytosis), a synthetic molecular approach for modulating synaptic connections. By engineering complementary ligand and receptor proteins, we enable physical interaction between two defined cell populations in culture, leading to a trogocytosis-like process in which receptor-expressing cells internalize membrane fragments and adjacent cytosolic material from ligand-expressing cells. Applying SynTrogo to hippocampal CA3 neurons and CA1 astrocytes in adult male mice results in ultrastructural changes at axon-astrocyte interfaces, accompanied by significantly reduced synaptic connectivity. The remaining synapses exhibit coordinated pre- and post-synaptic structural changes and reorganization of synaptic components and organelles, and are associated with enhanced synaptic plasticity and memory performance. These findings suggest that neural circuits can undergo adaptive reshaping under conditions of synaptic reduction and may provide a foundation for editing synaptic architecture with therapeutic potential for connectopathies.

The synapse, a fundamental unit connecting neurons, plays a crucial role in the rapid transmission of electrical and chemical signals across neural circuits[1], thereby finely regulating various brain functions. Over the past decades, remarkable technological advancements, such as the development of optogenetic and chemogenetic approaches, have provided capabilities for spatiotemporal control over information flow in the brain[2–4]. However, these methods heavily rely on existing synaptic connections, limiting their ability to structurally modify synaptic connectivity.

On slower timescales, spanning days or even longer periods, brain function is modulated by structural changes in neural circuits through the bidirectional control of synaptic connections[5]. Synaptic pruning, a key process for eliminating specific synapses throughout life, is regulated by cell-autonomous or non-cell-autonomous mechanisms[6–9]. The

[1]Center for Memory and Glioscience, Institute for Basic Science (IBS), Daejeon, Republic of Korea. [2]Neural Circuits Group, Korea Brain Research Institute (KBRI), Daegu, Republic of Korea. [3]Department of Neuroscience, Korea University College of Medicine, Seoul, Republic of Korea. [4]Department of Bioscience and Biotechnology, Graduate School, Chungnam National University, Daejeon, Republic of Korea. [5]Department of Bio and Brain Engineering, Korea Advanced Institute of Science and Technology (KAIST), Daejeon, Republic of Korea. [6]Department of Medical Science, Chungnam National University, Daejeon, Republic of Korea. [7]Department of Biomedical Engineering, Ulsan National Institute of Science and Technology (UNIST), Ulsan, Republic of Korea. [8]These authors contributed equally: Shin Heun Kim, Woojin Won, Gyu Hyun Kim. ✉e-mail: relaylee@kbri.re.kr; cjl@ibs.re.kr; sklee@ibs.re.kr

latter is orchestrated by glial cells, including microglia[10–13], astrocytes[14,15], Bergmann glia[16], and oligodendrocyte precursor cells[17,18], and substantially contributes to the refinement of neural circuits and the maintenance of optimal connectivity essential for brain function. Dysregulation of this process is associated with a spectrum of psychiatric and neurodegenerative disorders[6,7].

Astrocytes are characterized by highly ramified processes that intimately enwrap synapses[19–21]. Their extensive perisynaptic processes enable precise synapse-specific interactions[19–22] and exhibit phagocytic capabilities for synaptic elimination[14,15]. These characteristics make astrocytes ideal cellular targets for controlled synaptic pruning through molecular engineering approaches.

Although accumulating evidence indicates that neural activity drives the selective elimination of weaker synapses and subsequent strengthening of others[6,7,23,24], conventional methods coupling neural activity manipulation with synaptic pruning have made it challenging to isolate pruning-specific effects on circuit remodeling and behavior. To address this limitation, methods for directing synapse elimination independent of neural activity are needed.

In this study, we introduce SynTrogo (Synthetic Trogocytosis), a synthetic molecular approach that enables astrocyte-mediated reduction of synaptic connectivity within targeted neural circuits. In cultured conditions, expression of cognate synthetic ligand and receptor proteins in two distinct cell populations induces strong cell–cell interactions, leading to a trogocytosis-like process in which receptor-expressing cells take up membrane fragments and adjacent cytosolic material from ligand-expressing cells. By leveraging the natural proximity of astrocytes to synapses, we apply this strategy to promote direct interactions between ligand-labeled CA3 excitatory neurons and receptor-expressing CA1 astrocytes in vivo. Using immunohistochemistry, correlative light and electron microscopy (CLEM), electrophysiological recordings, and behavioral analyses, we show that enforced neuron-astrocyte associations give rise to ultrastructural alterations at their interfaces and are associated with decreased synaptic connectivity in target circuits. This reduction is accompanied by compensatory reorganization of the remaining synapses, characterized by structural and functional remodeling, enhanced synaptic plasticity, and improved memory performance. These findings provide insights into how synaptic connections can reorganize in the context of synaptic reduction, highlighting its impact on circuit function and cognition, and suggesting potential therapeutic strategies for neurological disorders associated with aberrant synaptic connectivity.

## Results

### Development and characterization of Synthetic Trogocytosis (SynTrogo)

In light of the prerequisite for direct cell-cell binding through molecular interactions of natural eat-me signals and cognate receptors in various glial cell-mediated synaptic pruning mechanisms[6,7], we aimed to design a synthetic approach to artificially induce physical interactions between two defined cell populations using a well-characterized protein pair. To this end, we employed the high-affinity binding pair of green fluorescent protein (GFP) and GFP nanobody (αGFP)[25]—a single-domain antibody fragment that specifically recognizes GFP—because they can reliably and selectively engage in tight binding, enabling us to mimic the cell-cell contact observed during natural synapse pruning (Fig. 1a).

These proteins were inserted between the igκ signal peptide and the transmembrane (TM) domain of the platelet-derived growth factor receptor (PDGFR) for efficient cell surface display. Co-incubation of HEK293T cells expressing igκ-GFP-TM (ligand) with HeLa cells expressing igκ-αGFP-TM-RFP (receptor) for 4 h resulted in substantial accumulation of ligand molecules within vesicles in HeLa cells (Supplementary Fig. 1a, b). Time-lapse imaging showed the enrichment of

ligands and receptors at the intercellular contact site, followed by engulfment of ligands by receptor cells (Supplementary Fig. 1c-f; Supplementary Movies 1 and 2). Ligand uptake was specifically induced by directly bound but not by neighboring non-bound receptor cells (Fig. 1b, c). Over time, the ingesting vesicles either merged into larger structures or split into smaller ones, respectively (Supplementary Fig. 1g, h). This process resembles trogocytosis—an intercellular interaction involving the uptake of membrane patches of one cell by another[26]—and is hereafter referred to as SynTrogo (Synthetic Trogocytosis).

SynTrogo was specifically induced only when both binding modules were displayed on the plasma membranes (Supplementary Fig. 2a, b), and attenuated by treatment of a competitive anti-GFP antibody (Fig. 1d). Yellow fluorescent protein (YFP), but not cyan fluorescent protein (CFP), can serve as a ligand for efficient uptake (Fig. 1e), likely due to stronger binding affinity of YFP to αGFP compared to CFP[27]. Consistently, the application of GFP nanobody variants (LaG5, 6, 17, 18, and 42) demonstrated a positive correlation between the SynTrogo efficiency and protein binding affinity[28] (Fig. 1f and Supplementary Fig. 2c). Analysis of receptor accumulation at the cell–cell contact site, used here as an indicator of intercellular binding affinity, revealed that only LaG18- and LaG5-conjugated receptors displayed significantly reduced enrichment at the contact site compared to the αGFP receptor, along with minimal ligand uptake (Supplementary Fig. 2c–e). These findings support that strong intercellular binding is essential for robust induction of SynTrogo. Other binding pairs[29] can also induce SynTrogo (Supplementary Fig. 2f, g), suggesting its expandability and orthogonality.

Volumetric time-lapse imaging revealed dynamic membrane protrusion and retraction cycles during SynTrogo (Supplementary Fig. 3a–d and Supplementary Movie 3), suggesting an involvement of actin polymerization. Indeed, inhibiting actin polymerization significantly reduced ligand uptake, and myosin-mediated actin contraction is crucial (Fig. 1g and Supplementary Fig. 3e). Flow cytometry analysis revealed that SynTrogo induction did not cause detrimental effects on cell viability up to 72 h after co-incubation of ligand- and receptor-expressing cells (Supplementary Fig. 4). In addition, live-cell imaging showed that cell proliferation was not affected up to 18 h after co-incubation (Supplementary Movie 4). This lack of cytotoxicity can be explained by the co-internalization of ligand–receptor complexes during the process (Supplementary Movies 2 and 4), which progressively reduces the pool of available surface pairs. This self-limiting mechanism likely constrains continued SynTrogo and may help preserve overall cellular integrity.

Using the same ligand–receptor pair, SynTrogo was efficiently induced in both homotypic and heterotypic interactions across various cell types, including fibroblasts, microglia, astrocytes, and neurons (Fig. 1h, i and Supplementary Fig. 5a–c), indicating the versatility of this platform. Time-lapse imaging of SynTrogo between HEK293T cells and cultured astrocytes revealed that HEK293T cells displayed dynamic migration along the astrocyte surface, facilitating contact and subsequent internalization of ligand (Supplementary Movie 5). In cultured hippocampal neurons, engulfed ligand-containing vesicles were observed to move along neurites in a retrograde manner toward the soma, where they accumulated. This suggests that the movement of internalized molecules is regulated by the vesicle trafficking mechanisms of host cells (Supplementary Fig. 5d).

The majority of internalized ligands localized to the lysosomes and Golgi apparatus, implying that these molecules undergo degradation and recycling processes (Fig. 1j and Supplementary Fig. 6a–c). To determine whether GFP fluorescence is quenched following fusion with acidic lysosomes, we tracked GFP-positive vesicles in receptor cells using time-lapse imaging, and found a substantial decrease in GFP fluorescence, whereas RFP signals of the receptor remained stable (Supplementary Fig. 6d–f). These results support the notion that

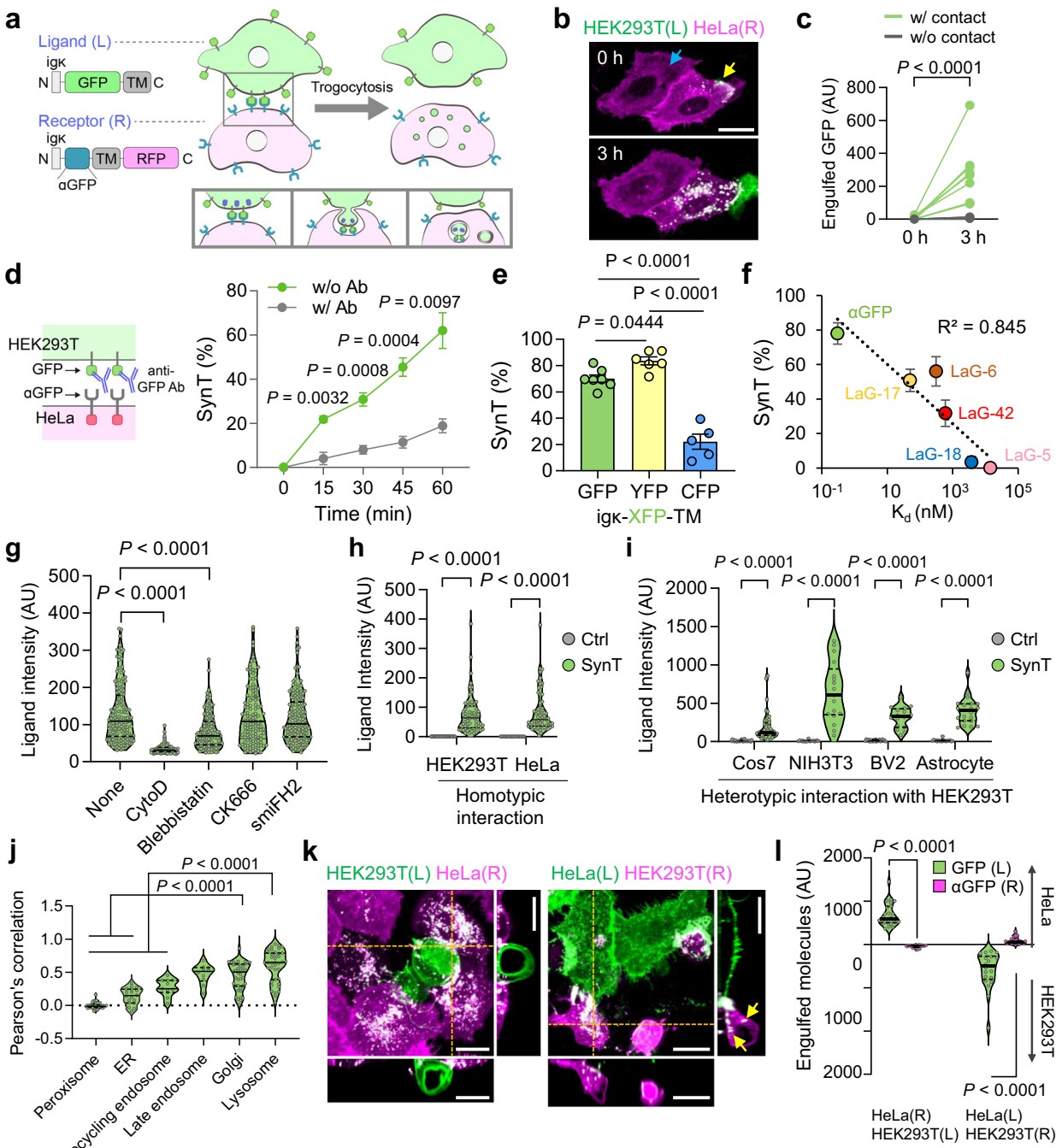

**Fig. 1 | Development and characterization of SynTrogo. a** Schematic illustration of ligand (L) and receptor (R) constructs inducing SynTrogo (SynT) via ligand–receptor interactions. **b** Representative fluorescence images of SynT between a ligand-expressing HEK293T and a receptor-expressing HeLa cell. Scale bar, 20 μm. **c** GFP intensity in receptor-expressing cells with or without contact after 3 h (n = 8 cells per condition). **d** Attenuation of SynT by anti-GFP antibody (n = 475 and 516 cells without and with antibody). **e** Percentage of GFP-positive HeLa cells using different ligand variants (n = 443 GFP, 390 YFP, 169 CFP cells). **f** Percentage of GFP-positive HeLa cells using different αGFP nanobodies (n = 134 αGFP, 122 LaG-5, 82 LaG-6, 91 LaG-17, 81 LaG-18, 74 LaG-42 cells). **g** Ligand signals in receptor-expressing cells treated with cytoskeletal inhibitors (n = 151 None, 133 CytoD, 132 Blebbistatin, 120 CK666, 156 smiFH2 cells). **h** Ligand signal quantification during homotypic interactions (HEK293T: n = 63 Ctrl, 63 SynT cells; HeLa: n = 30 Ctrl, 62 SynT cells). **i** Ligand signal quantification during heterotypic

interactions (Ctrl: n = 29 Cos7, 22 NIH3T3, 22 BV2, 19 astrocyte cells; SynT: n = 44 Cos7, 17 NIH3T3, 16 BV2, 21 astrocyte cells). **j** Pearson's correlation analysis of colocalization between engulfed ligands signals and organelle markers (n = 33 peroxisome, 18 ER, 11 recycling endosome, 10 late endosome, 40 Golgi, and 39 lysosome-positive cells). **k** Maximum-intensity z-projections showing unidirectional ligand transfer. Scale bars, 20 μm. **l** Ligand and receptor uptake during SynT (HeLa(R)-HEK293T(L): n = 24 HeLa, 25 HEK293T; HeLa(L)–HEK293T(R): n = 21 HEK293T, 17 HeLa cells). Data are mean ± s.e.m. in (**c**–**f**), or median with interquartiles (**g**–**j**, **l**). Statistical significance was determined using two-way repeated-measures ANOVA with Fisher's LSD multiple-comparison test for (**c**), two-way repeated-measures ANOVA with Sidak's multiple-comparison test for (**d**), one-way ANOVA with Tukey's multiple-comparison test for (**e**, **j**), one-way ANOVA with Dunnett's multiple-comparison test for (**g**), multiple unpaired t-tests for (**h**, **i**), and unpaired t-test for (**l**). Source data are provided as a Source Data file.

actual levels of internalized GFP ligand may be underestimated due to fluorescence quenching within the acidic lysosomal environment.

SynTrogo exhibited a property of unidirectional molecular transfer (Fig. 1k, l). For example, when HEK293T cells expressed the ligand and HeLa cells expressed the receptor, we observed a prominent transfer of the ligand from HEK293T cells to HeLa cells, whereas the transfer of the receptor was minimal. Conversely, when we reversed the expression pattern—expressing the receptor in HEK293T cells and the ligand in HeLa cells—ligands were transferred from HeLa cells to HEK293T cells. These results suggest that the directionality of SynTrogo is predominantly determined by the identity of the expressed proteins, rather than by the cell type.

Next, we examined which molecules could be co-transferred with the ligand to receptor cells. Adjacent membrane lipids, membrane-anchored bystander proteins, and cytosolic actin-bound proteins at the site of SynTrogo were engulfed with the ligand by receptor cells (Supplementary Fig. 7), whereas non-tagged cytosolic fluorescent proteins were not detected in the ligand-positive vesicles, indicating that only molecules physically close to the ligand were transferable.

## SynTrogo of cultured neurons by astrocytes

Given the natural proximity of astrocytic processes to synapses[20–22], these observations led us to hypothesize that SynTrogo of ligand-tagged neurons by receptor-expressing astrocytes could result in the uptake of portions of neuronal membranes along with endogenous synaptic molecules, potentially modulating synaptic connectivity. In cultured conditions, receptor astrocytes actively trogocytosed ligand-labeled neuronal membrane fragments, whereas no discernible ligand signal was detected in receptor-negative astrocytes (Fig. 2a, b). Time-lapse imaging revealed that the uptake was mediated by direct neuron-astrocyte interactions and occurred through multiple rounds of SynTrogo (Fig. 2c). Depending on their sites of contact with neurons, astrocytes were capable of trogocytosing various neuronal membrane regions, including axons, dendrites, and soma (Supplementary Fig. 8).

Importantly, the levels of endogenous synaptic molecules within astrocytes significantly increased following SynTrogo (Fig. 2d, e), indicating that these molecules were co-internalized with the GFP ligand. To further validate this process, we co-expressed the ligand and blue fluorescent protein-labeled synaptophysin (TagBFP-SYP) in cultured neurons, while expressing the receptor in astrocytes. Time-lapse imaging during SynTrogo revealed co-uptake of the ligand and synaptophysin by astrocytes, whereas TagBFP expressed alone in control neurons (not fused to synaptophysin) were not appreciably internalized (Fig. 2f, g). These results are consistent with our observation in Supplementary Fig. 7, and further support the idea that adjacent synaptic molecules can be co-internalized with the ligand during SynTrogo.

## SynTrogo of CA3 presynaptic axons by CA1 astrocytes

To apply SynTrogo in the mouse brain, we targeted the hippocampal CA3-CA1 circuit by expressing the ligand in CA3 excitatory neurons and the receptor in CA1 astrocytes (Fig. 3a). In the control mice, where astrocytes expressed membrane-anchored RFP (Lck-RFP) instead of the receptor, ligand signals from CA3 neuronal axons were broadly distributed throughout the CA1; however, receptor expression substantially altered ligand distribution, with signals largely overlapping with astrocytic territories (Fig. 3b, c). Within these territories, GFP signal intensity was significantly higher under SynTrogo conditions, whereas ligand levels outside astrocytic domains remained comparable to controls (Fig. 3d). High-magnification images revealed significantly increased ligand puncta colocalized with receptor-expressing astrocytes relative to the control group (Fig. 3e–g). Small yet discernable amounts of GFP puncta were also detected in control astrocytes, which may reflect the basal phagocytic activity of astrocytes.

To assess whether SynTrogo influences the astrocytic localization of synaptic components, we performed immunostaining analysis and found that excitatory presynaptic molecules (VGluT1) showed increased signal intensity within astrocytic soma areas under SynTrogo conditions, whereas excitatory postsynaptic (PSD95) and inhibitory synaptic molecules (VGAT and Gephyrin) did not differ significantly between groups (Fig. 3g, h). VGluT1 signals were observed partially colocalized with ligand puncta within astrocytic territories (Fig. 3g), and their signal intensities exhibited a significant positive correlation (Fig. 3i), suggesting that astrocytic colocalization of presynaptic components is positively associated with the extent of ligand localization under SynTrogo conditions. Consistent with these observations, when TagBFP-labeled synaptophysin was expressed in ligand-positive CA3 neurons, astrocytic regions displayed puncta in which ligand and synaptophysin signals were colocalized (Supplementary Fig. 9). We observed synaptophysin (+) but GFP (−) puncta in astrocytes. One plausible explanation is that, even when the GFP ligand and synaptophysin are present within the same astrocytic territory, subsequent vesicle trafficking may separate synaptophysin-containing membranes from those containing the GFP ligand, yielding structures that retain synaptophysin without detectable GFP. Alternatively, the presence of synaptophysin (+) puncta in control astrocytes suggests that synaptophysin can appear in astrocytes through SynTrogo-independent processes. Given the resolution limits of confocal light microscopy, these observations cannot definitively establish engulfment. Although they may be suggestive of internalization, definitive ultrastructural validation requires higher-resolution imaging analyses (e.g., correlative light and electron microscopy imaging).

Application of LaG18- or LaG5-conjugated receptors, which have lower binding affinity to the GFP ligand than the αGFP receptor, to CA1 astrocytes did not cause a significant increase in synaptic marker signal intensity within astrocytic territories (Supplementary Fig. 10). This result is consistent with our findings in cultured conditions (Fig. 1f and Supplementary Fig. 2c) and suggests that the increased colocalization of synaptic molecules with astrocytes depends on robust induction of SynTrogo, which requires strong cell-cell interactions.

Importantly, quantifying the number of synaptic markers revealed that the density of excitatory but not inhibitory synapses was significantly reduced within SynTrogo-positive astrocytic territories (Fig. 3j, k). When broader CA1 areas were analyzed, a similar decrease in excitatory synapse density was observed, while inhibitory synapse density was rather increased (Fig. 3l), suggesting a shift in synapse-type composition.

Under these experimental conditions, 78.3% (control) and 80.3% (SynTrogo) of CA3 neurons expressed the ligand, and 78.3% (control) and 84.4% (SynTrogo) of CA1 astrocytes expressed RFP and the receptor (Supplementary Fig. 11a–d). Immunostaining of GFAP and Iba1 revealed no signs of astrocyte or microglia reactivity (Supplementary Fig. 11e–h). In addition, the density of ligand-expressing neurons did not significantly differ between control and SynTrogo groups at three weeks and two months post-injection, indicating that SynTrogo in the brain did not lead to neuronal death (Supplementary Fig. 11d and 12a–c). To investigate whether SynTrogo might remove axons, we co-expressed the GFP ligand along with a non-tagged fluorescent protein (smFP-FLAG) in CA3 neurons to visualize axonal morphology and found no evidence of axonal loss even two months after viral injection (Supplementary Fig. 12d). These results, together with our time-lapse imaging observations in vitro (Supplementary Movies 2 and 4), are consistent with the possibility that SynTrogo operates in a self-limiting manner, allowing axons to retain their structural integrity without progressing to overt structural degeneration.

## SynTrogo of motor cortex neurons by striatal astrocytes

To examine whether SynTrogo is applicable beyond the hippocampus, we targeted the motor cortex-striatum pathway[30] by expressing the

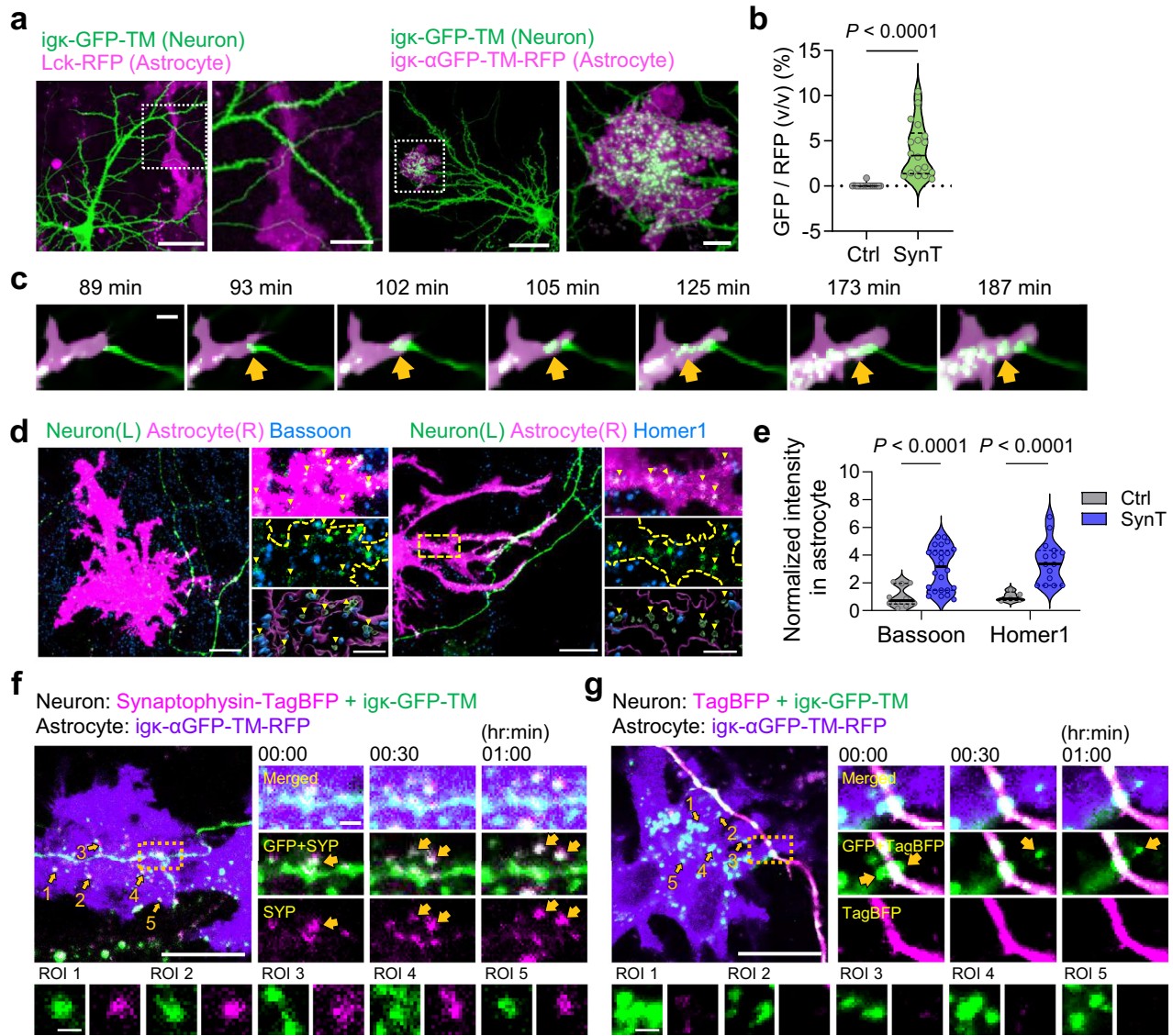

**Fig. 2 | SynTrogo induced by engineered neuron-astrocyte interactions.**
**a** Fluorescence images of SynTrogo (SynT) of cultured neurons by astrocytes mediated by ligand-receptor interactions. Scale bars, 50 μm (left), 10 μm (right). **b** Quantification of GFP uptake by astrocytes during SynT. *n* = 18 astrocytes per group. **c** Time-lapse images of ligand-labeled neuronal membrane fragment engulfment by a receptor-expressing astrocyte. Scale bar, 5 μm. **d** Fluorescence and 3D reconstruction images showing uptake of endogenous synaptic molecules (left, Bassoon; right, Homer1) by astrocytic SynT. Scale bar, 20 μm (large images), 5 μm (small images). **e** Quantification of Bassoon and Homer1 signals in receptor-expressing astrocytes. *n* = 10 (Ctrl), 24 (SynT) cells for Bassoon; 5 (Ctrl), 15 (SynT)

cells for Homer1. Time-lapse images of cultured neurons co-expressing the ligand with either **f** Synaptophysin (SYP)-TagBFP or **g** TagBFP, together with receptor-expressing astrocytes under SynTrogo conditions. Scale bars, 20 μm (left), 5 μm (right, magnified views), and 2 μm (bottom, magnified views). Yellow arrows in the left images indicate regions of interest (ROI) showing engulfed GFP-positive vesicles with or without TagBFP signals. Time is indicated as hours:minutes. Data are shown as median with interquartiles range; dotted lines in (**b**, **e**). Statistical significance was determined using an unpaired t-tests for (**b**) and multiple unpaired t-test for (**e**). Source data are provided as a Source Data file.

ligand in cortical neurons and the receptor in striatal astrocytes. Consistent with our observations in the CA3-CA1 hippocampal circuit, GFP ligand signals derived from cortical axons were redistributed into striatal astrocytic territories (Supplementary Fig. 13), suggesting that SynTrogo may be broadly applicable across diverse brain regions.

**Cellular mechanisms of SynTrogo underlying reduced synaptic connectivity**
To determine the ultrastructural basis of the colocalized signals of ligand and astrocytes (Fig. 3) and investigate the cellular mechanism of SynTrogo, we analyzed the ultrastructure of neurons and astrocytes using correlative light and electron microscopy (CLEM) imaging[31]. For this analysis, we first identified RFP-expressing astrocytes in the CA1

region that closely contacted GFP-positive axons using confocal microscopy (Fig. 4a). These regions of interest were then traced in serial electron microscope (EM) images to correlate ultrastructural features with the corresponding fluorescence signals (Supplementary Movies 6 and 7). This analysis revealed that the SynTrogo group exhibited a significantly increased percentage of axon–astrocyte interface (AAI) compared to the control group where astrocytes expressed Lck-RFP while lacking the receptor (Fig. 4b, c). To examine whether astrocytes preferentially bind to specific subregions of the axonal membrane, we measured the AAI ratio separately for boutons and axonal shafts. In the control group, the AAI ratio was comparable between boutons and shafts (Fig. 4d). By contrast, in the SynTrogo group, boutons exhibited a significantly higher AAI ratio than shafts,

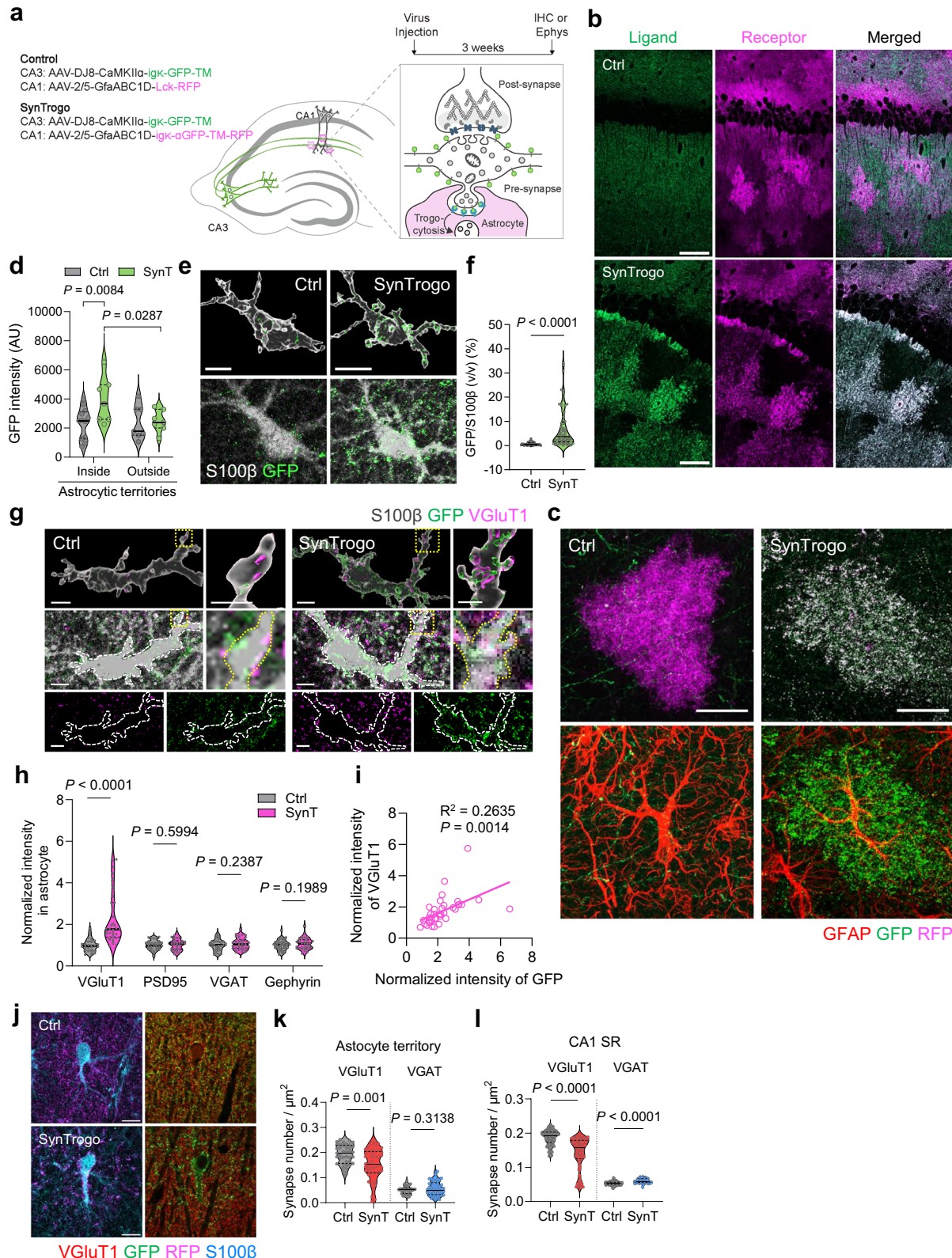

and both boutons and shafts showed increased AAI ratios relative to the control group (Fig. 4d, e). Moreover, fluorescence imaging revealed punctate appearance of ligands at discrete regions along the axons under SynTrogo conditions (Fig. 4f), suggesting localized Syn-Trogo events.

Notably, we found no significant correlation between bouton/shaft surface area and AAI ratio in either the control or SynTrogo group

(Fig. 4g, h). Additionally, in the control group, the AAI ratio was not significantly related to the number of synapses on the axon (Fig. 4i). Taken together, these results indicate that under SynTrogo conditions, astrocytes established more extensive contacts with boutons than with shafts, although this difference was not related to bouton/shaft size.

We also observed that membranes at sites of neuron–astrocyte contact in the SynTrogo group were significantly thicker (Mt =

**Fig. 3 | SynTrogo of CA3 presynaptic axons by CA1 astrocytes. a** Schematic of SynTrogo (SynT) targeting CA3 axons and CA1 astrocytes. An adeno-associated virus (AAV) expressing the ligand under the CaMKIIα promoter was injected into CA3. AAVs expressing membrane-anchored RFP (Lck-RFP) or the receptor under the GfaABC1D promoter were injected into CA1. Brains were analyzed by immunohistochemistry 3 weeks post-injection. **b** Fluorescence images of ligand and receptor signals in CA1. Scale bars, 50 µm. **c** Ligand signals with GFAP and either membrane marker (control) or receptor (SynT). Scale bars, 20 µm. **d** GFP intensity inside vs. outside astrocyte territories. $N = 3$ mice per group; $n = 12$ (inside), 7 (outside) region of interest (ROIs) (Ctrl); 7 (inside), 7 (outside) ROIs (SynT). **e** 3D-rendered astrocytes in Ctrl or SynT. Scale bar, 5 µm. **f** Volume ratio (v/v) of GFP ligand to astrocytic soma (S100β). $N = 3$ mice per group; $n = 42$ (Ctrl), 64 (SynT) astrocytes. **g** Fluorescence and 3D-rendered images showing GFP ligand and VGluT1 within astrocytic territories. Scale bar, 5 µm (large); 2 µm (magnified views).

**h** Normalized intensity of synaptic markers within astrocytic soma territories. $N = 4$ mice per group; VGluT1: $n = 30$ (Ctrl), 33 (SynT); PSD95: $n = 31$ (Ctrl), 29 (SynT) cells; VGAT: $n = 33$ (Ctrl), 35 (SynT) cells; Gephyrin: $n = 40$ (Ctrl), 33 (SynT) cells. **i** Correlation between GFP ligand and presynaptic marker levels within astrocytic territories ($n = 36$ cells). **j** Fluorescence images of synaptic marker. Scale bars, 10 µm. VGluT+ and VGAT+ synapse numbers within **k** astrocyte territories and **l** CA1 SR layer. $N = 3$ mice per group. For **k**: VGluT1, $n = 39$ (Ctrl), 43 (SynT); VGAT: $n = 26$ (Ctrl), 40 (SynT) astrocytes. For **l**, VGluT1: $n = 103$ (Ctrl), 99 (SynT); VGAT: $n = 99$ (Ctrl), 98 (SynT) imaging areas. Data are shown as median with interquartile range in (**d**, **f**, **h**, **k**, **l**). Statistical significance was determined using two-way ANOVA with Fisher's LSD multiple-comparison test for (**d**); unpaired t-test (**f**, **k**, **l**); multiple unpaired t-tests for (**h**); and simple linear regression with F test for (**i**). Source data are provided as a Source Data file.

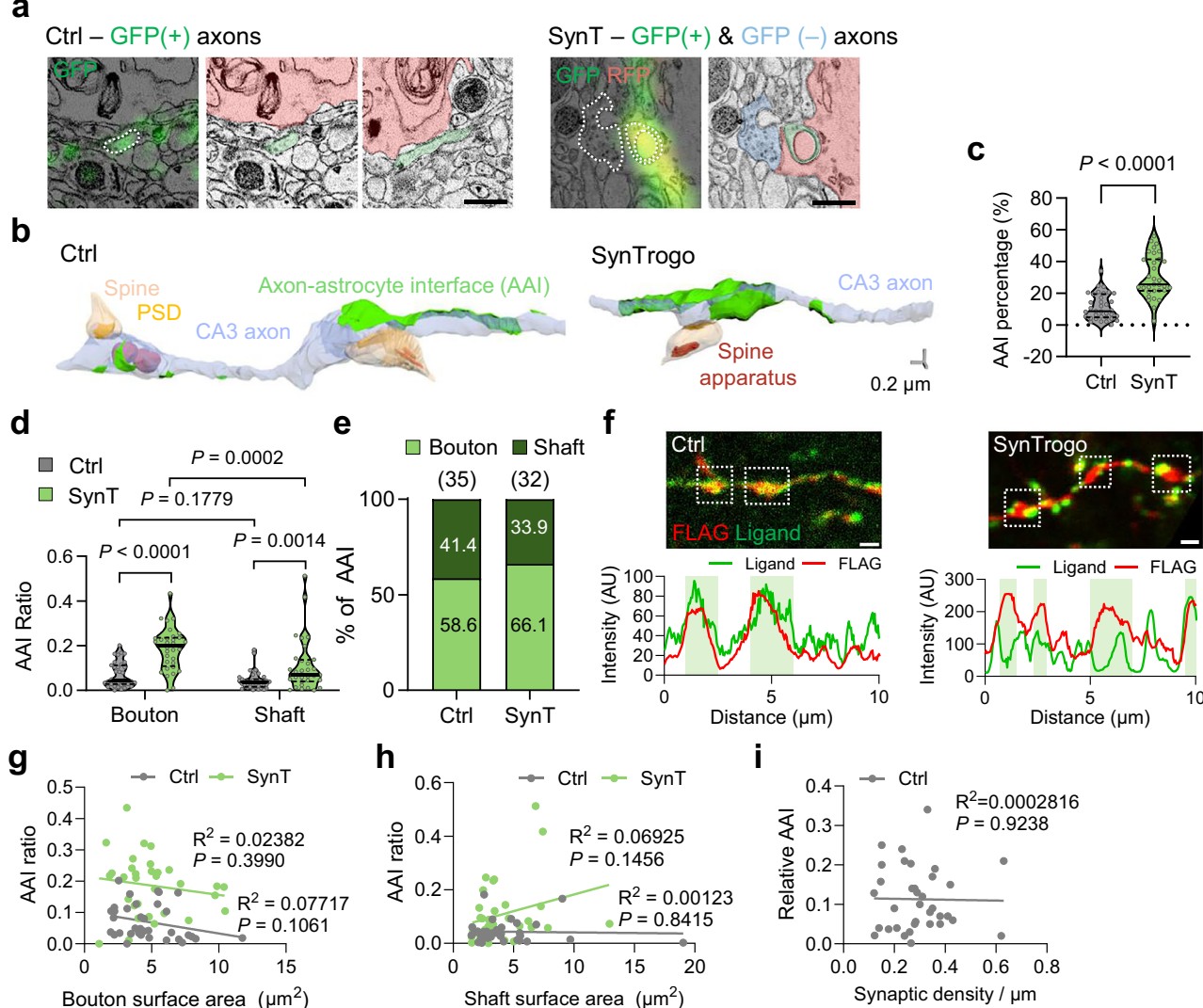

**Fig. 4 | Ultrastructural analysis of axon-astrocyte interfaces under SynTrogo. a** Representative CLEM images showing GFP(+) axons in the control group and both GFP(+) and GFP(−) axons in the SynTrogo (SynT) group. Astrocytic territories (pink), GFP(+) axons (green), and GFP(−) axons (blue) are pseudo-colored. Dotted lines indicate axonal boundaries. Scale bars, 0.5 µm. **b** 3D-rendered images showing axons, spines, and axon-astrocyte interfaces (AAI). Scale bar, 0.2 µm. **c** Quantification of AAI percentage. $N = 3$ (Ctrl), 3 (SynT) mice; $n = 35$ (Ctrl), 32 (SynT) axons. **d** Quantification of axon-astrocyte interfaces (AAI) associated with boutons and shafts in control and SynT conditions. $N = 3$ (Ctrl), 3 (SynT) mice; $n = 35$ (Ctrl), 32 (SynT) axons. **e** Quantification of the proportion of bouton- versus shaft-associated AAI as a percentage of total AAI events. **f** (top) Representative images of Schaffer

collateral axons visualized by expression of membrane-anchored ligand (igκ-GFP-TM) and cytosolic protein (smFP-FLAG) in control (left) and SynT (right) conditions. White-outlined boxes highlight axonal bouton regions. Scale bar, 1 µm. (bottom) Line profile analysis corresponding to control and SynT axons. Correlations between AAI ratio and bouton (**g**) or shaft (**h**) surface area. For **g**, $n = 35$ (Ctrl), 32 (SynT) boutons. For **h**, $n = 35$ (Ctrl), 32 (SynT) shafts. **i** Correlation between relative AAI size and synaptic density in control axons. $N = 3$ (Ctrl), 3 (SynT) mice. $n = 28$ (Ctrl), 26 (SynT) AAI. Data are shown as median with interquartiles range in (**c**, **d**). Statistical significance was determined using unpaired t-test for **c**; two-way ANOVA with Fisher's LSD multiple-comparison test for (**d**); and simple linear regression with F-test for (**g**–**i**). Source data are provided as a Source Data file.

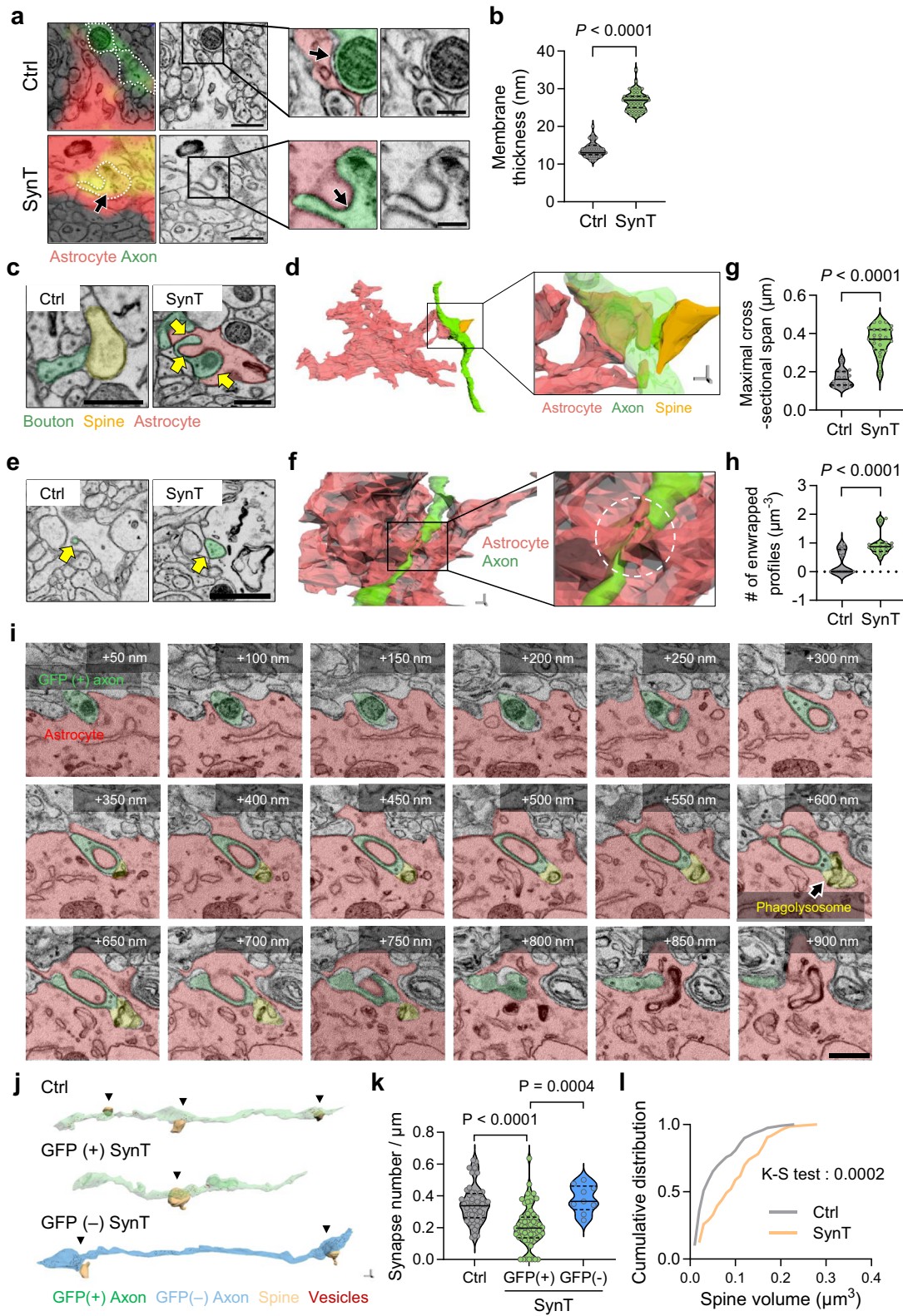

26.96 ± 0.55 nm) compared to those in the control group (Mt = 14.68 ± 0.40 nm), which exhibited clear intercellular separation (Fig. 5a, b). This thicker membrane structure, along with a narrowed intermembrane space, indicates tight associations between presynaptic boutons and astrocytic processes, rather than membrane fusion.

Closely apposed regions between astrocytic processes and presynaptic boutons frequently exhibited interdigitated or mutually indented membrane profiles (Fig. 5a, c), reminiscent of the interface of HeLa and HEK293T cells (Supplementary Fig. 3b). Three-dimensional (3D) EM reconstructions revealed membrane configurations in which astrocytic processes appeared to constrict portions of the presynaptic

**Fig. 5 | Ultrastructural features of neuron-astrocyte contact regions under SynTrogo. a** EM images showing tightly apposed membranes between presynaptic boutons and astrocytic processes with interdigitated profiles. Scale bars, 0.5 μm (large), 0.2 μm (magnified views). **b** Membrane thickness at neuron-astrocyte contact sites ($N = 3$ mice per group; $n = 45$ Ctrl, 47 SynT contacts). **c** EM images of presynaptic bouton-astrocyte membrane association (green, boutons; yellow, spine; red, astrocyte). Scale bar, 0.5 μm. **d** 3D-rendered EM images showing astrocytic processes closely apposed to and constricting segments of presynaptic axons, continuous with parent axons. Scale bar, 0.2 μm. **e** EM images of a membrane-enwrapped axonal profile within astrocytic processes containing presynaptic-derived material. Scale bar, 1 μm. **f** 3D reconstructions showing astrocytic extensions partially surrounding presynaptic structures; profiles remain continuous with parent axons and do not represent isolated inclusions. Scale bar, 0.2 μm. Quantification of **g** maximal cross-sectional span of enwrapped profiles and **h** enwrapped profile density ($N = 3$ mice per group; **g**: $n = 16$ Ctrl, 25 SynT profiles; **h**: $n = 18$ Ctrl, 17 SynT reconstructed volumes). **i** Serial EM images showing phagolysosome-like structures near SynTrogo-associated membrane configurations. Scale bars, 1 μm. **j** 3D-rendered EM images of ligand-positive or -negative axonal processes with spines (arrowheads). Scale bar, 0.2 μm. **k** Synaptic density along axons. Control axons and GFP (−) axons in the SynT group were selected based on close astrocytic apposition (≤100 nm) to minimize potential bias ($N = 3$ mice per group; $n = 48$ Ctrl, 53 GFP+ SynT, 9 GFP− SynT axons). **l** Cumulative distribution of spine volumes for spines contacting ligand-coupled axons ($N = 4$ mice per group; $n = 121$ Ctrl, 73 SynT spines). Data are shown as median with interquartile range in (**b, g, h, k**). Statistical significance was determined using unpaired t-tests in (**b, g, h**); one-way ANOVA with Tukey's test (**k**); and Kolmogorov-Smirnov test (**l**). Source data are provided as a Source Data file.

compartment (Fig. 5d and Supplementary Fig. 14a). At sites where these opposing membranes approached one another, we observed membrane-enwrapped axonal profiles in 2D EM images (indicated by green areas in Fig. 5e) containing vesicular structures and, in some cases, mitochondria, with a maximal cross-sectional span of ~335 nm and often connected by narrow membrane continuities (Fig. 5e–g and Supplementary Fig. 14b). Although similar structures were present in control conditions, their size and frequency were significantly higher under SynTrogo conditions (Fig. 5g, h). The vesicular elements within these membrane-enwrapped regions measured 35.63 ± 5.20 nm (Supplementary Fig. 14c, d), comparable to the typical diameter of presynaptic vesicles of hippocampal neurons[32]. 3D-reconstructions further verified that these membrane-enwrapped regions were continuous with axonal compartments (Fig. 5f; Supplementary Fig. 14e, f; Supplementary Movies 6 and 7), indicating that they do not represent fully internalized or severed structures. Of note, the membranes surrounding these regions measured 27.9 ± 0.69 nm in thickness, a value consistent with that observed at neuron-astrocyte contact sites (Fig. 5b), supporting that these configurations correspond to close membrane apposition between axonal and astrocytic processes. A similar enwrapped configuration was observed in SynTrogo of HEK293T cells by HeLa cells (Supplementary Fig. 3d), raising the possibility that related morphological transitions may occur across different cell types.

To further investigate where SynTrogo events occur, we examined 53 axonal segments identified in 3D-reconstructed EM images and found that 25 displayed interdigitated membrane profiles and 8 contained membrane-enwrapped configurations (Supplementary Fig. 14g). These structures were observed at both boutons and shafts (Supplementary Fig. 14h), indicating that SynTrogo-associated morphological features are not confined to specific axonal subregions. This pattern is consistent with the AAI analysis, which revealed that SynTrogo significantly increased astrocyte contacts with boutons and shafts compared to the control (Fig. 4d). In addition, analysis of enwrapped compartments showed that 62.5% contained vesicles and/or mitochondria, suggesting that these structures are frequently associated with presynaptic cytosolic components (Supplementary Fig. 14i). This observation aligns with previous reports of amebic trogocytosis, in which cytoplasmic material—including mitochondria—can be internalized[33]. Taken together, our findings show that SynTrogo-associated membrane interactions occur across diverse axonal domains and are associated with the encapsulation of presynaptic cytosolic organelles, a process shaped by the spatial proximity of astrocytes and axons.

In some cases, we observed phagolysosome-like structures in regions exhibiting interdigitated, membrane-approaching, or membrane-enwrapped configurations, suggesting that degradative compartments can appear alongside these morphologies (Fig. 5i and Supplementary Fig. 14a, b), although these observations do not necessarily imply that degradation invariably follow. This pattern is in line with the recent findings showing that Bergmann glial cells in the cerebellum engage neuronal compartments that display local deformation and are associated with multilamellar phagosomes at similar membrane configurations[16]. Notably, GFP signals were not completely lost but remained detectable in CLEM analysis even when membranes appeared closely apposed or partially enclosed (Fig. 5i and Supplementary Fig. 14f). Given that engulfed GFP ligands undergo multiple vesicle trafficking steps before reaching lysosomes in cultured cells (Fig. 1j and Supplementary Fig. 6a), and that GFP fluorescence is partially retained following lysosomal fusion (Supplementary Fig. 6e, f), these results support our confocal imaging data showing residual GFP signals within astrocytic territories under SynTrogo conditions. Collectively, we describe a set of morphological features—intercellular contact, membrane indentation and constriction, and partial membrane enclosure—that may represent SynTrogo-associated membrane interactions, providing a conceptual framework of how this process could contribute to astrocyte engagement with axonal components (Supplementary Fig. 14j).

3D reconstructions indicate that fluorescence ligand puncta overlapping with astrocytic territories correspond to axonal segments closely apposed to or partially wrapped by astrocytic processes rather than severed synaptic fragments. Taken together, our current dataset does not provide direct evidence of synaptic elimination. Nevertheless, these features resemble certain morphological patterns reported in studies of naturally occurring glia-mediated synaptic pruning[13,15,16,18].

Notably, 3D-reconstructed EM analyses of ligand-expressing axons contacting receptor-expressing astrocytes revealed that these axons contained fewer synapses compared to controls (Fig. 5j, k and Supplementary Figs. 15a, b and 16). In contrast, neighboring ligand-negative axons under SynTrogo conditions showed synapse densities comparable to those in the control group (Fig. 5k and Supplementary Fig. 16), indicating that the effect was confined to ligand-expressing axons in the same spatial context. There was also a significant decrease in the proportion of smaller spines (Fig. 5l), raising the possibility that synapses of different sizes vary in their susceptibility to astrocytic SynTrogo. Such differences could reflect variability in the levels or organization of synaptic adhesion molecules, which may influence the physical stability of pre- and postsynaptic connections.

## Electrophysiological properties of the CA3-CA1 connection under SynTrogo conditions

To examine the functional effects of SynTrogo on the hippocampal CA3−CA1 circuit, we analyzed miniature excitatory postsynaptic currents (mEPSCs) and found a significant decrease in frequency without changes in amplitude (Fig. 6a–c), consistent with reduced synapse numbers observed in fluorescence and EM analyses (Figs. 3k, l and 5k). In contrast, analysis of spontaneous excitatory postsynaptic currents (sEPSCs) and spontaneous inhibitory postsynaptic currents (sIPSCs) showed that SynTrogo significantly increased the frequency of sEPSCs

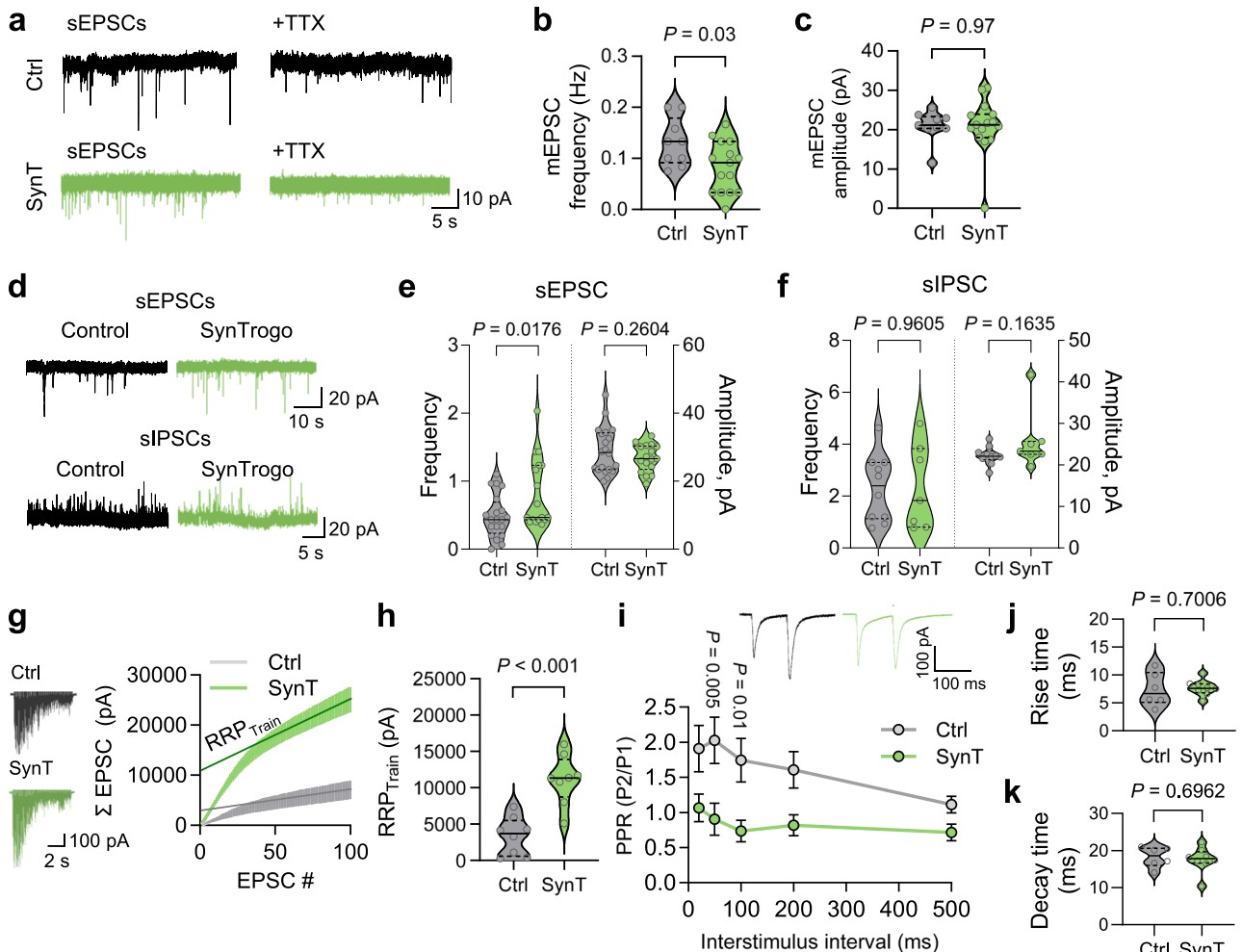

**Fig. 6 | Reduced synaptic connectivity of target neural circuit via SynTrogo.** **a** Representative traces of spontaneous (left) and miniature (right) EPSCs in CA1 pyramidal neurons. Quantification of **b** frequency and **c** amplitude of miniature EPSC. $N = 4$ (Ctrl), 6 (SynT) mice; $n = 9$ (Ctrl), 15 (SynT) cells. **d** Representative traces of spontaneous EPSCs and IPSCs in control and SynTrogo (SynT) groups. Graphs showing **e** frequency and amplitude of sEPSC, and **f** frequency and amplitude of IPSCs. $N = 6$ (Ctrl), 5 (SynT) mice; $n = 20$ (Ctrl), 17 (SynT) cells for (**e**). $N = 3$ (Ctrl), 3 (SynT) mice; $n = 10$ (Ctrl), 7 (SynT) cells for (**f**). **g** EPSCs traces elicited by 100 stimuli at 20 Hz (left). Average cumulative EPSC amplitudes (right). $N = 4$ (Ctrl), 5 (SynT) mice; $n = 8$ (Ctrl), 8 (SynT) cells. **h** Estimation of readily-releasable pool (RRP$_{train}$).

$N = 4$ (Ctrl), 5 (SynT) mice; $n = 8$ cells per group. **i** Paired-pulse ratio (PPR) as a function of interstimulus intervals. Representative PPR traces from Ctrl and SynT groups shown in upper right inset. AMPA receptor-mediated current kinetics showing (**j**) rise time and (**k**) decay time. Kinetic parameters were measured from data shown in (**j**). $N = 3$ (Ctrl), 4 (SynT) mice; $n = 6$ (Ctrl), 8 (SynT) cells. Data are shown as median with interquartile range in (**b, c, e, f, h, j, k**), and as the mean ± s.e.m. in (**g, i**). Statistical significance was determined using unpaired t-test for (**b, c, e, f, h, j, k**), and two-way repeated-measures ANOVA followed by Sidak's multiple comparison for (**i**). Source data are provided as a Source Data file.

without affecting their amplitude, and did not alter either the frequency or amplitude of sIPSCs (Fig. 6d–f), indicating that SynTrogo influences excitatory presynaptic function.

We further found that SynTrogo increased the readily releasable pool (RRP) of synaptic vesicles (Fig. 6g, h) and the release probability of presynaptic neurons (Fig. 6i–k). Under these conditions, the intrinsic properties of CA1 neurons were unaltered, and their responsiveness to input intensity was comparable between groups (Supplementary Fig. 17).

### Remodeling of remaining synapses associated with reduced synaptic connectivity

The paradoxical increase in presynaptic function (Fig. 6) despite reduced synaptic connectivity prompted us to investigate how neurons adapt to SynTrogo-mediated synaptic reduction by examining the structural and functional features of the remaining synapses. CLEM analysis revealed that GFP-labeled boutons in the SynTrogo group were significantly larger than those in the control group lacking

receptor expression (Fig. 7a, b), while neighboring non-labeled boutons in the SynTrogo group exhibit bouton volumes comparable to those of control boutons (Fig. 7b). These results suggest that the increase in bouton size is specific to ligand-expressing axons.

Consistent with this, analysis of bouton size distributions revealed an overall shift toward larger boutons in the SynTrogo group compared with the control group, including size ranges not observed under control conditions, while the proportion of small boutons was markedly reduced (Fig. 7c). Boutons within this larger size range contained higher numbers of presynaptic vesicles, both total and docked, than those in the control group (Fig. 7d, e). Furthermore, mitochondrial volume within boutons was increased in the SynTrogo group, while that outside boutons remained comparable across groups (Fig. 7f, g), suggesting this alteration may support the energy demand for enhanced synaptic transmission[34]. Together with the electrophysiological analyses above, these findings suggest that SynTrogo-mediated synaptic reduction is associated with both structural and functional strengthening of the remaining presynapses.

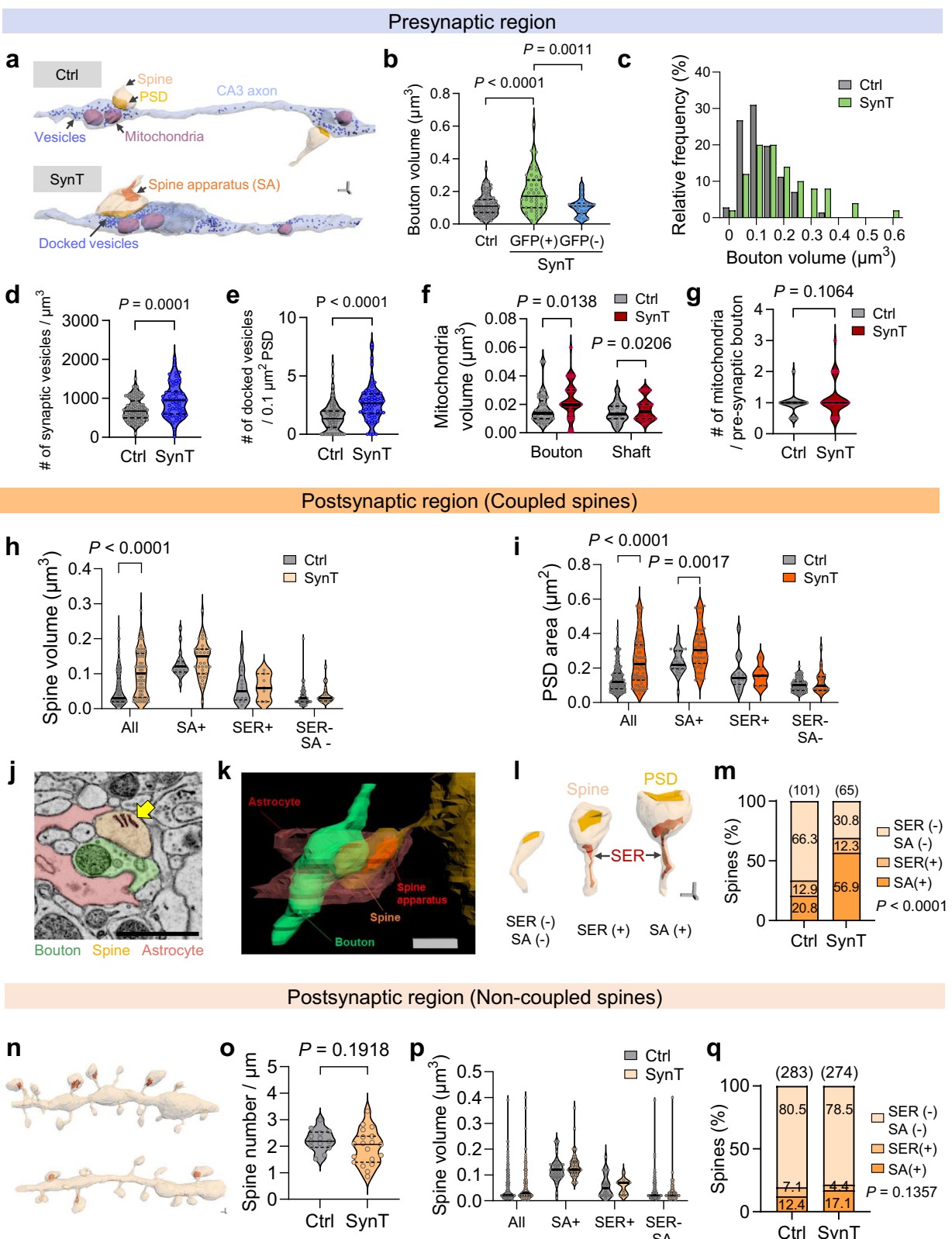

On the postsynaptic side, the spine volume and PSD area were significantly larger in the SynTrogo group compared to the control (Fig. 7h-i). We also observed an increased proportion of spines containing the spine apparatus (SA) (Fig. 7j–m and Supplementary Fig. 15c), an extended endoplasmic reticulum (ER) structure forming stacked disks in spines[35,36]. Given previous studies suggesting that smooth endoplasmic reticulum (SER) may serve as an intermediate for SA generation[36], we analyzed two additional populations: SA(−) SER(−), and SA(−) SER(+) spines. The proportion of SA(−) SER(−) spines decreased, while the proportion of SA(−) SER(+) spines remained unchanged, and was accompanied by an increase in SA(+) spines (Fig. 7m). Notably, the larger PSD area was predominantly associated with SA(+) spines (Fig. 7i). In dendrites not engaged in SynTrogo, spine number, type distribution, and volume were

**Fig. 7 | Structural remodeling of pre- and post-synaptic compartments following SynTrogo-associated synaptic reduction. a** 3D-rendered EM images of synaptic contents. Scale bar, 0.2 μm. Bouton volume (**b**) volume distribution (**c**) total synaptic vesicles (**d**) and docked vesicles (**e**) in presynaptic boutons. $N = 3$ mice per group; **b, c:** $n = 71$ Ctrl, 50 GFP+ SynT, 20 GFP− SynT boutons; **d:** $n = 85$ Ctrl, 59 SynT boutons; **e:** $n = 94$ Ctrl, 61 SynT boutons. **f** Mitochondrial volume in axonal boutons or shafts. **g** Number of mitochondria in presynaptic boutons ($N = 3$ mice per group; Ctrl: $n = 37$ bouton, 16 shaft; SynT: $n = 39$ bouton, 22 shaft mitochondria). Spine volume (**h**) and PSD area (**i**) in dendritic spines coupled with ligand-labeled axons ($N = 3$ mice per group). For **h:** Ctrl, $n = 25$ SA(+), 14 SER(+), 78 SER(−) SA(−); $n = 42$ SA(+), 8 SER(+), 21 SER(−) SA(−) spines. For **i:** Ctrl, $n = 25$ SA(+), 14 SER(+), 70 SER(−) SA(−); SynT, $n = 42$ SA(+), 6 SER(+), 22 SER(−) SA(−) spines. **j** EM image of spine apparatus (arrow). Scale bar, 1 μm. 3D reconstructions of astrocyte process, bouton, and spine (**k**), and representative SER(−) SA(−), SER(+), and SA(+) spines (**l**). Scale bar, 0.5 μm (**k**), 0.2 μm (**l**). **m** Percentage distribution of SER(−) SA(−), SER(+), and SA(+) spines ($n = 101$ Ctrl, 65 SynT spines). **n–q** Analysis of non-coupled dendrites. **n** 3D-rendered images of dendrites and spines. Scale bar, 0.2 μm. **o** Spine density of non-coupled dendrites ($n = 13$ Ctrl, 19 SynT dendrites). **p** Spine volume (Ctrl: $n = 35$ SA(+), 20 SER(+), 228 SER(−) SA(−); SynT: $n = 47$ SA(+), 12 SER(+), 215 SER(−) SA(−) spines). **q** Percentage distribution of spine subtypes (chi-square). Data are presented as median with interquartile range in (**b, d–i, o, p**). Statistical significance was determined using one-way ANOVA with Tukey's multiple-comparison test for (**b**); unpaired t-test for (**d, e, g, o**); two-way ANOVA with Sidak's multiple-comparison test for (**f, h, i, p**); and chi-square test for (**m, q**). Source data are provided as a Source Data file.

comparable to those in the control group (Fig. 7n–q). Taken together, these results indicate that postsynaptic spines under SynTrogo conditions display structural features compatible with remodeling, suggesting that the effects of SynTrogo may extend to postsynaptic as well as presynaptic elements.

## Enhanced synaptic plasticity and brain function following SynTrogo-mediated synaptic remodeling

Given the crucial roles of SA in regulating intracellular calcium signals, protein synthesis, and receptor trafficking[37], we hypothesized that SynTrogo-mediated synaptic remodeling could impact synaptic plasticity. Indeed, delivering theta-burst electrical stimulations to the Schaffer collateral pathway resulted in a sustained increase in long-term potentiation (LTP), as evidenced by a significantly elevated EPSC amplitude for 70 min (Fig. 8a, b).

To evaluate the functional significance of SynTrogo-mediated synaptic remodeling on cognitive processes, we conducted contextual fear conditioning experiments three weeks after virus injection. Using mild foot shocks (0.3 mA), mice in the SynTrogo group exhibited enhanced memory in both recent (day 2) and remote (day 23) memory tests compared to the control group, in which astrocytes lacked receptor expression (Fig. 8c, d). When challenged with stronger aversive stimuli (0.5 mA), the SynTrogo group maintained consistent freezing levels across both tests, while control mice showed significant memory decay in the remote test (Fig. 8e, f).

To assess whether these persistent fear memories remain amenable to extinction, we implemented a spaced extinction protocol consisting of two daily sessions separated by 2 h, conducted over four consecutive days[38] (Fig. 8e). During extinction training, control mice showed significant reduction only in the first extinction session (Ext1), likely due to their already decreased freezing levels in remote memory test. In contrast, the SynTrogo group, which maintained high freezing levels in remote memory test, showed significant reductions throughout the extinction period. Both groups exhibited similarly low freezing levels after extinction and in the spontaneous recovery test two weeks later (Fig. 8g, h). These results indicate that despite their persistent remote memory, SynTrogo mice remain capable of extinction learning. Notably, under the identical conditions, no significant changes in locomotion, anxiety, and working memory were observed (Supplementary Fig. 18).

To investigate the underlying synaptic mechanisms, we examined AMPA (α-amino-3-hydroxy-5-methyl-4-isoxazolepropionic acid) and NMDA (N-methyl-D-aspartate) receptor currents—key indicators of synaptic strength and maturation[39]—both in the absence and the presence of fear conditioning (Fig. 8i). In the absence of conditioning, neurons under SynTrogo conditions showed significantly reduced AMPA currents and AMPA/NMDA ratio compared to control neurons. However, these differences in both parameters vanished after fear learning (Fig. 8j–m), with both groups showing comparable receptor current profiles. These results suggest that SynTrogo-mediated synaptic reduction and remodeling could lead to a more plastic state, which might facilitate enhanced synaptic modification during learning.

Based on our EM analysis showing that SynTrogo significantly altered the proportion of SA(+) spines (Fig. 7m), we examined the role of these spines in AMPA receptor dynamics under fear conditioning. When we stained and analyzed GluA1 (an AMPA receptor subunit) and synaptopodin (a marker of the SA) signals with or without conditioning (Supplementary Fig. 19a, b), we found that GluA1 levels were initially decreased in the SynTrogo group without conditioning, but recovered to control levels after fear conditioning (Supplementary Fig. 19c, g). In contrast, synaptopodin levels in the SynTrogo group remained consistently elevated regardless of conditioning, compared to the control group (Supplementary Fig. 19d, h). Moreover, the number and proportion of spines containing both GluA1 and synaptopodin were reduced in the SynTrogo group without conditioning, but these values recovered to levels similar to those in the control group after conditioning (Supplementary Fig. 19e–f, i–j). These findings are consistent with our electrophysiological analysis and suggest that SA(+) spines under SynTrogo conditions may contribute to AMPA receptor trafficking in this experimental paradigm.

Collectively, our results demonstrate that SynTrogo-mediated synaptic remodeling creates a more plastic synaptic state characterized by enhanced LTP, improved memory formation and retention, while preserving the capacity for memory extinction. These effects, which may in part involve SA(+) spine-associated AMPA receptor trafficking, suggest that synaptic reduction can paradoxically support both increased synaptic strength and preserved circuit flexibility.

## Discussion

In this study, we present SynTrogo (Synthetic Trogocytosis), a synthetic molecular approach to synaptic reduction and remodeling within a target neural circuit via engineered neuron-astrocyte interactions. By employing the high-affinity GFP-GFP nanobody pair to generate synthetic ligand and receptor proteins, we induced tight cell-cell contact when these components were displayed separately on two cultured cell populations, independent of intracellular signaling. Under these conditions, we observed a trogocytosis-like process in which membrane fragments and adjacent cytosolic material from GFP ligand-labeled cells were taken up by receptor-expressing cells. Applying SynTrogo to CA3 neuronal axons and CA1 astrocytes substantially altered the ultrastructure at their interfaces—partly resembling morphological features reported in glia-mediated synaptic pruning—and was accompanied by a significant decrease in synaptic connectivity. Under reduced synaptic connectivity, the remaining presynaptic and postsynaptic compartments exhibited coordinated structural and functional remodeling, aligning with enhanced synaptic plasticity and adaptive circuit- and behavioral-level outcomes.

Unlike most previous studies where synaptic pruning was modulated by neural activity[6,7]—creating a complex feedback loop between activity and structure that is difficult to dissect—SynTrogo allows these processes to be experimentally separated. Our findings indicate that

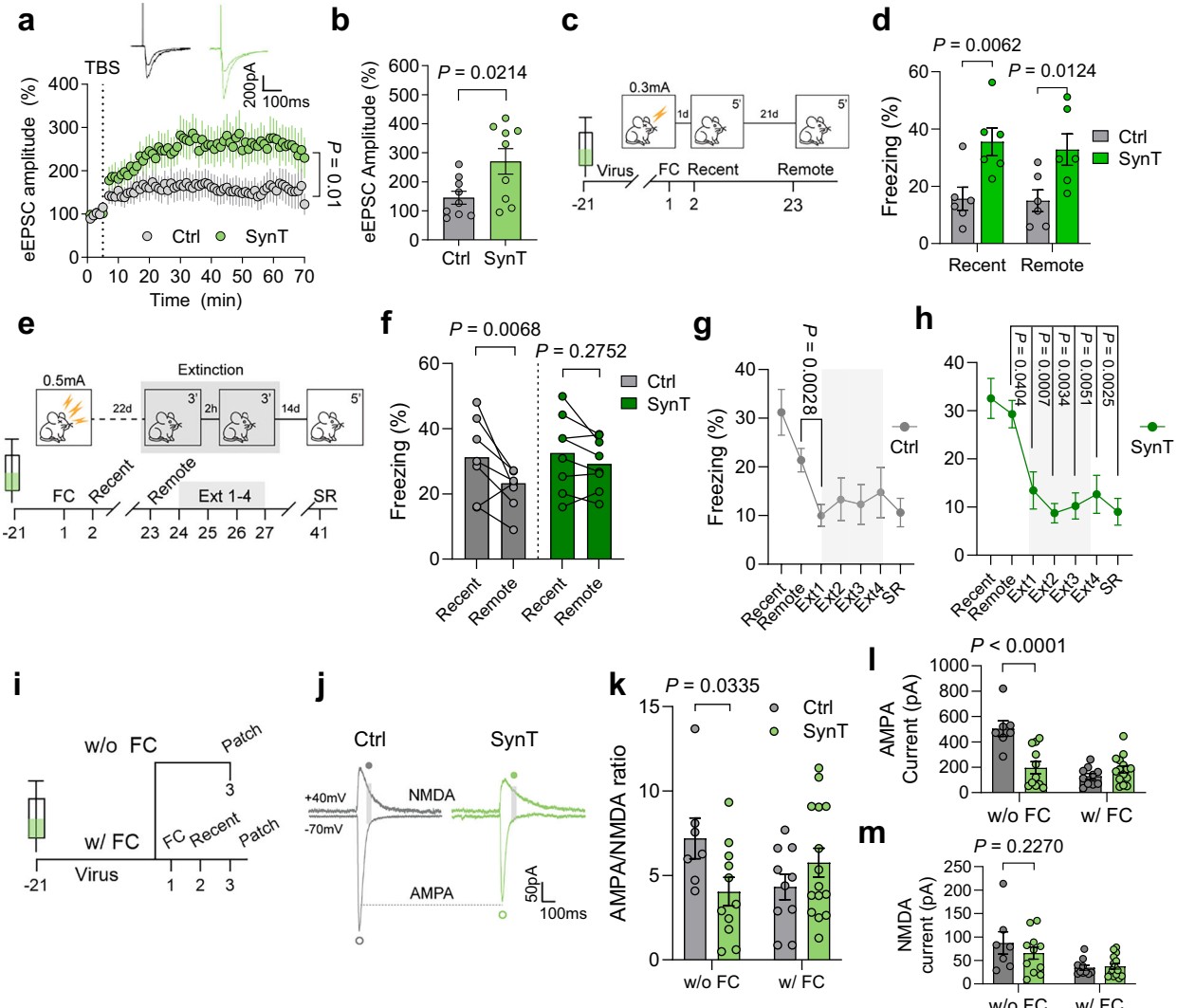

**Fig. 8 | Synaptic remodeling following SynTrogo-associated synaptic reduction supports enhanced synaptic plasticity and memory. a** LTP recording showing eEPSC amplitude changes following theta-burst stimulation. Representative traces are shown in the upper right. $N = 4$ (Ctrl), 5 (SynT) mice; $n = 9$ neurons per group. **b** Comparison of eEPSC amplitudes at $t = 60$ min. **c** Experimental timeline for viral injection, contextual fear conditioning and memory retrieval for (**d**). **d** Freezing behavior during memory recall following 0.3 mA conditioning. $N = 6$ mice per group. **e** Experimental timeline for (**f**–**h**). **f** Freezing behavior during recent and remote memory recall sessions (0.5 mA). $N = 7$ (Ctrl), 8 (SynT) mice. **g, h** Freezing behavior during memory recall and extinction process in **g** Ctrl and **h** SynT groups. SR refers to spontaneous recovery. $N = 7$ (Ctrl), 8 (SynT) mice. Asterisks indicate statistical significance compared to

the remote memory test. **i** Experimental timeline for (**j**–**m**). Representative traces of AMPA and NMDA currents **j**, measurements of **k** AMPA/NMDA ratio, **l** amplitude of AMPA current, and **m** amplitude of NMDA current. Without fear conditioning, $N = 3$ (Ctrl), 4 (SynT) mice; $n = 7$ (Ctrl), 11 (SynT) neurons. With fear conditioning, $N = 4$ (Ctrl), 6 (SynT) mice; $n = 10$ (Ctrl), 15 (SynT) neurons. Data are presented as mean ± s.e.m. Statistical significance was determined using two-way repeated-measures ANOVA with analysis of the time × group interaction for (**a**); unpaired t-test for (**b**); two-way repeated-measures ANOVA followed by Fisher's LSD multiple-comparison test for (**d, f**); two-way repeated-measures ANOVA followed by Tukey's multiple-comparison test for (**g, h**); and two-way ANOVA followed by Fisher's LSD multiple-comparison test for (**k**–**m**). Source data are provided as a Source Data file.

reduced synaptic connectivity is accompanied by broader changes in synaptic organization and properties of remaining connections, providing insight into cellular processes potentially relevant to homeostatic synaptic scaling[40].

SynTrogo is distinct from previously developed synthetic approaches for the structural remodeling of synaptic connections[41], including methods using engineered gap junction proteins[42], photon-based synaptic transmission[43], and neuropeptide-receptor coupling[44]. These approaches primarily focus on the generation, rather than deletion, of synapses and have been demonstrated mainly in invertebrate models. While other methods (e.g., As-PaRac1, SuperNova-Cofilin)[45,46] have been developed to perturb structural plasticity of synapses, they depend on

intracellular signaling within neurons rather than on neuron-glial interactions and have a limited temporal window for action.

By taking advantage of selective expression of ligands under the control of the CaMKIIα promoter, SynTrogo targeted CA3 excitatory neurons, thereby primarily influencing excitatory synapses with CA1 pyramidal neurons. This cell-type specificity, together with the natural proximity between synapses and astrocytic processes[20–22] enabled astrocytes to target synapses. However, our EM analyses revealed that SynTrogo occurs not only at boutons but also along axonal shafts (Supplementary Fig. 14h), indicating that while SynTrogo can serve as a tool to modulate synaptic connections, it may also influence non-synaptic axonal regions closely apposed to astrocytic processes.

To improve the spatial resolution of SynTrogo, various localization motifs or domains can be incorporated into ligand design[47]. For example, integrating our platform with the dual-eGRASP technique[48], which labels distinct synaptic populations through GFP reconstitution, may enable more precise targeting of synapses and facilitate analysis of their contributions to circuit function. Extending this concept further, SynTrogo could potentially be adapted to modulate synaptic connectivity that are not normally pruned, thereby enabling manipulation of otherwise stable connections beyond their physiological limits. Additionally, employing optogenetic or chemogenetic modules[47,49] in combination with activity-dependent or cell type-specific gene expression systems[2,3] holds promise for enhanced spatial and temporal control over SynTrogo-associated interactions while minimizing potential unintended effects.

The versatility of SynTrogo would allow its adaptation to diverse cell–cell interactions in the brain. In particular, one potential application would be to compare the capacities of astrocytes and microglia to interact with synaptic and perisynaptic structures under SynTrogo. Previous studies suggest that microglia may phagocytose larger portions of neuronal structures than astrocytes[50,51], which could result in differences in the extent or composition of material localized within glial cells. Such variability could influence the amount of residual synaptic or cellular components available for subsequent remodeling, thereby giving rise to distinct structural and functional outcomes. Thus, extending our approach to multiple cell types may provide a platform to assess how cell-type-specific modes of SynTrogo contribute to circuit-level reorganization across brain regions.

An important finding of this study is that the reduction in synaptic connectivity under SynTrogo conditions was associated with remodeling of both targeted presynaptic and coupled postsynaptic regions. In particular, presynaptic remodeling was associated with an increased proportion of SA(+) spines and a reduction in SER(−) SA(−) spines (Fig. 7m). These structural alterations were accompanied by enhanced synaptic transmission (Fig. 6d, e, g, i) and plasticity (Fig. 8a, b, k–m), as well as improved memory formation and retention (Fig. 8c–f). These results suggest that manipulating a subset of synaptic connections may contribute to changes at both circuit and cognitive levels.

Regarding the presynaptic remodeling, the observed increase in sEPSC frequency despite decreased mEPSC frequency raises the possibility of enhanced presynaptic neuronal activity, although direct recordings from presynaptic neurons will be required to confirm this possibility. Additionally, increased levels of inhibitory synapses in CA1 following SynTrogo (Fig. 3l) suggest that a reorganization of excitatory-inhibitory balance may be occurring while the functional significance of this change remains to be investigated.

At the postsynaptic side, we observed an increase in PSD area under SynTrogo from EM analysis (Fig. 7i). Despite this change typically being associated with increased AMPAR currents, sEPSC and mEPSC amplitudes remained unchanged (Fig. 6c, e). The comparable amplitudes of mEPSCs and sEPSCs argue against the involvement of multivesicular release or multisynaptic contacts, and instead support the interpretation that enhanced release probability at the remaining synapses underlies the increased synaptic efficacy. This apparent discrepancy may reflect a preparatory structural remodeling that may provide a scaffold that facilitates later AMPAR recruitment. Indeed, our data showed an initial reduction in AMPA currents followed by recovery after fear conditioning (Fig. 8k, l), together with similar trends observed in immunostaining analysis (Supplementary Fig. 19), suggesting that receptor incorporation occurs in an activity-dependent manner during learning.

The underlying mechanisms of synaptic remodeling may involve multiple dynamic and sequential processes. These include activity-dependent inter-synaptic competition for the redistribution of limited resources[36,52] via molecular diffusion and cytoskeleton-dependent vesicle trafficking, as well as reorganization of intracellular organelles (e.g., mitochondria and ER) within both pre- and postsynaptic compartments. Future studies using two-photon imaging with molecular perturbation approaches in live animals will be essential to elucidate the mechanisms underlying SynTrogo-associated synaptic remodeling.

Although SynTrogo enhanced memory performance in a contextual fear-conditioning paradigm, its effects varied depending on stimulus intensity (Fig. 8d, f). These findings suggest that the functional consequences of SynTrogo-associated synaptic remodeling may be context-dependent, and may not be uniformly beneficial across behavioral conditions. Clarifying the mechanisms and conditions that determine when such remodeling leads to adaptive outcomes across different neural circuits and behavioral contexts remains an important direction for future investigation.

The reduced synaptic connectivity through SynTrogo raises a question as to whether such manipulation could potentially disrupt essential connections. Despite high viral transduction efficiency (>70%) (Supplementary Fig. 11a–d), SynTrogo reduced synapse density by only ~27% (Fig. 3k, l and Fig. 5k) and displayed measurable variability across animals (see Source Data file), suggesting that factors beyond receptor–ligand expression levels may critically influence the extent of reduction and its functional outcomes. Such factors may include the activity-dependent accessibility of synapses to astrocytic processes, the heterogeneous astrocytic capacity for phagocytosis, and the structural stability of individual synaptic connections.

Although our time-lapse imaging in cultured cells clearly demonstrates complete internalization of cellular components by SynTrogo, analyses of brain tissues do not provide direct evidence that SynTrogo operates in vivo in the same manner as in vitro with respect to synaptic engulfment, owing to the limited z-resolution of light microscopy[13] and the lack of time-resolved information in both light and electron microscopy performed on fixed samples. Moreover, even if localized nibbling of axonal components were to occur in vivo, such partial removal of axons would not by itself substantiate the elimination of entire synapses. An additional possibility is that the observed reduction in synapse number may reflect perturbed synapse formation, potentially arising from limited access of axons to postsynaptic partners, due to extensive astrocyte-axon interaction under SynTrogo. As synaptic pruning and impaired synapse formation are not mutually exclusive, both mechanisms could contribute to the net decrease in synaptic connectivity. Therefore, longitudinal real-time imaging of SynTrogo events and associated synaptic changes in vivo conditions will be essential to fully elucidate processes and clarify the relative contribution of each mechanism.

Our findings that memory formation and retention are enhanced, and memory extinction is preserved, despite a reduction in synaptic connectivity, raise a question: why does the adult brain maintain an excess of synapses? One possible explanation is that maintaining an excess of synapses provides a buffer against sudden synaptic loss due to injury or disease, thereby supporting circuit resilience and facilitating adaptive remodeling under challenging conditions. It is also possible that the brain's evolutionary trajectory favors adaptability over strict optimization, ensuring that circuits remain flexible and responsive to dynamic environments.

Finally, the potential applications of SynTrogo in disease models deserve further exploration. In vivo, SynTrogo-mediated synapse reduction and remodeling may provide a tool for dissecting the cellular mechanisms underlying brain disorders. In conditions such as autism spectrum disorder and schizophrenia, where abnormal synapse numbers have been implicated, it remains unclear how these alterations contribute to disease progression[53]. By enabling manipulation of synapse density in vivo, SynTrogo offers a means to directly address this longstanding question. In the context of brain injury, such as stroke, previous studies have shown that synapse loss can activate homeostatic scaling mechanisms that strengthen remaining synapses

and promote circuit remodeling, thereby facilitating recovery[54,55]. SynTrogo could thus provide insights into how neuronal circuits reorganize after injury and may help inform strategies to optimize rehabilitation during periods of enhanced plasticity. In neurodegenerative diseases such as Alzheimer's disease[53,56], where synapse loss and impaired plasticity are early hallmarks, our findings may help lay the groundwork for identifying the molecular mechanisms underlying that support adaptive synaptic remodeling toward enhanced circuit function. Elucidating such mechanisms could guide future approaches aimed at restoring circuit function or resilience in neurodegenerative disease models.

Taken together, our approach advances understanding of the structure–function relationship in neural circuits and provides a foundation for developing connectome editing platforms[41,57].

## Methods

### Plasmid construction

Expression plasmids for piRFP670-N1 (ref. 58), pmiRFP670-N1 (ref. 59), pLAMP-iRFP703 (ref. 59) (Addgene plasmid #45457, #79987, and #79998 donated by V. Verkhusha), pmEGFP-N1, pFusionRed-C1 (ref. 60), GFP-Rab11a[61] (Addgene plasmid #54767, #54777, and #56444, donated by M. Davidson), pmScarlet-C1 (ref. 62) (Addgene plasmid #85042, donated by D. Gadella), pZac2.1-gfaABC1D-lck-GCaMP6f (Addgene plasmid #52924, donated by B. Khakh), pAAV-CaMKIIα-EGFP (Addgene plasmid #50469, donated by B. Roth), pAAV-hSyn-FLEx-mGFP-2A-Synaptophysin-mRuby[63] (Addgene plasmid #71760, donated by L. Luo), mTurquoise2-Golgi[64], mTurquoise2-ER[64] (Addgene plasmid #36205 and #36204, donated by D. Gadella, PMID: 22434194), GFP-EEA1wt[65] (Addgene plasmid #42307, donated by S, Corvera), and GFP-Rab7a[66] (Addgene plasmid #61803, donated by G. Volelts) were obtained from Addgene. To generate the ligand (igκ-mEGFP-TM) expression plasmid, the sequences encoding igκ signal peptide and the transmembrane domain of platelet-derived growth factor receptor (PDGFR) from the pDisplay vector (Invitrogen) were PCR-amplified and inserted at the N-terminus and C-terminus of mEGFP in mEGFP-N1 vector, respectively. mEGFP sequence in the ligand plasmid was replaced with the αGFP sequence[67] and FusionRed, mScarlet, or iRFP670 sequence was excised from pFusionRed-C1, pmScarlet-C1, or piRFP670-N1, respectively, and inserted at the C-terminus of the transmembrane domain to generate the receptor expression plasmids (igκ-αGFP-TM-FusionRed, igκ-αGFP-TM-mScarlet, or igκ-αGFP-TM-iRFP). mEGFP sequence in the ligand plasmid was replaced with the EYFP or ECFP sequence excised from pEYFP-N1 (Invitrogen) or pECFP-N1 (Invitrogen) vector by restriction enzyme digestion and ligation to generate igκ-EYFP-TM or igκ-ECFP-TM expression plasmids, respectively. Genes encoding GFP nanobodies (LaG-5, 6, 17, 18, and 42), mCherry nanobody (LaM4) (ref. 28), Ag1, Ag3, Nb1, and Nb3 (ref. 29) were synthesized after codon optimization for mammalian expression (Twist Bioscience). αGFP in igκ-αGFP-TM-FusionRed construct was replaced with Ag1 or Ag3, and PCR-amplified mEGFP was inserted at the C-terminus of TM after excision of FusionRed to generate igκ-Ag1-TM-mEGFP or igκ-Ag3-TM-mEGFP construct. αGFP in igκ-αGFP-TM-FusionRed construct was replaced with LaGs, LaM4, Nb1, or Nb3 to generated igκ-LaGs-TM-FusionRed, igκ-LaM4-TM-FusionRed, igκ-Nb1(or Nb3)-TM-FusionRed construct. To generate the Lck–mScarlet construct, the Lck coding sequence was PCR-amplified from the pZac2.1-gfaABC1D-Lck-GCaMP6f vector and inserted at the N-terminus of mScarlet. To generate iRFP-PM, we replaced EGFP from EGFP$_{CAAX}$ construct[67] with miRFP670. Construction of iRFP-Lifeact expression plasmid was described elsewhere[68]. For in vivo studies, the ligand and receptor constructs were cloned into the pAAV vector under the CaMKIIα or GfaABC1D promoter. To generate the pAAV-CaMKIIα-Synaptophysin-TagBFP vector, EGFP sequence in pAAV-CaMKIIα-EGFP was replaced with TagBFP, and Synaptophysin

sequence from pAAV-hSyn-FLEx-mGFP-2A-Synaptophysin-mRuby was PCR-amplified and inserted at the N-terminus of TagBFP. To generate organelle marker expression plasmids tagged with mTFP1, GalT-mTFP1, and mTFP1-KDEL constructs were derived by replacing the fluorescent proteins in mTurquoise2-Golgi and mTurquoise2-ER, respectively, with mTFP1. Coding sequences for EEA1, Rab7a, and Rab11a were obtained from GFP-EEA1wt, GFP-Rab7a, and GFP-Rab11a plasmids, and subcloned into the pmTFP1-C1 vector. For construction of LAMP1-TagBFP2, the LAMP1 coding region was extracted from pLAMP-iRFP703 and inserted into a TagBFP2-containing vector backbone. Synaptophysin-TagBFP was generated by subcloning the Synaptophysin coding sequence from pAAV-CaMKIIα-Synaptophysin-TagBFP into the mEGFP-N1 vector backbone, followed by replacement of the mEGFP sequence with TagBFP to reconstruct the Synaptophysin-TagBFP fusion construct. Oligonucleotide sequences used for plasmid construction are listed in Supplementary Data 1.

### Cell culture

HEK293T (ATCC, CRL-3216), HeLa (ATCC, CCL-2), Cos-7 (ATCC, CRL-1651), NIH3T3 (ATCC, CRL-1658), and BV2 cells (kindly provided by Dr. Dong Woon Kim, Kyung Hee University) were cultured in Dulbecco's Modified Eagle's Medium (DMEM; Gibco, #11965092, MA, USA) with 10% fetal bovine serum (FBS; Gibco, #16,000-044) at 37 °C in a 5% $CO_2$ humidified incubator. Hippocampal neurons were isolated from mouse embryos (E15-16) in Hank's balanced salt solution (HBSS; Gibco, #14175-095).

The hippocampi were treated with 0.05% trypsin for 5 min at 37 °C, filtered through a 0.4-μm filter, and plated on 50 μg/mL poly-D-lysine-coated (Millipore, #A003-E, MA, USA) 24-well plates (ibiTreat; ibidi, #82426, Gräfelfing, Germany). Neurons were maintained in Neurobasal medium (#21103-049) supplemented with 2% B-27, 2% N-2 supplements, 2 mM GlutaMAX (Gibco, #35050061), and 1000 units/mL penicillin-streptomycin and maintained at 37 °C in a 5% $CO_2$ humidified environment.

Primary cortical astrocytes were obtained from P0-P1 C57BL/6 male and female pups, dissociated, and plated on 50 μg/mL poly-D-lysine-coated 60-mm dishes. Astrocytes were grown in high-glucose DMEM supplemented with L-glutamine, 10% horse serum, 10% FBS, and 1000 units/mL penicillin-streptomycin at 37 °C in a 5% $CO_2$ humidified incubator.

Cell transfections were performed using Lipofectamine LTX (Invitrogen, #15338-100, CA, USA) following the manufacturer's protocol. For electroporation of cultured primary astrocytes, cells were trypsinized and resuspended in PBS. Collected cells in test tubes were resuspended in 10 μl of resuspension buffer (Invitrogen, #MPK1096) and transferred to the 10 μl Neon™ tips. The cells were transfected using the Neon® Transfection System (1100 V, 25 ms, 2 pulses) according to the manufacturer's instructions. Immediately after transfection, the cells were resuspended in culture medium, and plated into a PDL-coated 24-well plate (ibidi, #82427).

### Synthetic Trogocytosis between cultured cells

For the synthetic trogocytosis, ligand-expressing HEK293T cells cultured in 24-well plate were detached by trypsinization and collected in test tubes. These cells were resuspended in fresh media and added to receptor-expressing cells (HEK293T, HeLa, Cos-7, NIH3T3, BV2, astrocytes) cultured in 96-well plate, or hippocampal neurons cultured in 24-well plates. Live cell imaging was conducted immediately after mixing the ligand- and receptor-expressing cells. For the specific case of synthetic trogocytosis of cultured neurons by astrocytes, suspended astrocytes were added to neurons in 24-well plates and co-incubated for at least 6 h prior to imaging and analysis. Cytochalasin D (Sigma, #C8273), Blebbistatin (Sigma, #B0560), CK666 (Sigma, #SML0006), and smiFH2 (Sigma, #S4826) were applied 2 h prior to cell

mixing. PKH26 (Sigma, #PKH26GL) was applied to ligand-expressing HEK293T cells for 30 min and washed twice with DPBS before these cells were added to receptor-expressing cells to induce SynTrogo.

## Live cell imaging and analysis

Live-cell imaging was performed using a Nikon A1R confocal microscope (Nikon Instruments), mounted on a Nikon Eclipse Ti body and equipped with a CFI Plan Apochromat VC objective (×60/1.4-numerical aperture (NA)) and digital zoom Nikon imaging software (NIS Element AR 64-bit version 3.21; Laboratory Imaging), was used. During imaging, cells were maintained at 37 °C and 5% $CO_2$ using the microscope-mounted Chamlide TC System (Live Cell Instruments, Inc., Korea). Image analysis was conducted using NIS-Element AR microscope imaging software (NIS-element AR 64-bit version 3.21; Nikon). Time-lapse images of synthetic trogocytosis were analyzed in defined regions of interest (ROI) within the receptor cells.

## Flow cytometry

After culturing HeLa and HEK293T cells in a 6-well plate (SPL, Cat# 31006, Korea), igκ-GFP-TM-RFP (or igκ-LaM4-TM-RFP) and miRFP were co-transfected into HeLa cells, and igκ-EGFP-TM was transfected into HEK293T cells using Lipofectamine LTX (Invitrogen, Cat# 15338-100, CA, USA) according to the manufacturer's instructions. HEK293T cells expressing igκ-EGFP-TM were co-cultured with HeLa cells for 4 h. The cells were then trypsinized, resuspended in PBS, and collected into 5 ml Round-bottom tube with cell-strainer cap (FALCON, Cat# 352235, NY, USA). For dead cell staining, SYTOX™ Blue Dead Cell Stain (Invitrogen, Cat# 34857, CA, USA) was add at 1 μM for 15 min in the dark on ice. Ten thousand events were collected for each sample using flow cytometry (BD, LSRFortessa SORP, USA) and data were analyzed in FlowJo (FlowJo LLC version 10.7.1).

## Mice

All experiments were approved by the Institutional Animal Care and Use Committee (IACUC) of the Institute for Basic Science (IBS, IBS-2023-23). Male C57BL/6J mice (Jackson Laboratory, Stock No. 000664), 9–16-week old, were housed in groups and maintained in a controlled vivarium under a 12/12-h light/dark cycle (light on: 08:00 AM, light off: 08:00 PM), at 23–25 °C, and 50% humidity with ad libitum access to food and water. The mice were divided into experimental groups using random assignment.

## Animal Surgery for stereotactic viral injections

Surgical procedures were conducted in accordance with IBS IACUC guidelines. 9–16-week old mice were anesthetized with isoflurane (4% for induction, 1.5% for maintenance) during stereotaxic surgery. The skull was secured in a stereotaxic apparatus (RWD), the skin was shaved and incised, and a small craniotomy was performed for viral injection. The virus was bilaterally injected into the hippocampal CA3 (0.3 μl) and CA1 (0.3 μl) regions using the following stereotaxic coordinates: CA3: AP −2 mm, ML ± 2.75 mm, DV −2.5 mm and CA1: AP −2 mm, ML ± 1.4 mm, DV −1.4 mm. The following viruses were used: AAVDJ8-CaMKIIα-igκ-mEGFP-TM, AAVDJ8-CaMKIIα-smFP-FLAG, and AAVDJ8-CaMKIIα-synaptophysin-TagBFP for CA3, and AAV2/5-GfaABC1D-lck-mScarlet and AAV2/5-GfaABC1D-igκ-αGFP-TM-mScarlet for CA1. All viruses were at a titer of $2.0 \times 10^{12}$ genome copies per ml. Viral injections were performed using a pulled glass capillary with an end diameter of 30 μm, connected to a pressure microinjector (Picospritzer, Parker). The capillary was left in place for 5 min after injection to allow proper virus diffusion. Most experiments (Figs. 2–6) were conducted 3 weeks after virus injection, encompassing immunohistochemistry, electron microscopy, and electrophysiology. Long-term effects were evaluated at 7 weeks and 2 months post-injection through remote memory test, extinction, and NeuN density analysis.

## Immunostaining

Mice were anesthetized with a mixture of Alfaxan (40 mg/kg) and xylazine (10 mg/kg), followed by transcardial perfusion with ice-cold PBS and then with 4% paraformaldehyde (PFA) in PBS. After perfusion, brains were extracted and post-fixed in 4% PFA overnight at 4 °C. The brains were then sectioned using a vibratome (Leica VT1200) at a thickness of 30 μm.

For immunostaining, free-floating brain slices were washed in PBS, and then incubated in a blocking solution containing 5% normal goat serum in 0.3% Triton X-100/PBS (PBST) for 1 h. The primary antibodies were diluted in the same blocking solution overnight at 4 °C. The secondary antibodies were incubated with the same solution for 2 h at 4 °C. Slices were washed in PBS and mounted on slides with an antifade mounting medium (Vector Laboratories, H-1000, CA, USA). The following primary antibodies were used in this study: anti-GFP (Invitrogen, A10262, 1:1000), anti-RFP (Chromo, 5f8-150, 1:1000), anti-GFAP (EMD Millipore, AB5541, 1:1000), anti-GFAP (Abcam, AB7260, 1:1000), anti-S100β (Synaptic systems, 287 004, 287 003, 1:500), anti-Iba1 (Wako, 019 19741, 1:500), anti-VGluT1 (Synaptic systems, 135 302, 1:500), anti-PSD95 (Abcam, AB18258, 1:500), anti-VGAT (Synaptic systems, 131 003, 1:500), anti-Gephyrin (Abcam, Ab32206, 1:500). anti-NeuN (Synaptic systems, 266 004, 1:1000), anti-FLAG Tag (Invitrogen, 740001, 1:1000), anti-GluA1 (EMD Millipore, AB1504, 1:500), anti-Synaptopodin (Synaptic systems, 163 004, 1:500). All secondary antibodies were conjugated with Alexa dyes. (Invitrogen, 1:1000).

## Image analysis in brain slices

Confocal images were acquired using an LSM 900 (Zeiss) microscope with ×20/0.8 NA (air) and ×63/1.4 NA (oil) objectives, as well as a Nikon A1 microscope with ×20/0.75 (air), ×40/0.95 (air), and ×60/1.4 NA (oil) objectives. Three-dimensional reconstructions were generated from x63 and ×60 images using the "Surfaces" function in Imaris (version 10.0.0, Oxford Instruments) to quantify the volume of astrocyte and engulfed synaptic components. S100β staining was used to define the region of interest (ROI) for astrocytes. The percentage of GFP ligand volume engulfed relative to the total astrocyte volume (%v/v) was calculated as a measure of engulfment for each cell. To assess relative changes, %v/v values were normalized to those of the control group. To estimate the amount of synaptic components within astrocytic territories, the average intensity of synaptic markers (VGluT1, Bassoon, PSD95, Homer1, VGAT, and Gephyrin) within the S100β mask was measured.

To minimize individual variation in synaptic marker quantification within astrocytes due to viral expression (CaMKIIα-synaptophysin-TagBFP), one hemisphere was used as a control, while the other hemisphere received the SynTrogo virus. Unlike immunofluorescence, the amount of virally expressed synaptophysin was influenced by the surrounding viral expression, so we normalized the synaptophysin %v/v by the non-virally expressed astrocytes. To measure the relationship between engulfed synaptophysin and GFP puncta, an "overlapped volume ratio" function was used. A ratio greater than zero was considered a contact group, while a ratio of zero indicated a non-contact group. To quantify astrocyte and microglia reactivity, the fluorescent intensity and cell size were measured using FIJI. Z-series images were max projected into a single image to capture the full morphology of astrocytes and microglia. Images were preprocessed by enhancing contrast, despeckling, and creating masks for astrocytes or microglia based on their intensity.

To determine neuronal density, z-stack images of NeuN and GFAP co-immunostained brain sections were processed using Imaris software's "Surfaces" function. NeuN-positive neurons within the CA3 hippocampal region were identified through defined selection function, with individual neurons separated using the object separation function. CA1 GFAP-positive astrocytes were similarly analyzed using Imaris "Surfaces" function based on fluorescence intensity and

morphological criteria. Transduction efficiency was quantified by measuring GFP fluorescence intensity in neurons and RFP fluorescence intensity in astrocytes. Cells with fluorescence intensity exceeding established threshold values were classified as GFP+ neurons or RFP+ astrocytes, respectively.

To quantify the colocalization between GluA1 and synaptopodin, high-resolution confocal Z-stack images were contrast-enhanced and processed using a Difference of Gaussian (DoG) filter to emphasize puncta-like structures. Each fluorescence channel was independently binarized via Otsu algorithm-based auto-thresholding, converted into binary masks, and refined by removing small outliers. Colocalized signals were identified by applying the AND operation in the Image Calculator to the two binary masks, and overlapping puncta were classified as contact.

To determine synaptic density within the CA1 stratum radiatum (SR) layer and astrocyte territories, z-stack images were acquired with analysis restricted exclusively to the SR region. Following the protocol established for GluA1 and synaptopodin quantification, synaptic marker images underwent Difference of Gaussians (DoG) filtering and intensity-based thresholding before automated counting using FIJI's particle analyzer function.

For astrocyte territorial analysis, S100β immunostaining defined astrocytic boundaries. Astrocytes were identified by dual labeling (RFP +/S100β+), and circular ROIs with 30-μm radii were centered on astrocyte soma to demarcate territorial domains. Synaptic quantification was performed within these defined territories after excluding S100β-positive cellular areas.

To visualize receptor expression in axonal boutons and shaft regions, image analysis was performed on brain slices co-expressing CaMKIIα-igk-mEGFP-TM (transmembrane ligand) and CaMKIIα-smFP-FLAG (cytosolic marker). Line profile analysis was performed by manually tracing individual axons with 10-pixel-wide ROIs, enabling quantitative assessment of protein expression patterns across axonal shafts and synaptic boutons.

## Behavioral analyses

**Contextual fear conditioning.** Contextual fear conditioning (CFC) was conducted in a fear conditioning chamber (Coulbourn Instruments). The procedure consisted of a 3-min habituation period followed by a 7-min conditioning phase, during which mice received five tone-shock pairings. Each tone lasted 30 s, followed by a 2-s foot shock at either 0.3 mA or 0.5 mA and the tone and shock were coterminated. After the conditioning phase, mice remained in the chamber for an additional 60 s before being returned to their home cages. For contextual memory recall, mice were reintroduced to the chamber for 5 min one day and 21 days after the conditioning phase. Behavior was recorded and analyzed using FreezeFrame software (version 3; Coulbourn Instruments), with motionless bouts lasting over 1 s considered as freezing.

**Spaced extinction.** Mice were reintroduced to the conditioning chamber for 3 min twice, with a 2-h interval between sessions, without the footshock or sound[38]. This protocol was repeated for 4 days, and freezing behavior was averaged across the two sessions for each day. Two weeks later, spontaneous recovery (SR) of the extinguished memory was assessed by re-exposing mice to the chamber for 5 min and measuring freezing behavior.

**Open field test.** Mice were placed in a 40 × 40 cm white plastic box and allowed to explore freely for 10 min. Their behavior was recorded and analyzed using EthoVision XT (Noldus). A 20 × 20 cm area in the center of the box was defined as the central zone. The total distance traveled, time spent in the central zone, and average velocity were analyzed to evaluate the mice's basal locomotor activity and anxiety levels.

**Y-maze test.** Mice were placed in a Y-shaped gray plastic apparatus for 5 min. The apparatus had three arms, each 34 cm long, 16 cm high, and 4 cm wide, arranged at 120° angles. Behavior was recorded and analyzed using EthoVision XT (Noldus). The total number of entry and spontaneous alteration were analyzed to evaluate the mice's working memory.

## Electrophysiology

Three weeks after virus injection, mice were deeply anesthetized with 1% isoflurane followed by decapitation. Then, the brain was excised from the skull soon as possible, and submerged in the chilled cutting solution: 250 mM sucrose, 26 mM $NaHCO_3$, 10 mM D(+)-glucose, 4 mM $MgCl_2$, 0.1 mM $CaCl_2$, 2.5 mM KCl, 2 mM sodium pyruvate, 1.25 mM $NaH_2PO_4$, 0.5 mM ascorbic acid, and 1 mM kynurenic acid, pH 7.4. The whole solution was gassed with 95% $O_2$ and 5% $CO_2$. After trimming the brain, coronal slices were cut 300-μm-thick in thickness with a vibrating microtome (LinearSlicer PRO7N; Dosaka EM Co. Ltd, Japan) and transferred in a chamber filled with oxygenated extracellular artificial cerebrospinal fluid (aCSF) solution: 126 mM NaCl, 24 mM NaHCO3, 1.25 mM NaH2PO4, 3.5 mM KCl, 1.5 mM MgCl2, 1.5 mM CaCl2, and 10 mM D(+)-glucose, pH 7.4 at 20–22 °C for at least 1 h. For electrophysiology recording, the slices were transferred to a recording chamber which was mounted on the stage of an upright microscope (Examiner D1; Zeiss, Germany) and visualized with a 60× water immersion objective with infrared differential interference optics. Cellular morphology and fluorescence were visualized by a CMOS camera (ORCA-Flash 4.0; Hamamatsu, Japan) and Imaging Workbench software (INDEC BioSystems). The whole-cell patch-clamp recording was performed within pyramidal neurons in the hippocampus CA1. The patch pipette (Warner Instruments, MA, USA) resistance was 4–7 MΩ was pulled using a micropipette puller (PC-100; Narishige, Japan).

Pipette internal solution for spontaneous- and evoked EPSCs recording contained: 132 mM CsMeSO4, 8 mM NaCl, 0.5 mM $CaCl_2$, 10 mM HEPES, 0.25 mM EGTA, 2 mM Mg-ATP, 0.5 mM $Na_2$-GTP, 10 mM QX-314, and pH 7.3 adjusted with CsOH, 290–295 mOsmol/kg. Both spontaneous EPSCs and IPSCs were recorded at holding potentials of −70 and 0 mV, respectively. The currents were measured after 3–5 min from recordings. Miniature EPSCs were measured during the bath application of 1 μM tetrodotoxin (T-550; Alomone Labs). To evoke EPSCs in the CA1 pyramidal neurons, synaptic responses in pyramidal neurons were evoked by 0.1 Hz stimulation of Schaffer collateral (100 μs duration; 100–700 μA intensity) via a constant current isolation unit. Schaffer collateral was stimulated by a tungsten bipolar electrode in the striatum radiatum layer in the lateral CA1 area.

To measure paired-pulse ratio (PPR) with varying interstimulus intervals, Schaffer collateral fibers in the lateral CA1 area were stimulated by paired electrical stimuli (20, 50, 100, 200, and 500 ms interstimulus interval). Altering the interstimulus enabled the calculation of the PPR as the ratio of the second pulse amplitude (P2) to the first pulse amplitude (P1). The intensity of the paired electrical stimulation was carefully adjusted to 100 pA of P1. The PPR (P2/P1) demonstrated the assessment of synaptic efficacy and plasticity by observing changes in the PPR across different intervals.

For the measurement of the size readily releasable pool (RRP), we followed the protocol as previously described[69]. Pipette internal solution contained: 132 mM CsMeSO4, 8 mM NaCl, 0.5 mM $CaCl_2$, 10 mM HEPES, 0.25 mM EGTA, 2 mM Mg-ATP, 0.5 mM $Na_2$-GTP, 10 mM QX-314, and pH 7.3 adjusted with CsOH, 290–295 mOsmol/kg. Schaffer collateral fibers were stimulated by repetitive electrical stimuli (20 Hz, 100 stimulation). The cumulative EPSCs (y-axis) were plotted against the number of stimuli (x-axis). To estimate the RRP$_{train}$, a linear line was fitted from the 81–100 points (last 20 points) of the cumulative EPSCs and back-extrapolation to the y-axis which estimates the RRP size.

Pipette internal solution for the long-term potentiation (LTP) recording contained: 135 mM CsMeSO4, 8 mM NaCl, 0.5 mM CaCl₂, 10 mM HEPES, 0.25 mM EGTA, 1 mM Mg-ATP, 0.25 mM Na₂-GTP, 20 mM QX-314, and pH 7.3 adjusted with CsOH, 290–295 mOsmol kg⁻¹. After a stable 5-min baseline, theta-burst stimulation (TBS) consisting of four trains of 10 bursts was applied as previously described[70]. Within each burst, there were five pulses at 100 Hz within 200 ms intervals and 5-s gaps were between trains. The TBS protocol was applied while holding the current at 0 (I = 0 mode) to ensure consistent conditions during the application of the train stimulation. Stimulation intensity was adjusted to 40% of maximal evoked EPSCs. Evoked EPSCs were collected every 15 s with a holding potential of −70 mV for 70 min. All evoked EPSCs were normalized to the average of the baseline.

To measure the AMPA/NMDA ratio, GABA receptor antagonists (50 μM bicuculline and 5 μM CGP 25348) were present during the recording. For the AMPA current recording, the pyramidal neuron was held at a membrane potential of −70 mV. For the NMDA current, the membrane potential was held at a membrane potential of +40 mV. The evoked stimulation intensity was adjusted to 70% of the maximum NMDA current. NMDA peak amplitude was selected at 65 ms after the peak of the response to prevent contamination.

For the measurement of intrinsic properties of CA1 pyramidal neurons, the pipette for action potential recording contained: 145 mM K-gluconate, 10 mM HEPES, 5 mM NaCl, 0.2 mM EGTA, 5 mM Mg-ATP, 0.5 mM Na₂-GTP, and pH 7.3 adjusted with KOH, 290–295 mOsmol/kg. Measurement was performed in a whole-cell current-clamp mode with no membrane potential adjustment. Once stable recordings were achieved, the intrinsic properties, including membrane potential, action potential threshold, the amplitude of peak, input resistance, and amplitude afterhyperpolarization, were measured. Currents were injected from −220 pA to 140 pA for every 20 pA. All electrical data were digitized and collected with a Digidata 1550B and the Axon Multiclamp 700B patch-clamp amplifier (Molecular Devices, CA, USA) using the pClamp software (ver.11.0.3).

## Correlative light and electron microscopy (CLEM) imaging

**Acquisition and volume imaging.** Structural interactions between astrocytes and axons from CA3 pyramidal neurons were confirmed using a CLEM technique that combines the advantages of confocal and electron microscopy (EM). For three-dimensional visualization of fine morphological features of axons and astrocytes, we used ATUM-SEM (Automated tape-collecting ultramicrotome combined with scanning electron microscopy). For sample preparation, mice injected with AAV virus were perfused with 2% paraformaldehyde and 2.5% glutaraldehyde in 0.1 M cacodylate buffer solution (pH 7.4) three weeks post-injection. Fixed brains were sliced into 150 μm-thick sections using a vibratome (VT1000S, Leica) and stained with 4',6-diamidino-2-phenylindole (DAPI) to visualize nuclei and blood vessels. In the CA1 region of a coronal brain slice, ROIs containing astrocytes with both RFP and GFP fluorescence signals were imaged using the NIKON A1R confocal microscope mounted onto a Nikon Eclipse Ti-E body. Bright field images served as references to aid in the precise dissection of imaged regions for EM imaging. Dissected tissues were then processed for heavy metal staining, dehydration, and resin embedding for scanning electron microscopy (SEM), following the previously reported protocol[71]. Samples were manually trimmed and serially sectioned into 50 nm-thick thin sections with an ultra-Maxi knife (DiATOME, Biel, Switzerland). Serial sections were collected on plasma-hydrophilized carbon nanotube (CNT)-coated PET tapes (Boeckeler Instruments) using an ATUMtome (Boeckeler Instruments, Inc., Tucson, USA). Sections on the CNT tapes were mounted on aluminum wafers with double-sided adhesive conductive tape (Ted Pella) and then imaged using a SEM (Sigma, Carl-Zeiss Microscopy GmbH, Oberkochen, Germany) equipped with In-lens secondary electron (SE) or backscattered electron detector (BSD). Atlas 5 software (Fibics Incorporated, Ottawa,

Canada) was used for large area imaging. By correlating low-resolution images (80 nm pixel size) with the ROI, we confirmed the position relative to marked structures, such as blood vessels and nuclei. This information enabled precise correlation between imaging modalities for high-resolution ROI imaging. After identifying the target astrocyte, serial SEM imaging was performed with a beam voltage of 5 kV, using BSD detection with a dwell time of 7 μs. EM images were captured at a resolution of 5 nm by 5 nm per pixel in the X and Y directions, stitched into a single plane of tiled images. Serial mosaic images were aligned using ImageJ and Fiji plugins with the TrakEM2 software (http://fiji.sc/wiki/index.php/Fiji).

**3D reconstruction and structural segmentation.** For each experimental group (control and SynTrogo), we analyzed three brain samples, obtaining one EM image stack per brain (3 stacks per group in total). Image stacks covered approximately 65 × 75 μm in the XY plane with 70 serial sections at 50 nm thickness (total depth: 3.5 μm) in the control group, and approximately 80 × 91 μm in the XY plane with 80 serial sections at 50 nm thickness (total depth: 4 μm) in the SynTrogo group.

Following the identification of fluorescent astrocytes, we performed 3D reconstruction of GFP-positive CA3 axons in contact with the labeled astrocyte, along with the dendrites of CA1 neurons forming synapses with these CA3 axons. Reconstructions were conducted using software available from SynapseWeb (version 1.1.0.0, https://synapseweb.clm.utexas.edu/software-0). Structural segmentation included mitochondria and synaptic vesicles within CA3 axons, as well as spine apparatuses within dendritic spines. The astrocyte-axon interface was also manually segmented to visualize the contact area. To quantify bouton versus shaft engagement, we measured the proportion of the axonal surface directly apposed to astrocytic processes, defined as the axon–astrocyte interface (AAI; yellow contour in Supplementary Movies 6 and 7). The distances between the apposed membranes of axons and astrocytes in this analysis were all within 100 nm. For each reconstructed axonal segment, the AAI ratio was calculated as:

$$AAI\,ratio = (Surface\,area\,of\,axon - astrocyte\,contact)/(Total\,axonal\,surface\,area)$$

Boutons were identified as enlarged varicosities containing dense clusters of synaptic vesicles and forming synaptic contacts with postsynaptic structures. Shafts were defined as the thinner axonal processes interconnecting boutons, which may contain sparse vesicles but lack synaptic specializations. Serial profiles of axon-astrocyte contacts were manually segmented, and the corresponding contact regions on axons were annotated. Reconstructed 3D objects were then exported to 3ds Max 2020 software for quantitative measurements of surface areas. To enhance the visualization of these complex structures, segmented models were imported into Autodesk 3ds Max for advanced 3D rendering.

Membrane thickness was quantified from serial EM sections at the astrocyte-axon contact interface. At each contact site, the electron-dense line of the GFP-positive axonal membrane was measured at three distinct points and averaged to yield a mean thickness per axon. For quantification of bouton volume, vesicular and mitochondrial contents, and synapse density, axons from all groups were selected based on direct contact with astrocytes (≤100 nm), in order to minimize potential bias.

## Statistics and reproducibility

All experiments were performed with at least three biological replicates, and the results are shown as median with upper and lower quartiles (25th and 75th percentiles) or mean ± s.e.m., unless stated otherwise. Excel (Microsoft) and GraphPad Prism 10.0 (GraphPad Software Inc.) were used for plotting, data fitting, graphing and

statistical analysis. All micrographs shown in this study are representative images of experiments repeated at least three times. All statistical tests were conducted as two-sided unless otherwise specified.

### Reporting summary

Further information on research design is available in the Nature Portfolio Reporting Summary linked to this article.

## Data availability

The data supporting the findings of this study are provided in the paper and its supplementary information files. Raw data are available from the corresponding author upon request. Source data are provided with this paper.

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

## Acknowledgements

We thank all members of the Biomolecular Design Lab and the Center for Memory and Glioscience, IBS, for their valuable discussions and feedback on this research. We thank Dr. Doyun Lee (IBS) for valuable insights during the conceptual development of this work, and Dr. Jongryul Hong (KAIST) for helpful comments on the design of ligands and receptors. We are grateful to the IBS Research Solution Center (RSC) for supporting our imaging experiments and analysis using confocal and fluorescence microscopes, and to the IBS virus facility for providing a virus packaging service for in vivo experiments. Electron microscopy data were obtained at the Brain Research Core Facilities at KBRI. This work was supported by the Institute for Basic Science (IBS), Center for Memory and Glioscience (IBS-R001-Y4 to S.L., and IBS-R001-D2 to C.J.L. and S.L.), the KBRI basic research program funded by Ministry of Science and ICT (25-BR-01-03 to K.J.L.).

## Author contributions

S.L. conceived the initial idea of engineering neuron-astrocyte interactions. The research plan and interpretation of the results were developed through discussions among S.H.K., W.W., G.H.K., Y.J., C.J.L., K.J.L., and S.L. S.H.K., W.W., G.H.K., Y.H.K., S.S., S.C., D.Y.K., M.G.P., Y.-J.C., S.S.W., J.S., and S.L. performed the experiments and analyzed the data. S.H.K., W.W., G.H.K., C.J.L., K.J.L., and S.L. wrote the manuscript.

## Competing interests

The authors declare no competing interests.
