## [Transparent Peer Review File · Nature Communications]

Remodeling Synaptic Connections via Engineered Neuron-Astrocyte Interactions

Corresponding Author: Dr Sangkyu Lee

Version 0:

Reviewer comments:

Reviewer #1

(Remarks to the Author)

Summary:

"Remodeling Synaptic Connections via Engineered Neuron-Astrocyte Interactions" by Kim, Won, Kim et al. introduces SynTrogo, a synthetic molecular approach to manipulating neuron-astrocyte interactions. By engineering neurons with an "eat-me" signal and astrocytes with a complementary receptor, the study leverages astrocytic trogocytosis to remove presynaptic components selectively. The authors conducted extraordinary, high-quality work to validate SynTrogo in vitro. Using HEK293T, HeLa cells, and primary astrocyte-neuron co-cultures, they employed advanced live-cell microscopy and flow cytometry to rigorously demonstrate a trogocytosis-like process, providing a compelling mechanistic foundation for their findings. They then extend their findings in vivo, focusing on the hippocampus and showing through light microscopy, electrophysiology, correlative light-electron microscopy, and fear conditioning paradigms—that SynTrogo induces targeted synapse elimination, leading to compensatory synaptic strengthening, enhanced long-term potentiation (LTP), and improved fear memory stability. This manuscript effectively contextualizes its findings within the existing literature. While the approach is novel and elegant, our enthusiasm is tempered by the absence of key experiments and some overreaching claims that require clarification. Before acceptance and publication, addressing the following concerns is needed to ensure the accuracy of the main claims of the study:

Major Comments:

- The authors claim throughout the manuscript that SynTrogo induces a synaptic "eat-me" signal, but experimental data do not support this assertion. Their evidence is limited to showing that SynTrogo-positive astrocytes exhibit increased internalization of presynaptic GFP and a slight elevation of internalized VGLUT1 (Figure 2.G). However, this does not conclusively demonstrate that SynTrogo functions as an "eat-me" signal. Rather, it suggests that a small amount of presynaptic VGLUT1 may be incidentally internalized alongside the GFP-tagged receptor rather than actively targeted for removal. Additionally, the observed synaptic effects could stem from excessive adhesion between astrocytes and neurons due to synthetic interactions between the GFP nanobody and the GFP receptor rather than active synaptic remodeling. Conducting an adhesion strength analysis would significantly strengthen the authors' claims by distinguishing between adhesion-driven and trogocytosis-mediated effects, namely the notion that the synaptic abnormalities are due to the intake of the GFP-positive membrane.
- Throughout the paper, the authors use Correlative light-electron microscopy (CLEM) to illustrate changes in synaptic architecture in SynTrogo-positive astrocyte-neuron interactions. However, the criteria for selecting specific images for analysis are unclear, and light microscopy reference images can be misleading. To strengthen the manuscript, the authors should provide CLEM or Immuno-Gold EM evidence showing a SynTrogo-positive astrocyte containing VGLUT1+GFP+ presynaptic vesicles.
- In the manuscript, the authors demonstrate that SynTrogo induces synapse elimination using electron microscopy to quantify the number of synapses lost. It is unclear how the authors selected the axons and synapses quantified in Figure 4A-C. In addition, it would be interesting to determine which types of synapses are lost due to SynTrogo. Using fluorescence analysis of inhibitory and excitatory synaptic markers to quantify the number of excitatory versus inhibitory synapses within the territory of SynTrogo-positive astrocytes.
- The authors claim that SynTrogo specifically targets excitatory synapses in the CA1 region Figure 4. However, to

rigorously confirm this selectivity, they should target CA1 as the presynaptic site rather than CA3, since CA1 contains most presynaptic neurons forming local inhibitory synapses. This approach would provide clearer evidence that SynTrogo preferentially affects excitatory, rather than inhibitory, synapses.

- If SynTrogo is truly specific to excitatory synapses, what drives this selectivity? The authors report increased release probability, as evidenced by more docked presynaptic vesicles. However, whether this is merely a compensatory response to synapse elimination remains unclear, or if a distinct molecular mechanism underlies this effect. Clarifying this point would significantly strengthen the study's conclusions.

Minor Comments

Sample Size Reporting: The manuscript lacks clarity in sample size reporting. Mouse N values should be provided for all experiments to ensure data are not biased by specific animals.

Figure 1I-J: Consider adding 3D reconstructions of astrocytes with Homer and Bassoon puncta, similar to Figure 2D.

Figure 2A: The experiment timeline should be clearly labeled, and the control group should be more distinctly marked.

Figure 2H: The relevance of this figure to the authors' claims is unclear. Further clarification is needed.

Figure 2J: Ensure consistency in sample sizes for sEPSC frequency (n=20) and amplitude (n=19) to avoid discrepancies.

Figure 3E: The method for identifying the green mask as a presynaptic component is unclear. Further validation or explanation is needed.

Figure 4E: The control group for mEPSC (n=6 cells) is too small to draw strong conclusions. Additional N is recommended.

Figure 5G: The slower eEPSC decay kinetics in SynTrogo-treated cells could result from high release probability leading to AMPAR saturation. Analyzing AMPAR decay kinetics would further support conclusions on increased release probability.

Figure 5E-F: Sample sizes should be consistent between Figures 5E and 5F, which appear to be derived from the same experiments. The absence of two control cells in Figure 5F should be addressed.

Figure 6K-L / Extended Figure 14J: There is a discrepancy between AMPAR-mediated EPSCs and the SynTrogo condition. In Figure 6L, AMPA currents decrease in SynTrogo (w/o FC), yet eEPSC amplitude increases in SynTrogo (Extended Figure 14J). Since eEPSCs are AMPAR-dependent, these trends should be consistent. The authors should clarify this inconsistency.

Figure 6A-B / Extended Figure 14J: Representative traces should be provided for LTP data and eEPSC input-output curves. Broader Applicability: To enhance the impact of this strategy, it is essential to determine whether SynTrogo functions in brain regions beyond the hippocampus. Testing its efficacy in other circuits would broaden its potential applications.

Therapeutic Relevance: The authors should clarify how their findings support potential therapeutic applications for Alzheimer's disease or remove this statement from the manuscript.

Reviewer #2

(Remarks to the Author)

The authors devised a new strategy to artificially enhance structural remodeling of synapses by boosting the phagocytic capabilities of astrocytes. A simple approach of introducing an 'eat-me' signal in neurons and the complementary receptor in astrocytes led to enhanced synapse elimination, as well as improved synaptic plasticity and memory. The study was conducted meticulously, and the findings are both novel and exciting. I have just a few fundamental questions for the authors.

(1)

The SynTrogo system created by the authors does not use a traditional or endogenous phagocytosis signal. Therefore, it may not be ideal to introduce it as the 'eat-me' signal and its complementary receptor. The presynaptically expressed ligand is merely an extracellularly facing, membrane-targeted GFP, and it is not an 'eat-me' signal—it simply happened to function as one. The postsynaptically expressed receptor is an antibody (nanobody) for GFP, rather than a phagocytosis receptor. If you had modified and enhanced an endogenous 'eat-me' signal and its receptor, then this terminology would be appropriate. I would suggest using a different term when introducing this system in the abstract. Afterwards, explaining that this system essentially functions as an 'eat-me' signal and its receptor is perfectly fine.

(2)

With the enhanced 'random' synapse elimination capabilities, wouldn't you expect the neuronal circuit to become disorganized, potentially leading to less optimal circuitry? I understand that synapse 'elimination' is just as important as new synapse 'formation' in reorganizing information flow and encoding memory. However, with this artificial enhancement of phagocytosis, important and less important synapses may not be properly distinguished.

I understand that the behavioral experiments suggest an enhanced memory effect. However, while the enhanced phagocytic capabilities may appear adaptive in the specific behavioral tests you conducted, they may not be in other contexts. If you have performed additional memory tests that yielded less favorable outcomes, it might be wise to include these data, at least as Supplementary Data. If such experiments were not conducted, discussing the possibility that enhanced phagocytic capability may not be optimal in all situations would be beneficial.

(3)
I was not aware of the GFP nanobody (aGFP). A brief explanation of this material upon its introduction would benefit a broad readership.

(4)
It appears that phagocytosed materials eventually end up in the lysosome. Given the acidic intra-lysosomal environment, one would expect GFP fluorescence to be quenched and rapidly degraded. However, the fluorescence seems to persist in the authors' experiments. Have the authors tracked the vesicles up to their fusion with the lysosome using live imaging? Would GFP fluorescence decrease substantially upon lysosomal uptake?

If GFP fluorescence is lost in the lysosome, the overall estimation of cellular uptake by phagocytosis may underestimate the actual amount internalized. Please discuss the possibility that GFP may be quenched in the lysosome and how the loss of fluorescence could affect the analysis somewhere in the text.

(5)
Figure 1f, right panel: The label "(Astrocyte)" does not need to start on a new line.

(6)
It appears that neuronal igk-GFP-TM is taken up only by astrocytes expressing the aGFP nanobody (Figure 1f, g). Does this imply that there is almost no phagocytic activity in naïve astrocytes? Are the trogocytotic events observed by the authors purely artificially induced, requiring both the artificial ligand and the artificial receptor in adjacent cell pairs?

If this is the case, then these trogocytotic events are not an enhancement of an existing endogenous process but rather a completely artificial one. I assume the authors would not want to make this assumption, so it may be worth presenting some evidence that these phagocytic events occur, albeit to a lesser extent, under naïve conditions. If the authors believe that these trogocytotic events are entirely artificial, then that is fine. However, in that case, it may be more appropriate to avoid describing the effects of SynTrogo as an "enhancement" of a natural phagocytic process.

(7)
The EM studies presented in Extended Data Fig. 11 are particularly enlightening. In Morizawa et al. (2022), the degradation process of internal neuronal compartments was apparent even during the transition from stage 3 (Pinching) to stage 4 (Closure) in the cerebellum. Similarly, I notice some deformation of the internal compartments in panel d (Degradation). The authors could specifically highlight this observation in the manuscript and perhaps discuss the possibility of apparent deformation occurring even during stages 3 and 4, if that is indeed what they observe.

(8)
The authors observed an increase in sEPSC frequency and a decrease in mEPSC frequency in Trogo preparations. At first glance, this suggests an increase in presynaptic spontaneous action potential firing in acute slice preparations from the Trogo mice. The authors could mention this possibility early in the Results section (page 7, line 189-). The intrinsic properties of the postsynaptic CA1 pyramidal neurons were evaluated (Extended Data Fig. 14), and no differences were found. Similarly, the authors could have evaluated the presynaptic CA3 pyramidal neurons or assessed spontaneous firing frequency through unit extracellular recording. If these experiments were not done, that is fine; however, discussing the possibility of an increase in presynaptic spontaneous action potential firing could be valuable.

If the spontaneous action potential firing frequency is assumed to be approximately the same between the two preparations, then the enhancement of release probability in response to presynaptic spontaneous action potential firing becomes more likely. It may also be important to point out that mEPSC amplitude (Fig. 4f) and sEPSC amplitude (Fig. 2j) are similar, suggesting the absence of multisynaptic contact between an axon and a postsynaptic dendrite, as well as the lack of multivesicular release at a single synaptic contact. While these points may seem trivial, they would support the authors' claim that presynaptic release probability is enhanced in the remaining synapses in SynTrogo preparations.

(9)
The authors found an increased PSD area (Extended Data Fig. 13). In many previous studies, an increase in PSD area usually reflects an increase in the number of postsynaptic AMPA receptors that the PSD accommodates. With an increase in the number of AMPA receptors, one would typically expect an increase in the amplitude of mEPSC and sEPSC. However, such an increase in amplitude was not observed in the authors' data. This discrepancy is acceptable, as it reflects the actual recordings.

However, it may be worthwhile to point out this discrepancy between ultrastructural and electrophysiological data. The noise in the electrophysiological data, which may prevent the smallest events from being captured, could explain this discrepancy. If the average amplitude of mEPSC or sEPSC events actually increases, the smallest events would become large enough to be detectable. This would give the appearance that mEPSC or sEPSC frequency had increased without an increase in their

amplitude. There is no doubt that presynaptic release probability has increased, as shown in the paired pulse experiment in Fig. 5g and 5h. Collectively, the authors' data support the idea that both presynaptic and postsynaptic properties are enhanced in the pruning-resistant synapses.

Logical arguments such as the one above could be introduced somewhere in the text.

(10)

Not only showing the enhancement of fear-conditioning memories but also demonstrating the enhanced capability for extinction was a great experiment conducted by the authors. This shows that the remaining neuronal circuit, resistant to artificially enhanced phagocytic activity, has higher adaptability.

However, this raises the question of what these excess synapses were originally for. If the system can function adaptively well without these synapses, why were they prepared in the first place? It would be worthwhile to discuss the adaptive advantage of having excess, non-essential synapses in naïve animals.

(11)

It would also be important to discuss the number of CA3 pyramidal cells that express igk-GFP-TM. If all CA3 cells express this protein and all synapses are eliminated, then there would no longer be a functional hippocampal circuit. The authors could provide an estimate of the relative ratio of infected cells along with the relative ratio of total endogenous synapses that have been eliminated. The outcome of the behavioral experiments may depend crucially on this ratio. It may well be that the surviving synapse ratio just happened to be optimal for encoding and extinction of fear memories. Please discuss the possibility of the serendipity of the virus infection ratio that resulted in a 'favorable' outcome.

(12)

I assume that not all synapses that presynaptically express igk-GFP-TM are eliminated before fear conditioning. I also assume that endogenous phagocytosis may be enhanced in control animals after fear conditioning. It may be worthwhile to observe whether there is an increase in the phagocytosis of igk-GFP-TM after fear conditioning in control animals, where there is no expression of igk-alphaGFP-TM in CA1 astrocytes, and in those with its expression.

I do not believe the authors have conducted such experiments. Therefore, it would be valuable to discuss the possibility of using this system to evaluate (1) whether there is endogenous phagocytic activity during memory paradigms, and (2) whether the enhancement of the elimination of remaining synapses in SynTrogo animals can also be observed.

(13)

In the Discussion (page 9, line 250), the authors write "selective elimination of synapses." I do not recall whether the authors demonstrated that trogocytosis occurs selectively for eliminating synaptic components. My impression was that both the ligand and receptor are expressed broadly on neurons and astrocytes, respectively, and that phagocytosis could occur anywhere on the cell. The use of the term "selective" may be misleading.

Additionally, the term "selective" may also mislead readers into assuming that the SynTrogo system was targeted to selectively eliminate memory-unrelated synapses. This was not the case. The igk-GFP-TM was expressed randomly on CA3 pyramidal cells using the CamKII promoter with AAV. The expression was not driven, for example, by an activity-dependent promoter such as *cfos*. Since there was no selective expression, the term "selective" may be confusing.

(14)

I think I may have understood why the memory performance was enhanced in animals where a portion of random synapses was eliminated by artificial trogocytosis. The AAV injection, three weeks prior, would have significantly reduced the number of synapses by the time of the behavioral experiments. This is an unusual situation, which could be similar to a pathological condition such as focal stroke in some respects. The neuronal circuit would attempt to adapt to this situation through homeostatic synaptic scaling, and the overall meta-plastic state may be enhanced to adapt to this emergency. It is known that neuronal circuit plasticity increases sometime after a stroke, and rehabilitation during this window of opportunity can result in favorable outcomes or reorganization of the remaining neuronal circuit, leading to functional recovery. The behavioral experiments were conducted specifically during this period of enhanced plasticity. Perhaps a discussion like this, comparing it to focal stroke pathology, may provide additional insights.

Reviewer #3

(Remarks to the Author)

In this study, Kim et al engineer a method to target specific cells for partial digestion by another cell, by displaying a GFP domain on the target cell surface and a anti-GFP nanobody with a RFP intracellular domain on the other cell surface. They show this works with HEK293 and HeLa digester cells, name the system synthetic trogocytosis (SynTrogo), then apply it to CA3 neuronal targets and CA1 astrocyte digesters in vivo. They perform detailed EM and ephys studies that show ingestion of presynaptic components by the CA1 astrocytes and reduced presynaptic bouton density with enlargement of the

remaining boutons in axons that express the SynTrogo ligand, compared to axons in uninjected control mice (if I interpret the controls correctly). They also find structural changes in postsynaptic structures at baseline compared to control. Interestingly, they observe enhanced LTP after theta-burst stimulation in CA3-CA1 connections and faster learning in mice subjected to CA3 axon - CA1 astrocytic SynTrogo.

Synthetic trogocytosis is a conceptually novel method for controlling cell structure and function and is likely to have various applications in development, tissue engineering, and neurobiology. Thus I am very enthusiastic about this study as a candidate for publication in Nature Communications. There are just some clarifications needed that I expect the authors can address in a revision.

Important information necessary for reproducing the results, understanding the strength of the effects, and understanding the limitations of the method are missing, and should be provided as follows:

1. In Figures 2-6 and related extended figures, the time of analysis relative to injection and the age of the mice should be stated.
2. The percent of neurons that express SynTrogo ligand and the percent of astrocytes that express SynTrogo receptor in the ephys and behavioral experiments (currently Figs 4-6) should be quantified and presented. Since presumably < 100% of CA3 neurons express SynTrogo ligand and <100% of CA1 astrocytes express SynTrogo receptor, this information is necessary to interpret the significance of the observed changes in synaptic function measured in the CA1 neurons. The numbers should be discernable from the EM reconstructions of Figs 3-5 if these were performed under the same conditions, but this would require volumetric reconstruction of axons and astrocytes by EM which may be difficult. Alternatively, the investigators can redo the experiment of Fig 2 under the conditions used for Figures 5 and 6 with a NeuN stain in CA3 and a GFAP stain in CA1 so they can estimate transduction efficiency in CA3 neurons and CA1 astrocytes.
3. Long-term effects of the system are unclear. The experiment to investigate effects on cell viability is too short, with just 4h of cocubation of SynTrogo ligand and receptor cells (Ext Data Fig 4). This should be extended to 1 and 3 days.
4. Likewise, will the targeted axons eventually get completely removed? If so, will the CA3 neurons lacking axons die? Thus investigating the effects of SynTrogo on the CA3 neuron morphology and survival at one or more timepoints at least 1 month after the latest timepoint shown here (which I assume to be 3 weeks as that was mentioned for the ephys) would be useful. I might suggest 2 and 3 months after viral transduction.

In addition, there are a few revisions needed to make the conclusions more understandable or more robust.

5. It appears the EM observations of Trogo CA3 axonal and CA1 dendritic structural changes are comparing Trogo groups to non-transduced groups, based on the mention of "groups" in the methods. However this needs to be confirmed and mentioned explicitly in each figure and in the text. Right now there is a consistent lack of mention of what the observed changes are being compared to, for example, "SynTrogo significantly reduced the number of ligand-labeled axonal boutons and their coupled dendritic spines", "mitochondria volume of presynaptic boutons increased", "the spine volume and PSD area significantly increased" etc. That is, it is important to confirm if the comparisons are always between hippocampi with our without the SynTrogo treatment (ideally same mouse, unilaterally injected and using the contralateral hippocampus as control), or between axons/dendrites in apposition to a SynTrogo receptor-expressing astrocyte vs axons/dendrites not in apparent contact.
6. Related to this, in Fig 5, besides presenting the changes in bouton number and size in GFP (ligand)-expressing axons relative to uninjected hippocampi, the authors should also determine whether there are any differences in bouton number and size in non-expressing axons relative to uninjected hippocampi. Given the changes in EPSC frequency observed in postsynaptic non-genetically altered CA1 neurons, one wonders if even non-genetically altered CA3 axons in injected hippocampi might experience some feedback regulation that alters their presynaptic structure. These comparisons should be achievable using the existing EM dataset.
7. I would suggest a reorganization of Figures 4-6 as currently the way they present the data make the conclusions difficult to evaluate. The main problem is I do not think one can robustly relate EM data to ephys data, or ephys data to behavior, without detailed manipulations that prove the connection, but each of these figures presents a correlation to imply the functional relationship without proving it. Meanwhile data that would be easier to understand together are divided between different figures. Specifically, there is EM data in Figs 4-5 and ephys data in Figs 4-6. In addition Fig 4 as currently structured does not stand well as a main figure on its own, as it does not generate a robust conclusion, while it analyzes the same dataset as the EM panels in Fig 5. The authors can consider grouping the EM panels from Figs 4-5 together into Fig 4 only, and grouping ephys data from Figs 4-6 into Fig 5 only. The text of the results section would then also have to be reorganized to present EM, ephys, and behavioral findings in that order.
8. "We found a decrease in the proportion of SER(-) spines, but no change in the SER(+) spine percentage" sounds like an internally illogical sentence. To fix this, it should be rewritten as "We found a decrease in the proportion of SA(-) SER(-) spines, but no change in the SA(-) SER(+) spine percentage, due to an increase in the percentage of SA(+) spines"

Reviewer #4

(Remarks to the Author)

In this manuscript, Kim and co-authors engineered a system to induce trogocytosis and study its downstream effects. To achieve this, a selected cell bearing anti-GFP (receptor) recognizes surface-GFP (ligand) expressed by a target cell, leading to their interaction and the engulfment of small particles from the ligand-expressing cell by the receptor-expressing cell. The authors first extensively characterized this system *in vitro*, demonstrating that it requires actin polymerization and that the ingested material consists primarily of membrane. This tool was subsequently applied *in vivo* to investigate the functional consequences of axonal trogocytosis. Using viral injections, the authors ectopically expressed the receptor in CA1 astrocytes and the ligand in CA3 neurons in the hippocampus, inducing synthetic interactions and axonal nibbling by astrocytes. Staining and colocalization analyses suggested that synaptic material was engulfed. Correlative light and electron microscopy revealed that axons in contact with astrocytes contained fewer synapses, and electrophysiology data indicated reduced mEPSC frequency. Based on these findings, the authors concluded that axonal trogocytosis leads to synapse elimination. Additionally, astrocyte-contacted axons appeared to contain larger synapses, leading to the hypothesis that synaptic elimination strengthens a subset of remaining synapses. This hypothesis was further supported by electrophysiology, which revealed increased sEPSC frequency and enhanced LTP, suggesting increased synaptic connectivity. Behavioral experiments also indicated enhanced memory. The generation of a tool to precisely and specifically induce trogocytosis is undoubtedly valuable for future research across multiple fields. This study is elegant and thorough, addressing a key question in the neuron-glia field: Does presynaptic trogocytosis lead to synapse elimination? What are its functional consequences? However, I believe that some conclusions are not strongly supported by the current data, and I have recommendations to strengthen the manuscript.

Main Concerns and Recommendations

While several results point toward synapse elimination via trogocytosis, it cannot be confidently stated that this is the definitive outcome:

1. Colocalization analysis of astrocytes with synaptic markers in fixed tissue, assessed using light microscopy, is unreliable for determining engulfment due to poor axial resolution. Many colocalized puncta may not actually reside within astrocytes. The observed increase in colocalized puncta could result from increased synaptic density outside of astrocytes, as suggested in Fig. 2c. Has this possibility been explored? Additionally, presenting Imaris-rendered colocalized puncta while omitting external signal may be misleading. It would be beneficial to include original images alongside the renderings.
2. The presence of synaptic material within astrocytes does not necessarily imply that the originating synapse was eliminated.
3. The reduced synapse count in axons contacting astrocytes could be due to trogocytosis, but it is also possible that astrocytes preferentially contact or trogocytose axons with fewer synapses. This raises the question of axonal selectivity, discussed below.

Collectively, the data suggest that trogocytosis leads to synapse elimination, but I would tone down the conclusions and address the points above in the discussion.

A significant portion of the manuscript focuses on the downstream effects of induced trogocytosis, particularly on synaptic function. However, further investigation into astrocyte-neuron interactions would provide critical insight into the process. Specifically, it remains unclear whether astrocytes target select axons and whether they preferentially trogocytose certain regions:

1. What axons are being targeted by astrocytes? Is targeting purely a matter of proximity, or do astrocytes preferentially interact with axons exhibiting specific activity patterns or lower synaptic counts? If syntrogo has been active for some time (unclear from the results section), what proportion of CA1 axons would be expected to undergo trogocytosis by the time of analysis? Addressing this question would clarify the dynamics of the process. Additionally, analyzing only axons in contact with astrocytes may be misleading if these represent new targets rather than the consequence of trogocytosis. A general change in synaptic density across the entire CA1 axonal population, rather than just in astrocyte-contacted axons, might be expected.
2. What part of the axon is being trogocytosed—shaft or bouton? Comparing the shaft/bouton ratio of ongoing trogocytosis with the overall shaft/bouton ratio of axons would clarify whether specific structures are targeted. If boutons are preferentially engulfed, does this depend on astrocytic process proximity?
3. What is being engulfed? Is it primarily membrane (as shown *in vitro*), cytoplasm with vesicles (as suggested in the text), or mitochondria (as suggested by figures)? A characterization of vesicle content similar to the *in vitro* analysis would shed light on the underlying cellular mechanisms.

The authors propose that selective synapse elimination strengthens remaining synapses through resource reallocation. While this is an attractive hypothesis, the data do not allow for strong conclusions:

1. Syntrogo axons in contact with astrocytes exhibit larger synapses than controls. The authors conclude that synapses enlarge as a result from synapse elimination. However, as synapse elimination has not been unequivocally proven, this interpretation should be cautious. Instead of enlargement of synapse, it is possible that axons with larger synapses might be preferentially contacted, or if elimination occurred, the largest synapses may have been spared, shifting the average size. A bouton size distribution analysis would help clarify this point.
2. If boutons are preferentially targeted, how are the eliminated synapses selected? Why are some synapses spared?
3. The resource allocation hypothesis is appealing, but how would resources be redistributed if microglia eliminate entire synaptic boutons? This highlights the importance of a more detailed analysis of astrocytic inclusion content.

Minor Concerns and Recommendations

1. Scattered electrophysiology experiments throughout the manuscript (Fig. 2, Fig. 4, Fig. 5, Fig. 6) make it difficult to follow. Consolidating these into a single figure focused on synaptic function would improve clarity.
2. In the CLEM section, the correlation between light microscopy and EM data should be shown. Additional examples of trogocytosis with EM depth sequences and 3D rendering would strengthen the argument that axonal structures are being trogocytosed rather than merely enwrapped.
3. For all 3D Imaris renderings, corresponding original images should be shown, as presenting only renderings can be misleading.
4. References to original research should be prioritized over reviews wherever possible, as reviews do not allow readers to assess primary data.
5. The results and discussion sections could be expanded. The rationale for engineering the synthetic interaction is not well explained, and methodology descriptions are sometimes insufficient. The discussion should address the major concerns raised above and consider whether astrocytic trogocytosis occurs physiologically in vivo.
6. Supplementary Fig. 2a lacks a legend—what does iRFP label?
7. The control group needs clearer explanation. Including it in the schematic in Fig. 2 would help.
8. The use of "trogocytosis" or "trogocytosis" labeling can be confusing, as trogocytosis occurs in the control group. Using "synthetic trogocytosis" (synTrogo) or "induced trogocytosis" (i-Trogo) would provide clarity.
9. L113: What does Supplementary Video 5 illustrate? Expanding on this would help.
10. L121: Fig. 7b may contain a labeling error—should "igk- α GFP-TM-iRFP" be "igk- α GFP-TM-RFP"?
11. L129: "No discernible ligand was detected in astrocytes"—does this suggest that astrocytes do not trogocytose axons under physiological conditions?
12. L141: Most GFP signal does not overlap with astrocytes, which seems inconsistent with high-magnification images showing colocalization.
13. L160: How were membrane thickness measurements performed, and how does this indicate association? More methodological detail is needed, and the number of analyzed axons should be increased.
14. L172: Are these components presynaptic? Providing examples and a z-stack series would clarify the origin of engulfed material.

Version 1:

Reviewer comments:

Reviewer #2

(Remarks to the Author)

The authors have devised a novel strategy to artificially enhance structural remodeling of synapses by augmenting the phagocytic capabilities of astrocytes. My main concern was whether this strategy can truly be regarded as the introduction of a synthetic 'eat-me' signal. The interaction between GFP and the GFP nanobody is unrelated to endogenous 'eat-me' signals. The authors have appropriately reconsidered how to describe this experimental paradigm.

Other concerns previously raised have also been sincerely and adequately addressed, including the addition of new experiments where feasible.

The authors' approach to introducing a synthetic 'find-me' signal does not specifically induce phagocytosis of circuits associated with a particular learning paradigm. As I noted in my previous review, I still believe that the enhancement of seemingly random phagocytic activity may have promoted a meta-plastic state in the brain, thereby facilitating learning in the mice—akin to the enhanced plasticity observed after stroke. The authors have acknowledged this possibility and incorporated it into the Discussion. I believe this explanation best accounts for the observed effects in the authors' experiments. While emphasizing this interpretation further would be welcome, I respect the authors' perspective and find their discussion of this point satisfactory.

Overall, the manuscript has been thoroughly revised, and I have no further comments.

Reviewer #3

(Remarks to the Author)

The authors have addressed all my questions well and have greatly improved the study, and I believe they have also addressed the concerns of the other reviewers as reasonably as possible. Thus I would recommend publication of the current manuscript.

Reviewer #4

(Remarks to the Author)

I thank the authors for their efforts in addressing the reviewer's previous comments and for the additional experimental work. SynTrogo appears to be a potentially powerful tool, and the in vitro characterization is notably thorough. However, several critical issues raised in the initial review remain insufficiently addressed, and some of the newly added data raise further

concerns. In particular, the manuscript would benefit from a more objective presentation of the results—especially regarding the *in vivo* data, where the authors occasionally make overreaching claims. The manuscript advances the idea that SynTrogo enables selective synapse removal and subsequent strengthening of the remaining synapses, yet I believe the current dataset does not fully support these conclusions. Several interpretations would benefit from more cautious phrasing, and some additional data or clarification is needed. Below, I outline what I consider essential points to address prior to publication.

Major 1: Claim that syntrogo induces synapse engulfment and elimination without axonal degradation

1) Assessment of long-term axonal fate:

The long-term integrity of SynTrogo-expressing axons following sustained interaction with astrocytes expressing anti-GFP remains unclear. If I understand correctly, the added Extended Data Fig. 12 examines SynTrogo axons in animals lacking receptor expression on astrocytes. If this is the case, it should be explicitly clarified in the methods.

One could expect that syntrogo axons in proximity to astrocytes will be nibbled on as long as they stay close to astrocytes bearing the anti-GFP, leading to axonal fragmentation. If this is not the case, the authors should propose an explanation for what prevents syntrogo axons from being entirely munched by astrocyte until disappearance.

Moreover, if SynTrogo clusters at specific sites along the axon (as suggested by the punctate appearance of GFP signal), this may promote localized elimination.

2) Interpretation of colocalization analyses:

Colocalization analyses using confocal to prove internalization is problematic. Overlay of puncta with astrocytes does not prove that these puncta are within astrocytes, because the axial resolution of confocal microscopy is poor. A punctum sitting above or below the cell in the z dimension will produce colocalized signal in confocal, mistakenly giving the impression that the punctum is within the cell. This is a common issue in fluorescence microscopy, and the issue was clearly exemplified using CLEM in Weinhard et al. 2018.

I think the authors should tone-down their interpretations and avoid claiming synapses are internalized based on these experiments. Instead, it can only be stated that synaptic material appears colocalized with astrocytes, which suggests engulfment.

3) Concerns regarding CLEM methodology:

The criteria used to select SynTrogo versus control axons for further analyses appear inconsistent. The criteria used to select SynTrogo versus control axons for CLEM appear inconsistent. According to the authors' response, SynTrogo GFP+ axons were selected based on contact with astrocytes, while control GFP+ axons were selected based on proximity, and no information was provided for SynTrogo GFP- axons. This introduces a confound, as contact status could influence synapse density independently of SynTrogo expression. A consistent selection criterion—especially with respect to astrocyte proximity—is essential.

In addition, the length of axon analyzed in each EM volume should be clearly reported, as short segments could introduce significant variability. Ideally, a table listing all analyzed axons, their contact status, length, and annotated synapses (for both GFP+ and GFP- axons) should be provided. This would improve transparency and allow more rigorous evaluation of synapse density data (e.g., in Extended Data Fig. 16).

4) Interpretation of synapse density differences:

That SynTrogo axons contacted by astrocytes contain fewer synapses than control axons is consistent with pruning, but not conclusive. It remains possible that astrocytes preferentially contact axons with fewer synapses. While the authors attempt to rule this out using the AAI index, the method and interpretation remain unclear. A more informative control might compare axons contacted by astrocytes, with or without SynTrogo, in the same spatial context.

5) Quantification of bouton vs. shaft engagement:

The astrocyte–axon interface ratio used to assess bouton vs shaft preference lacks methodological detail. Moreover, surface contact area does not necessarily correlate with engulfment frequency. Given the emphasis on engulfment events (e.g., pinching and closure), it would be more appropriate to quantify how frequently such events occur on boutons vs shafts. Comparing this to the relative surface area of boutons and shafts could help determine whether synaptic structures are disproportionately targeted. That said, if trogocytosis is uniformly induced and astrocytes happen to reside near synapses more than shafts, this could create an apparent bias that is not mechanistically meaningful. The term “preferential pruning” should be used with caution.

6) Lack of correlation between confocal and EM images:

The manuscript lacks direct correlation between light and EM images of GFP+ inclusions. For a claim as central as synaptic internalization, it is important to provide examples where confocal images showing GFP+ inclusions can be directly correlated to closed inclusions in EM. Despite the high density of GFP+ structures observed in anti-GFP astrocytes, no such examples are shown with synaptic vesicles clearly enclosed. Furthermore, many EM examples show engulfment of mitochondria, yet not synaptic vesicles.

If mitochondria are frequently present in closed inclusions, this raises questions about whether cytosolic content is being ingested, which contrasts with earlier claims that membrane components are primarily transferred. A more systematic report of how many closed inclusions contain synaptic vesicles and/or mitochondria (out of the total observed) would greatly improve clarity. Confocal–EM correlations, presented as z-stacks, would also strengthen the interpretation.

7) Evidence for complete synapse elimination remains indirect:

Even if synaptic material is ingested, this does not establish that full synaptic structures are eliminated. While the data

suggest SynTrogo may promote synaptic pruning, no single experiment provides definitive evidence. To strengthen this interpretation, live imaging of synaptic elimination in the presence of astrocyte–SynTrogo interaction would be highly informative. In the absence of such evidence, the conclusions regarding pruning should be more cautiously framed.

Major 2: Claim that syntrogo induces synaptic pruning which strengthen remaining synapses

1) Dynamic language such as “progressive stages,” “enlargement,” or “thickening” is used to describe observations from fixed tissues:

L275: “astrocytes display progressive stages of engulfment”

L295: “significant enlargement of GFP-labeled boutons”

L241: “This increased membrane thickness”

Such terms may imply temporal progression, which cannot be concluded from static EM images. More neutral descriptions would be appropriate.

2) Interpretation of higher bouton size:

That SynTrogo axons contain larger boutons than controls may suggest strengthening of remaining synapses, but this remains speculative. It is also important to confirm whether the compared groups were equitably selected. If SynTrogo axons were chosen based on confirmed contact with astrocytes, and controls were selected only by proximity, this would introduce bias. I encourage the authors to clarify their selection criteria and to rephrase speculative conclusions.

Minor comments:

Figure 2: Please indicate which AAV serotype was used for infection.

Figure 2: Image processing appears inconsistent between SynTrogo and control conditions—e.g., SynTrogo images appear more smoothed. Please confirm uniform processing.

Extended data Figure 9: Since Synaptophysin-TagBFP was co-infected with GFP-TM, why would synaptophysin not co-localize with GFP?

Figure 2/ extended data Figure 10: The quantification methods for colocalized puncta differ between Extended Data Fig. 10 and Fig. 2, making direct comparison difficult. A standardized approach or clearer justification for the differences would be helpful

L105: “astrocytes can promote natural synaptic pruning”: Please provide a reference

L135: “Cell types”: This should be corrected to “cell lines” to more accurately describe the experimental system shown.

L175: “in the absence of synaptic engulfment”: Consider rephrasing

L180: “[cartoon showing engulfment]”: The cartoon implies definitive engulfment of synaptic material, which is not yet convincingly demonstrated; suggest modifying it to reflect uncertainty.

L198: “S100B used to label astrocytes”: Why was GFAP, which provided better signal quality, replaced with S100B?

L199: “SynTrogo-treated animals showed dense GFP signal in astrocytes”: What is the origin of this dense GFP signal in SynTrogo astrocytes? If due to engulfment, why is it not degraded quickly as shown in extended data?

L200: “GFP signal was significantly higher in SynTrogo animals”: A potential control to make sure higher GFP signal colocalized with astrocytes in synTrogo is not fortuitous due to higher signal in the field of view is to compare the amount of non-colocalized GFP signal between control and syntrogo. Expectedly, the non-colocalized (ie external) signal should be lower or at least the same between the two conditions.

L201: “[figure showing colocalization]”: Show separated channels for clarity on the proportion of signal truly inside vs. outside astrocytes.

L203: “vGlut1-RFP”: If RFP is used for colocalization analysis, please include this channel in the figure panel.

L205: “Colocalization quantification of GFP and vGlut1-RFP in astrocytic soma”: I don’t see clear colocalization in the images. Clarify how the quantification was done—was it restricted to the soma such as in extended data figure 10?

L208: “[figure panels with dotted lines showing astrocytic processes]”: The yellow outline in the control appears offset. Also, image smoothing looks more pronounced in SynTrogo, which could bias the impression of colocalization.

L210: “Fig. ED10a”: This part of the figure and caption could benefit from clearer explanation. Why does the GFP signal appear more clustered in SynTrogo? Were images processed differently? This experiment might be important enough to include in the main figure.

L214: “[Imaris reconstructions vs original confocal]”: Some GFP puncta overlaid with astrocytes in the raw images are missing in the Imaris analysis. This could affect quantification and reduce the apparent overlap between GFP and synaptic markers.

L215: “[unlabeled color channels in image]”: Please add a legend to specify which markers and channels are shown in this panel.

L217: “Density of synaptic vesicle signal”: Clarify what this refers to—is this the entire field of view, or only areas that colocalize with astrocytes?

L230: “astrocytes strongly favor contact with SynTrogo-infected axons”: This phrasing may be too strong. Suggest rewording to indicate a trend or preference rather than a definitive conclusion.

L240: “ratio of engulfed to total GFP+ puncta”: Please define what this ratio represents, how it's calculated, and what units (if any) are used.

L248: “[Methods describing sample size not shown in figure]”: How many astrocytes were analyzed per group? How many closed inclusions were identified, and how many contained synaptic vesicles?

L250: “control = AAV expressing GFP alone”: Please clarify in each section what “control” refers to, especially if control axons are also GFP-positive—it can otherwise be misleading in figures and legends.

L265: "[confocal images with varying smoothness]": The GFP channel appears filtered or smoothed more in SynTrogo than control images, which could affect interpretation—please standardize image processing.

L270: "astrocytes show no preference in contact": This conclusion would require comparing contacted vs non-contacted axons. As phrased, it may be overreaching.

L273: "Closed inclusions show vesicular content including mitochondria": Please include z-stacks to confirm these are truly closed inclusions. Also, are mitochondria transfer events consistent with earlier claims that only membrane-associated components are transferred?

L275: "astrocytes display progressive stages of engulfment": Since EM is static, terms like "pinching stage" or "emergence" are not appropriate. Consider rephrasing to describe morphological differences rather than temporal stages.

Version 2:

Reviewer comments:

Reviewer #4

(Remarks to the Author)

I would like to thank the authors for their thorough and thoughtful responses, for addressing the issues previously raised, and for providing additional material. It is clear that this manuscript represents a considerable amount of work, and the approaches employed are both elegant and technically demanding. The concept of artificially inducing trogocytosis is highly innovative—particularly in the context of potential synaptic targeting—and the in vitro characterization is well executed.

However, the in vivo evidence remains unconvincing. As noted in the previous revision round, the colocalization of VGlut1 puncta with astrocytes does not constitute proof of synaptic elimination, as the resolution of confocal microscopy is insufficient to confirm internalization, especially along the z-axis. For this reason, identification and further examination of colocalized puncta by Correlative Light and Electron Microscopy (CLEM) were crucial, and I commend the authors for undertaking this technically challenging approach.

Despite these efforts, and despite the numerous GFP- and synapse-positive puncta shown to colocalize with astrocytes in light microscopy, no double-membrane inclusion in the EM dataset could be correlated with a colocalized GFP punctum. Likewise, no examples of fully closed inclusions containing synaptic vesicles were provided. Together, these observations argue against synaptic elimination and instead suggest that the apparent vGlut/GFP puncta colocalized with astrocytes may represent imaging artefacts arising from limited confocal resolution. As the authors mention, it remains possible that, by the time of imaging (three weeks after Syntrogo expression), all GFP–receptor pairs had already been depleted, precluding observation of active elimination at that stage.

Rather than showing closed inclusions, the data predominantly display extensive apposition of GFP+ axons and RFP+ astrocytes, with axons traversing astrocytic territories and occasionally wrapped or fenestrated by astrocytic processes (see Figure 3, Extended Figures 14 and 15, and Videos 6 and 7). This extensive coverage is likely a consequence of GFP/anti-GFP binding. Such physical interaction could plausibly reduce axonal accessibility for synapse formation and thereby contribute to the observed decrease in synapse number, redistribution of presynaptic components, enlargement of remaining synapses, and associated electrophysiological and behavioral phenotypes.

In light of these points, if the manuscript were to be considered for publication, I would recommend a substantial revision of the abstract, text, and figure legends to ensure the results are presented more cautiously and without implying that GFP/anti-GFP coupling induces synaptic engulfment. While synaptic elimination cannot be definitively excluded as a possible mechanism preceding the imaging time point, it is not demonstrated convincingly in the current data. Additionally, descriptive terms such as "pinching," "closure," "degradation," or "lysis" should be used with care, as these processes remain speculative. For example, an axon passing through an astrocyte and contacting a phagocytic compartment (Extended Data Figure 15) cannot be confidently described as undergoing degradation.

Furthermore, the quantification strategy should be clarified. It would be important to explicitly describe what n represents in each analysis and to ensure that trends remain consistent when using biological replicates (n = animals) rather than individual cells or synapses. Given that n = 3 animals, it is possible that the overall trend could be disproportionately influenced by a single individual. Demonstrating that the same effect holds true when averaging at the animal level would strengthen the robustness of the conclusions.

Finally, reorganization of the figures could greatly improve readability. Some data currently placed in the Extended Data could be incorporated into the main figures, and the overall number of figures could be reduced to enhance clarity and narrative flow.

Version 3:

Reviewer comments:

Reviewer #4

(Remarks to the Author)

I appreciate the substantial efforts the authors have made over several rounds of revision to refine the manuscript and more carefully articulate the limitations of the in vivo data. At this final stage, my comments are focused on ensuring that terminology and figure presentation consistently and unambiguously reflect what is directly supported by the evidence. Given the current level of scrutiny in the field of glial-mediated pruning, clear distinction between demonstrated astrocyte–axon interactions and more speculative interpretations regarding synaptic elimination will be particularly helpful for readers. Addressing the minor points below will help prevent potential misinterpretation and further strengthen the clarity and impact of the manuscript upon publication.

1. Terminology: The term “SynTrogo” remains ambiguous. Since “Syn” implies “synaptic,” it suggests “synaptic trogocytosis”—a framing no longer supported by the in vivo data. I recommend a term that explicitly reflects induced trogocytosis (e.g., “i-Trogo”, or anything of the like) to avoid reader confusion.

2. Correlation vs. Causation: Passages such as L344 (“Remodelling of remaining synapses under reduced synaptic connectivity”) and L439–445 (“Following this synaptic reduction, the remaining presynaptic and postsynaptic compartments exhibited coordinated structural and functional remodelling...”, “Our findings indicate that a reduction in synaptic connectivity alone can reshape the synaptic landscape...”) still imply a causal relationship. Please explicitly state that remodelling is associated with reduced connectivity, rather than necessarily resulting from it.

3. Distinguishing Colocalization from Engulfment: The wording in L191 (and anywhere else relevant) should be adjusted for technical accuracy. The phrase “accumulation of ligand puncta within soma area...” should be revised into “increased puncta colocalized with astrocytes”. It is necessary to state in the results section that light microscopy cannot definitively prove engulfment due to resolution limits, and that these observations are suggestive of engulfment but require the higher-resolution validation provided by the CLEM experiments.

4. Presentation of “Inclusions” and Figure Organization (Figs 4, 5, & 7): CLEM data show that confocal puncta correspond to axons closely apposed to or wrapped by astrocytic processes, rather than discrete, double-membrane synaptic inclusions. To clarify this:

- The CLEM data should be presented explicitly as validation of the light-microscopy colocalization analysis (e.g., by swapping or merging Figures 4 and 5).
- In the current Figure 5, no bona fide inclusions (double-membrane structures) are shown. It is therefore unclear what structures are quantified in Fig. 5g and 5h. Please avoid the term “inclusion” for enwrapped axons that are not severed. Please label structures in Fig 5d/f accordingly.
- Clarify how continuous axons could be “individualized” for diameter measurements in Fig 5g/h. Since I assume this refers to enwrapped continuous axons and not round, double membrane inclusions, I do not think diameter quantification is appropriate, and these measurements are probably redundant to the AAI previously measured. Consider removing these panels if they duplicate the contact quantification in Fig 4.
- Move panels 5j, k, and l to Figure 7, as they characterize synaptic morphology. Note: Fig 5k lacks a y-axis label.

5. While the EM data could be compatible with a “grab-and-nibble” model, they do not constitute direct evidence of elimination. Explicitly stating that colocalized puncta represent proximity or wrapping, rather than severed fragments, aligns with recent literature questioning colocalization-based pruning claims.

6. Figure 8n: The schematic depicts synaptic elimination, which is not demonstrated. This should be revised to reflect astrocyte–axon interaction and remodelling without implying pruning.

Overall, I believe this work presents a valuable and technically impressive tool, and the authors have made substantial efforts to revise the manuscript in response to prior concerns. The remaining points above are intended to further align the language, figures, and schematics with the strength and limitations of the in vivo data, and to avoid potential misinterpretation by readers. Addressing these minor clarifications would, in my view, strengthen the manuscript’s impact and ensure that its conclusions are conveyed as precisely and transparently as possible.

Point-by-point responses to reviewers' comments

We express our gratitude to all the reviewers for their constructive comments, which have been immensely helpful in identifying critical issues and improving our manuscript. We have carefully addressed all the reviewers' points and conducted the requested experiments. The newly added or revised figures are summarized in the table below. Additionally, the revised parts in the manuscript are highlighted in yellow.

Figures newly added or revised	
Main Fig. 1i	3D-rendered images of engulfed GFP ligands and endogenous synaptic markers in cultured astrocytes
Main Fig. 2c	Fluorescence images showing ligand signals along with GFAP and either a membrane marker (control) or the receptor (SynT)
Main Fig. 2d	Original confocal and 3D-rendered images showing engulfed GFP ligands and the astrocytic marker S100 β
Main Fig. 2i-k	Quantification of synaptic marker density within astrocyte territories and the CA1 stratum radiatum (SR)
Main Fig. 3b, g, h, k	Quantification of AAI percentage, inclusion number and size, and spine volume distribution with increased sample size
Main Fig. 3j	Quantification of synaptic density in control and SynTrogo groups with increased sample size, including newly added analysis of GFP-negative axons in the SynTrogo group
Main Fig. 4e,f	Analysis of mEPSC frequency and amplitude with increased sample size
Main Fig. 4i	Paired-pulse ratio analysis with representative trace added in upper right corner
Main Fig. 4j,k	AMPA receptor-mediated current kinetics: (j) rise time, (k) decay time
Main Fig. 5b	Quantification of bouton volume in control and SynTrogo groups, including updated analysis of GFP-negative axons
Main Fig. 5c	Comparative analysis of bouton size distribution between control and SynTrogo groups
Main Fig. 6a	LTP recording showing eEPSC amplitude changes with an added representative trace in the upper right corner
Extended Data Fig 2f-h	Comparative analysis of GFP nanobody-conjugated receptors for protein enrichment at cell-cell contact sites and SynTrogo efficiency
Extended Data Fig 4c-e	Assessment of cell viability at multiple time points (4, 24, 48, and 72h)
Extended Data Fig 6e-g	Analysis of quenched GFP ligand signals following lysosomal fusion

Extended Data Fig 8d-e	Co-uptake of GFP ligands and BFP-labeled synaptophysin from cultured neurons by astrocytic SynTrogo
Extended Data Fig 9b	Original confocal and 3D-rendered images showing localization of engulfed GFP ligands and synaptophysin-TagBFP puncta within S100 β -positive astrocytes
Extended Data Fig 10	Low-affinity GFP receptors failed to uptake endogenous synaptic molecules
Extended Data Fig 11a-d	Assessment of viral transduction efficiency and lack of glial reactivity changes under SynTrogo
Extended Data Fig 12	SynTrogo did not induce neuronal or axonal loss at 3 weeks and 2 months
Extended Data Fig 13	Application of SynTrogo in the motor cortex–striatum circuit
Extended Data Fig 14	Preferential astrocytic contact with axonal boutons over shafts
Extended Data Fig 15b-c	Quantification of membrane thickness and presynaptic vesicle diameter
Extended Data Fig 15d-e	EM images showing membrane inclusion lysis during Pinching and Closure stages
Extended Data Fig 15f-g	3D-reconstructed EM images showing inclusion bodies originating from axons
Extended Data Fig 17j	Representative traces of evoked EPSCs
Extended Data Fig 19	Added new experimental data examining colocalization of GluA1 and synaptopodin using immunofluorescence

Reviewer #1 (Remarks to the Author):

We sincerely appreciate Reviewer #1's positive assessment of our work and constructive remarks. According to these comments, we have made the following changes and further improved our manuscript.

Summary:

"Remodeling Synaptic Connections via Engineered Neuron-Astrocyte Interactions" by Kim, Won, Kim et al. introduces SynTrogo, a synthetic molecular approach to manipulating neuron-astrocyte interactions. By engineering neurons with an "eat-me" signal and astrocytes with a complementary receptor, the study leverages astrocytic trogocytosis to remove presynaptic components selectively. The authors conducted extraordinary, high-quality work to validate SynTrogo in vitro. Using HEK293T, HeLa cells, and primary astrocyte-neuron co-cultures, they employed advanced live-cell microscopy and flow cytometry to rigorously demonstrate a trogocytosis-like process, providing a compelling mechanistic foundation for their findings. They then extend their findings in vivo, focusing on the hippocampus and showing through light microscopy, electrophysiology, correlative light-electron microscopy, and fear conditioning paradigms—that SynTrogo induces targeted synapse elimination, leading to compensatory synaptic strengthening, enhanced long-term potentiation (LTP), and improved fear memory stability. This manuscript effectively contextualizes its findings within the existing literature. While the approach is novel and elegant, our enthusiasm is tempered by the absence of key experiments and some overreaching claims that require clarification. Before acceptance and publication, addressing the following concerns is needed to ensure the accuracy of the main claims of the study:

Major Comments:

Major 1 : The authors claim throughout the manuscript that SynTrogo induces a synaptic "eat-me" signal, but experimental data do not support this assertion. Their evidence is limited to showing that SynTrogo-positive astrocytes exhibit increased internalization of presynaptic GFP and a slight elevation of internalized VGLUT1(Figure 2.G). However, this does not conclusively demonstrate that SynTrogo functions as an "eat-me" signal. Rather, it suggests that a small amount of presynaptic VGLUT1 may be incidentally internalized alongside the GFP-tagged receptor rather than actively targeted for removal. Additionally, the observed synaptic effects could stem from excessive adhesion between astrocytes and neurons due to synthetic interactions between the GFP nanobody and the GFP receptor rather than active synaptic remodeling. Conducting an adhesion strength analysis would significantly strengthen the authors' claims by distinguishing between adhesion-driven and trogocytosis-mediated effects, namely the notion that the synaptic abnormalities are due to the intake of the GFP-positive membrane.

We thank the reviewer for raising this important concern regarding the mechanistic basis of SynTrogo. As the reviewer pointed out, endogenous synaptic molecules can indeed be co-

internalized alongside the GFP ligand by receptor-expressing cells through SynTrogo. However, as shown in **Extended Data Fig. 7**, this co-uptake likely results from the engulfment of molecules adjacent to the GFP ligand during its internalization. Furthermore, as newly presented in **Extended Data Fig. 8**, when receptor-expressing astrocytes engulfed ligand-expressing cultured neurons, neighboring synaptophysin-TagBFP was co-internalized together with the GFP ligand, while non-tagged TagBFP was not. This finding further supports the proposed SynTrogo mechanism, in which adjacent synaptic components are co-engulfed along with the targeted membrane.

Importantly, in our original manuscript, we referred to the GFP ligand as an “eat-me” signal to describe its role in marking presynaptic neuronal membranes, rather than to suggest it directly labels synaptic components for removal. Related to this point, as also noted by reviewer #2, we have revised this terminology to “find-me” signal throughout the revised manuscript to avoid any confusion.

We also appreciate the reviewer’s insightful comment regarding the distinction between adhesion-driven and trogocytosis-mediated effects. Ideally, to fully disentangle these processes, one would require a receptor–ligand pair that allows strong intercellular adhesion without triggering trogocytosis. However, as shown in **Extended Data Fig. 2e–h**, binding affinity and trogocytosis efficiency show a significant positive correlation, making it experimentally challenging to separate these two phenomena.

Nevertheless, we further explored this issue by testing alternative GFP nanobody receptors (LaG18 and LaG5) that retain ligand-binding capability but exhibit low trogocytosis efficiency (**Extended Data Fig. 2f–h**). These nanobodies induced cell–cell binding to a lesser extent than the α GFP receptor and showed minimal GFP ligand uptake. Extending these experiments in vivo, we observed that neither LaG18 nor LaG5 led to significant uptake of endogenous synaptic molecules by astrocytes (**Extended Data Fig. 10**). These results suggest that strong intercellular binding is essential for robust induction of SynTrogo and drive the internalization of synaptic molecules. We have elaborated on these points in the revised manuscript (lines 110–114, 191-196).

Major 2 Throughout the paper, the authors use Correlative light-electron microscopy (CLEM) to illustrate changes in synaptic architecture in SynTrogo-positive astrocyte-neuron interactions. However, the criteria for selecting specific images for analysis are unclear, and light microscopy reference images can be misleading. To strengthen the manuscript, the authors should provide CLEM or Immuno-Gold EM evidence showing a SynTrogo-positive astrocyte containing VGLUT1+GFP+ presynaptic vesicles.

We appreciate the reviewer’s helpful comment. For CLEM analysis, we first identified RFP-expressing astrocytes in the CA1 region that closely contacted GFP-positive axons using confocal microscopy. These regions of interest were then traced in serial electron microscopy

(EM) images to correlate ultrastructural features with the corresponding fluorescence signals. During this process, we carefully examined the serial EM datasets bidirectionally to ensure accurate identification of astrocytic processes and their associated axons. The selected astrocytic processes and the contacting axonal profiles were subsequently reconstructed in three dimensions (3D). These reconstructions confirmed direct interactions between astrocytes and axons, including enwrapping of axonal boutons containing synaptic vesicles. Furthermore, vesicle-like structures within the engulfed inclusion bodies in astrocytic territories exhibited diameters consistent with typical presynaptic vesicles (35.2 ± 3.4 nm in PMID: 9221783; 35.6 ± 5.2 nm in **Extended Data Fig. 15c**), supporting the conclusion that SynTrogo-mediated interactions between presynaptic neurons and astrocytes lead to engulfment of synaptic components. We have clarified the criteria for CLEM imaging and analysis in the revised manuscript (lines 220-225 in the Results section and lines 836–852 in the Methods section) and included additional examples of CLEM images in **Extended Data Fig. 15f-g**.

Regarding the reviewer's suggestion to provide direct CLEM or immuno-gold EM evidence of VGLUT1(+) GFP(+) vesicles within astrocytes, we attempted to detect such structures but were unable to do so. We believe this is due to several factors. First, the number of synaptic vesicles in each engulfed inclusion body is typically low, making it difficult to detect sufficient VGLUT1(+) signals using fluorescence imaging. Second, some VGLUT1(+) structures within astrocytic territories may have fused with phagolysosomes, resulting in an acidic environment that quenches GFP signals, complicating the detection of VGLUT1(+) GFP(+) vesicles. Supporting this explanation, we have now included data showing that fusion of GFP(+) vesicles with lysosomes in cultured cells leads to a substantial decrease in GFP signal (**Extended Data Fig. 6e-g**).

Nonetheless, we have provided several lines of evidence through fluorescence imaging and CLEM analyses supporting the co-uptake of the GFP ligand and synaptic components by SynTrogo (**Fig. 1i, j, Fig. 2f-k, Fig. 3e, and Extended Data Fig. 8d, 9, 15**). For example, in **Extended Data Fig. 9**, when the GFP ligand and Synaptophysin-TagBFP were co-expressed in CA3 neurons, we observed that 75.65% of Synaptophysin(+) vesicles within astrocytes also contained GFP, which is significantly higher than in the SynTrogo-negative group (34.93%). Additionally, newly added results from cultured cells (**Extended Data Fig. 8d**) further demonstrate that synaptic molecules can indeed be co-engulfed together with the GFP ligand. Taken together, these data support our conclusion that astrocytic SynTrogo can efficiently remove not only ligand-labeled neuronal membrane fragments but also adjacent synaptic components.

Major 3: In the manuscript, the authors demonstrate that SynTrogo induces synapse elimination using electron microscopy to quantify the number of synapses lost. It is unclear how the authors selected the axons and synapses quantified in Figure 4A-C.

We thank the reviewer for this important comment. In **Fig. 4a-c** of the original manuscript (now updated to **Fig. 3i-k**), axons and synapses were quantified based on CLEM analysis. Specifically, we identified axons that were in direct contact with RFP-labeled astrocytes and then quantified the number of synapses along the axonal processes. For the control group, we selected GFP-positive axons located in close proximity to RFP-positive astrocytes, even though they were not SynTrogo-positive, to ensure consistent selection criteria across experimental conditions. We have described the details of this analysis in the revised manuscript (lines 270–273 in the Results section and lines 836–852 in the Methods section).

In addition, it would be interesting to determine which types of synapses are lost due to SynTrogo. Using fluorescence analysis of inhibitory and excitatory synaptic markers to quantify the number of excitatory versus inhibitory synapses within the territory of SynTrogo-positive astrocytes.

We thank the reviewer for this helpful comment. In **Fig. 2g**, we showed a significant increase in engulfed excitatory presynaptic molecules, but not in excitatory postsynaptic or inhibitory pre-/postsynaptic molecules, within astrocytes under SynTrogo conditions compared to the control group. This suggests that SynTrogo specifically targets excitatory presynaptic neurons. Furthermore, our results in **Fig. 3i, j** demonstrated a decrease in the number of excitatory synapses under SynTrogo conditions. We have now included additional results showing that the density of excitatory, but not inhibitory, synapses was specifically reduced within SynTrogo-positive astrocytic territories (**Fig. 2j**). Moreover, the decrease in the mEPSC frequency also supports the loss of excitatory synapses (**Fig. 4e**). Taken together, these findings suggest that SynTrogo eliminates excitatory synapses. We have described newly added results in the revised manuscript (lines 197-201).

Major 4: The authors claim that SynTrogo specifically targets excitatory synapses in the CA1 region Figure 4. However, to rigorously confirm this selectivity, they should target CA1 as the presynaptic site rather than CA3, since CA1 contains most presynaptic neurons forming local inhibitory synapses. This approach would provide clearer evidence that SynTrogo preferentially affects excitatory, rather than inhibitory, synapses.

Major 5: If SynTrogo is truly specific to excitatory synapses, what drives this selectivity? The authors report increased release probability, as evidenced by more docked presynaptic vesicles. However, whether this is merely a compensatory response to synapse elimination remains unclear, or if a distinct molecular mechanism underlies this effect. Clarifying this point would significantly strengthen the study's conclusions.

We thank the reviewer for these important comments regarding the specificity and selectivity of SynTrogo towards excitatory synapses. In our current study, we targeted CA3 neurons using the CaMKII α promoter, which is well established to drive gene expression predominantly in excitatory neurons (**Fig. 2a**). This strategy allowed us to selectively label excitatory presynaptic

neurons projecting to CA1 and examine the effects of SynTrogo-mediated astrocytic pruning on excitatory synapses. Our observations that SynTrogo altered the frequency of sEPSCs, but not the amplitude of sEPSCs or the frequency and amplitude of sIPSCs (**Fig. 4b-c**), further support the notion that excitatory presynaptic function was selectively affected. While our results demonstrate that SynTrogo can effectively target and eliminate excitatory synapses (**Fig. 2g-k, 3j, and 4e**), we agree that to rigorously confirm its selectivity, future studies targeting CA1 – which contains both excitatory and inhibitory synapses – would be informative. Additionally, combining synapse labeling methods (e.g., eGRASP; PMID: 29700265) with SynTrogo in diverse neural circuits would also be valuable for more precisely defining its selectivity and expanding its applicability across different types of synapses.

In addition, our current study revealed that the SynTrogo group exhibited substantial structural and functional remodeling of the remaining synapses (**Fig. 4b, 4g-i, 5, and 6a-b**), suggesting that synaptic components undergo redistribution and reorganization following SynTrogo-mediated pruning. This is further supported by our newly added result (**Fig. 5c**), which shows that bouton size distribution analysis revealed large presynaptic boutons – absent in the control group – emerged in the SynTrogo group. These findings led us to hypothesize that this remodeling may involve the activity-dependent inter-synaptic competition for the redistribution of limited resources through molecular diffusion and cytoskeleton-dependent vesicle trafficking, as well as reorganization of intracellular organelles (e.g., mitochondria and ER) within both pre- and postsynaptic compartments (lines 436-442). Further studies dissecting these processes will be essential to fully understand the underlying mechanisms of synaptic remodeling following SynTrogo.

Minor Comments

Minor 1: Sample Size Reporting: The manuscript lacks clarity in sample size reporting. Mouse N values should be provided for all experiments to ensure data are not biased by specific animals.

As suggested by the reviewer, we have reported the number of animals used in each experiment in the figure legends.

Minor 2: Figure 1I-J: Consider adding 3D reconstructions of astrocytes with Homer and Bassoon puncta, similar to Figure 2D.

In line with the suggestion, we have added 3D-rendered images in **Fig. 1i**, to clearly illustrate the localization of Homer and Bassoon puncta within astrocytic territories.

Minor 3: Figure 2A: The experiment timeline should be clearly labeled, and the control group should be more distinctly marked.

According to the reviewer's comment, we have clarified the experimental timeline and

indicated the sets of AAVs used in both control and SynTrogo groups in **Fig. 2a**.

Minor 4: Figure 2H: The relevance of this figure to the authors' claims is unclear. Further clarification is needed.

Fig. 2h presents a significant positive correlation between the signals of engulfed GFP ligand and the excitatory presynaptic molecule, VGluT1, within individual astrocytes under SynTrogo conditions. Together with the results shown in **Fig. 2f** and **Extended Data Fig. 9b** – demonstrating partial colocalization of GFP and presynaptic molecules in astrocytic somata – these data support the idea that presynaptic component uptake is dependent on the extent of SynTrogo-mediated ligand engulfment (lines 185-188). Furthermore, newly added images in **Extended Data Fig. 8d** provide direct visual evidence of the co-uptake of GFP ligand and presynaptic molecules from cultured neurons by receptor-expressing astrocytes.

Minor 5: Figure 2J: Ensure consistency in sample sizes for sEPSC frequency (n=20) and amplitude (n=19) to avoid discrepancies.

We appreciate the reviewer's attention to this point. We identified and corrected the inconsistency between the sample sizes for sEPSC frequency (n=20) and amplitude (n=19). Specifically, we added one additional recording to the amplitude dataset and excluded one frequency-only recording, so that both analyses are now based on matched recordings from 20 cells. The updated values are reflected in the revised figure (**Fig. 4b**).

Minor 6: Figure 3E: The method for identifying the green mask as a presynaptic component is unclear. Further validation or explanation is needed.

We thank the reviewer for this helpful comment. The green mask in **Fig. 3e** highlights presynaptic components, which were identified based on characteristic ultrastructural features observed in serial EM sections. Specifically, when a structure appeared to be enwrapped by astrocytic processes, we carefully examined and reconstructed its serial profiles. This analysis consistently revealed that the enwrapped structures were presynaptic boutons, which could be identified by clusters of synaptic vesicles and their close apposition to a postsynaptic density (PSD) on a dendritic spine. Moreover, the size of the vesicle-like structures within the membrane inclusion bodies in **Fig. 3e** is consistent with the typical diameter of presynaptic vesicles (35.2 ± 3.4 nm; PMID: 9221783), further supporting the conclusion that the green mask represents engulfed presynaptic components. We have clarified this point in the revised manuscript (lines 248-256).

Figure 4E: The control group for mEPSC (n=6 cells) is too small to draw strong conclusions. Additional N is recommended.

In response to the reviewer's suggestion, we conducted additional electrophysiological recordings to increase the number of cells analyzed in both groups (n=9 and 15 cells for the control and SynTrogo groups, respectively). The updated data are presented in **Fig. 4e-f** and

confirm our original finding that mEPSC frequency, but not amplitude, is altered under SynTrogo conditions.

Figure 5G: The slower eEPSC decay kinetics in SynTrogo-treated cells could result from high release probability leading to AMPAR saturation. Analyzing AMPAR decay kinetics would further support conclusions on increased release probability.

We appreciate the reviewer's thoughtful interpretation regarding the possibility of AMPAR saturation as a cause for altered eEPSC kinetics. However, our data show that AMPAR decay and rise kinetics did not differ significantly between groups (**Fig. 4j-k**), and notably, AMPAR-mediated currents were reduced under SynTrogo conditions (**Fig. 6l**). As the previously presented traces did not appropriately represent the overall dataset, we have now replaced them with more accurate examples. These results argue against AMPAR saturation and instead suggest that the synapses are in a more plastic state, characterized by reduced basal AMPAR activity that can be subsequently upregulated upon learning (**Fig. 6l**).

Figure 5E-F: Sample sizes should be consistent between Figures 5E and 5F, which appear to be derived from the same experiments. The absence of two control cells in Figure 5F should be addressed.

We appreciate the reviewer's careful observation. In **Fig. 5e** of the original manuscript (now updated to **Fig. 4g**), two control recordings were mistakenly included in the n count, even though they did not reach 100 EPSC events due to a protocol issue (80 EPSC events). This led to an inflated n number in the frequency analysis. In contrast, the n=8 in **Fig. 5f** (now updated to **Fig. 4h**) accurately reflects the number of successful recordings that met the analysis criteria. We have corrected the sample size in **Fig. 4g** to match this and updated the figure legend accordingly.

Figure 6K-L / Extended Figure 14J: There is a discrepancy between AMPAR-mediated EPSCs and the SynTrogo condition. In Figure 6L, AMPA currents decrease in SynTrogo (w/o FC), yet eEPSC amplitude increases in SynTrogo (Extended Figure 14J). Since eEPSCs are AMPAR-dependent, these trends should be consistent. The authors should clarify this inconsistency.

We thank the reviewer for pointing out the apparent discrepancy between the reduced AMPA currents (**Fig. 6l**) and the trend toward increased evoked EPSC amplitudes, which did not reach statistical significance (now updated in **Extended Data Fig. 17k**) observed in the SynTrogo condition. Although both measurements reflect AMPAR-mediated evoked responses, they were performed under different pharmacological and recording conditions and may therefore reflect distinct physiological mechanisms. In **Fig. 6l**, AMPAR currents were recorded under pharmacological isolation, minimizing recurrent activity and thereby primarily reflecting postsynaptic AMPAR availability. In contrast, **Extended Data Fig. 17k** was recorded under more intact network conditions, and the observed EPSCs may reflect contributions from both presynaptic release dynamics and postsynaptic receptor properties.

Notably, the reduction in AMPA current is consistent with newly included IHC data showing decreased GluA1 expression (**Extended Data Fig. 19**, lines 353-365), suggesting reduced postsynaptic AMPAR levels. In contrast, the preserved evoked EPSC amplitude may reflect compensatory increases in presynaptic release probability and vesicle pool size (**Fig. 4g-i, 5d-e**), which could functionally offset the reduced receptor availability. While we cannot exclude additional factors such as local connectivity differences, we interpret these results as reflecting complementary adaptations across synaptic compartments in response to SynTrogo. We have revised the Discussion section to clarify this interpretation (lines 425-435).

Figure 6A-B / Extended Figure 14J: Representative traces should be provided for LTP data and eEPSC input-output curves.

As recommended, we have included representative traces for the LTP recordings (**Fig. 6a**) and the eEPSC input-output curves (**Extended Data Fig. 17j**) to improve the clarity and interpretability of the electrophysiological data.

Broader Applicability: To enhance the impact of this strategy, it is essential to determine whether SynTrogo functions in brain regions beyond the hippocampus. Testing its efficacy in other circuits would broaden its potential applications.

We appreciate the reviewer's suggestion regarding the broader applicability of SynTrogo. To address this, we tested SynTrogo in the motor cortex-striatum pathway by expressing the ligand in cortical neurons and the receptor in striatal astrocytes. Similar to our findings in the hippocampus, we observed that astrocytic uptake of GFP-labeled neuronal membrane fragments derived from cortical neurons, suggesting that SynTrogo can function across diverse brain regions. These results have been included in **Extended Data Fig. 13** in the revised manuscript (lines 212-218).

Therapeutic Relevance: The authors should clarify how their findings support potential therapeutic applications for Alzheimer's disease or remove this statement from the manuscript. We thank the reviewer for raising this point. Our findings demonstrate that synaptic remodeling following pruning involves both structural and functional strengthening of synapses, along with enhanced synaptic plasticity. Given that synaptic weakening and loss are early hallmarks of AD, we speculate that uncovering the molecular mechanisms underlying this remodeling process may offer insights into how degenerating circuits could be functionally restored in the early stages of AD. We have clarified this point in the revised manuscript (lines 495-499).

Reviewer #2 (Remarks to the Author):

We are grateful to Reviewer #2 for the positive evaluation and insightful comments. According to the comments, we refined the interpretation of our findings and further improved our manuscript.

The authors devised a new strategy to artificially enhance structural remodeling of synapses by boosting the phagocytic capabilities of astrocytes. A simple approach of introducing an 'eat-me' signal in neurons and the complementary receptor in astrocytes led to enhanced synapse elimination, as well as improved synaptic plasticity and memory. The study was conducted meticulously, and the findings are both novel and exciting. I have just a few fundamental questions for the authors.

Comment 1: The SynTrogo system created by the authors does not use a traditional or endogenous phagocytosis signal. Therefore, it may not be ideal to introduce it as the 'eat-me' signal and its complementary receptor. The presynaptically expressed ligand is merely an extracellularly facing, membrane-targeted GFP, and it is not an 'eat-me' signal—it simply happened to function as one. The postsynaptically expressed receptor is an antibody (nanobody) for GFP, rather than a phagocytosis receptor. If you had modified and enhanced an endogenous 'eat-me' signal and its receptor, then this terminology would be appropriate. I would suggest using a different term when introducing this system in the abstract. Afterwards, explaining that this system essentially functions as an 'eat-me' signal and its receptor is perfectly fine.

We appreciate and agree with the reviewer's critical comment. In response, we have replaced the term 'eat-me signal' with 'find-me signal' to more accurately reflect the nature of the GFP ligand in our system. As suggested, we also clarified in the revised manuscript that although we refer to the ligand as a 'find-me' signal, it ultimately functions as an 'eat-me' signal in facilitating the uptake of neuronal membranes by the receptor-expressing astrocytes (line 34, 374, 376-379).

Comment 2: With the enhanced 'random' synapse elimination capabilities, wouldn't you expect the neuronal circuit to become disorganized, potentially leading to less optimal circuitry? I understand that synapse 'elimination' is just as important as new synapse 'formation' in reorganizing information flow and encoding memory. However, with this artificial enhancement of phagocytosis, important and less important synapses may not be properly distinguished.

We appreciate the reviewer's critical question. As the reviewer pointed out, we initially anticipated that SynTrogo-mediated phagocytosis by astrocytes might lead to circuit dysfunction due to non-selective or 'random' elimination of synapses. However, contrary to our initial expectations, our fluorescence imaging, EM image analyses, electrophysiological analyses, and behavioral analyses revealed a structural and functional remodeling of remaining synapses, accompanied by enhancements of synaptic plasticity and memory. Although we do

not yet have direct evidence to clarify whether SynTrogo-mediated synaptic pruning is random or selective for particular subpopulations of synapses in the intact brain, further studies on spatiotemporal control of SynTrogo and two-photon imaging of live animal will be necessary to address this important point. We have discussed this issue in the Discussion section of the revised manuscript (lines 450-456, 464-474).

I understand that the behavioral experiments suggest an enhanced memory effect. However, while the enhanced phagocytic capabilities may appear adaptive in the specific behavioral tests you conducted, they may not be in other contexts. If you have performed additional memory tests that yielded less favorable outcomes, it might be wise to include these data, at least as Supplementary Data. If such experiments were not conducted, discussing the possibility that enhanced phagocytic capability may not be optimal in all situations would be beneficial.

We appreciate and fully agree with the reviewer's thoughtful suggestion. Although we observed a significant increase in contextual fear memory under SynTrogo conditions overall, the results varied depending on shock intensity: mice exposed to 0.5 mA foot shocks did not show increased recent memory but exhibited memory persistence, whereas mice subjected to milder shocks (0.3 mA) displayed increases in both recent and remote memories (**Fig. 6c-f**). Additionally, we did not find any change in working memory performance in the Y-maze test (**Extended Data Fig. 18b**). These results suggest that partial elimination of synapses and subsequent remodeling in our experimental conditions affect brain functions in a task- and context-dependent manner. Although we did not comprehensively evaluate other memory paradigms in this study, we agree that enhanced phagocytic capability and synaptic pruning may not be universally optimal or beneficial in all situations. We have addressed this point in the Discussion section of the revised manuscript (lines 443–449).

Comment 3: I was not aware of the GFP nanobody (α GFP). A brief explanation of this material upon its introduction would benefit a broad readership.

In response to the comment, we have added a brief description of the GFP nanobody (α GFP) in the Result section (line 87-88) to aid clarity for a broader readership.

Comment 4: It appears that phagocytosed materials eventually end up in the lysosome. Given the acidic intra-lysosomal environment, one would expect GFP fluorescence to be quenched and rapidly degraded. However, the fluorescence seems to persist in the authors' experiments. Have the authors tracked the vesicles up to their fusion with the lysosome using live imaging? Would GFP fluorescence decrease substantially upon lysosomal uptake? If GFP fluorescence is lost in the lysosome, the overall estimation of cellular uptake by phagocytosis may underestimate the actual amount internalized. Please discuss the possibility that GFP may be quenched in the lysosome and how the loss of fluorescence could affect the analysis somewhere in the text.

We appreciate the reviewer's critical comment. In response, we conducted time-lapse imaging

to examine whether engulfed GFP ligand signals are quenched upon fusion with lysosomes. These experiments revealed that GFP fluorescence intensity decreased by approximately 50% immediately after the vesicles containing ligand-receptor complexes fused with lysosomes (**Extended Data Fig. 6e-g**). This effect was unlikely due to vesicle movement out of the focal plane, as the intensity of receptor-conjugated RFP – which is more resistant to quenching in acidic environments – remain largely unchanged. Therefore, as the reviewer suggested, the total GFP ligand may underestimate the actual amount of ligand internalized. We have included this point in the revised manuscript (lines 134-140).

Comment 5:

Figure 1f, right panel: The label "(Astrocyte)" does not need to start on a new line.

We thank the reviewer for pointing this out, and we have corrected the label in **Fig. 1f** accordingly.

Comment 6: It appears that neuronal igk-GFP-TM is taken up only by astrocytes expressing the aGFP nanobody (Figure 1f, g). Does this imply that there is almost no phagocytic activity in naïve astrocytes? Are the trogocytotic events observed by the authors purely artificially induced, requiring both the artificial ligand and the artificial receptor in adjacent cell pairs?

If this is the case, then these trogocytotic events are not an enhancement of an existing endogenous process but rather a completely artificial one. I assume the authors would not want to make this assumption, so it may be worth presenting some evidence that these phagocytic events occur, albeit to a lesser extent, under naïve conditions. If the authors believe that these trogocytotic events are entirely artificial, then that is fine. However, in that case, it may be more appropriate to avoid describing the effects of SynTrogo as an "enhancement" of a natural phagocytic process.

We appreciate the reviewer's critical comment. As noted, we barely observed uptake of GFP ligand-labeled neuronal membrane fragments by receptor-negative astrocytes in cultured conditions (**Fig. 1f-g**), whereas small but discernible ligand signals were detected in astrocytic somata in brain tissues (**Fig. 2d,f and Extended Data Fig. 9b**). This discrepancy may reflect differences in the cellular environments or the duration of neuron-astrocyte interaction (1~2 days in vitro vs. ~ 3 weeks in vivo). Moreover, EM analysis revealed that receptor-negative astrocytes can internalize neuronal membrane fragments, albeit at a lower frequency compared to the SynTrogo group. These results, together with previous studies demonstrating endogenous synaptic uptake by hippocampal astrocytes in adult mice (PMID: 33361813), support the idea that SynTrogo enhances a naturally occurring phagocytic process rather than inducing an entirely artificial phenomenon (lines 400–406).

That said, we agree with the reviewer that SynTrogo may have the potential to elicit artificial synapse removal beyond what occurs physiologically. Because SynTrogo enables ligand expression to be spatially defined, it may allow synapse removal in patterns that do not

naturally occur. This possibility represents an important direction for future studies and is now discussed in the revised manuscript (lines 469–471).

Comment 7: The EM studies presented in Extended Data Fig. 11 are particularly enlightening. In Morizawa et al. (2022), the degradation process of internal neuronal compartments was apparent even during the transition from stage 3 (Pinching) to stage 4 (Closure) in the cerebellum. Similarly, I notice some deformation of the internal compartments in panel d (Degradation). The authors could specifically highlight this observation in the manuscript and perhaps discuss the possibility of apparent deformation occurring even during stages 3 and 4, if that is indeed what they observe.

We appreciate the reviewer for this insightful suggestion. As the reviewer pointed out, the study by Morizawa et al. (2022) demonstrated degradation of internal neuronal compartments during astrocyte-mediated phagocytosis, particularly during the transition from stage 3 (Pinching) to stage 4 (Closure) in the cerebellum. Similarly, in our study, we observed structural deformation of internal axonal compartments during these same stages (**Extended Data Fig. 15 d-e**). While the extent of deformation varied, it was frequently localized to specific membrane regions and, in some cases, involved more extensive structural breakdown. We have now highlighted this similarity between our observations and the study by Morizawa et al. (2022) in the revised manuscript (lines 259–264).

Comment 8: The authors observed an increase in sEPSC frequency and a decrease in mEPSC frequency in Trogo preparations. At first glance, this suggests an increase in presynaptic spontaneous action potential firing in acute slice preparations from the Trogo mice. The authors could mention this possibility early in the Results section (page 7, line 189-). The intrinsic properties of the postsynaptic CA1 pyramidal neurons were evaluated (Extended Data Fig. 14), and no differences were found. Similarly, the authors could have evaluated the presynaptic CA3 pyramidal neurons or assessed spontaneous firing frequency through unit extracellular recording. If these experiments were not done, that is fine; however, discussing the possibility of an increase in presynaptic spontaneous action potential firing could be valuable.

We thank the reviewer for this insightful suggestion and fully agree that the observed increase in sEPSC frequency, alongside a reduction in mEPSC frequency, may reflect elevated spontaneous presynaptic action potential firing in SynTrogo-treated slices. To directly assess presynaptic excitability, we attempted whole-cell patch-clamp recordings from CA3 pyramidal neurons using various optimized cutting solutions. However, we encountered consistent technical difficulties in obtaining stable recordings, likely due to a combination of the animals' age and additional tissue fragility induced by viral injections. As an alternative preliminary study, we performed c-Fos immunostaining and found a significant increase in c-Fos expression in CA3 pyramidal neurons of SynTrogo-treated mice (**Supporting Figure 1**), supporting the possibility of enhanced neuronal activity in this presynaptic population.

Together with the increased release probability observed in our paired-pulse recordings (Fig. 4i), these findings suggest that presynaptic excitability and firing activity in CA3 may be elevated following SynTrogo. We now mention this possibility in the Discussion section of the revised manuscript (lines 420-422).

Supporting Figure 1. Analysis of cFos expression in CA3 and CA1 neurons under control and SynTrogo conditions

If the spontaneous action potential firing frequency is assumed to be approximately the same between the two preparations, then the enhancement of release probability in response to presynaptic spontaneous action potential firing becomes more likely. It may also be important to point out that mEPSC amplitude (Fig. 4f) and sEPSC amplitude (Fig. 2j) are similar, suggesting the absence of multisynaptic contact between an axon and a postsynaptic dendrite, as well as the lack of multivesicular release at a single synaptic contact. While these points may seem trivial, they would support the authors' claim that presynaptic release probability is enhanced in the remaining synapses in SynTrogo preparations.

We thank the reviewer for the thoughtful suggestion. To test whether the increased sEPSC frequency in CA1 neurons reflects enhanced presynaptic release probability rather than increased spontaneous firing from CA3, we attempted whole-cell recordings from CA3 pyramidal neurons. Due to technical limitations in obtaining stable recordings in this unique region, we were unable to acquire reliable data. As we noted above, we found a modest increase in cFos levels in CA3 neurons under SynTrogo conditions compared to the control conditions. This suggests the possibility of elevated presynaptic activity, although we recognize that c-Fos does not directly reflect action potential firing and may be influenced by additional factors.

Although we cannot rule out a contribution from increased AP input, the interpretation that SynTrogo enhances presynaptic release probability remains best supported by the data. The amplitudes of mEPSC and sEPSC were unchanged, which argues against multivesicular release or multisynaptic connections, as suggested by the reviewer. Paired-pulse ratio was reduced and vesicle docking was increased, both consistent with enhanced presynaptic release efficacy. We have revised the Discussion to incorporate these clarifications (lines 425-435).

Comment 9: The authors found an increased PSD area (Extended Data Fig. 13). In many

previous studies, an increase in PSD area usually reflects an increase in the number of postsynaptic AMPA receptors that the PSD accommodates. With an increase in the number of AMPA receptors, one would typically expect an increase in the amplitude of mEPSC and sEPSC. However, such an increase in amplitude was not observed in the authors' data. This discrepancy is acceptable, as it reflects the actual recordings.

However, it may be worthwhile to point out this discrepancy between ultrastructural and electrophysiological data. The noise in the electrophysiological data, which may prevent the smallest events from being captured, could explain this discrepancy. If the average amplitude of mEPSC or sEPSC events actually increases, the smallest events would become large enough to be detectable. This would give the appearance that mEPSC or sEPSC frequency had increased without an increase in their amplitude. There is no doubt that presynaptic release probability has increased, as shown in the paired pulse experiment in Fig. 5g and 5h. Collectively, the authors' data support the idea that both presynaptic and postsynaptic properties are enhanced in the pruning-resistant synapses.

We thank the reviewer for this thoughtful and insightful comment. As noted by the reviewer, our EM analyses showed an increase in PSD area under SynTrogo conditions (**Fig. 5m**), which is often associated with increased AMPA receptor content in postsynaptic membranes. However, we did not observe a corresponding increase in the amplitude of mEPSCs or sEPSCs, and also found that AMPA current and the AMPA/NMDA current ratio were decreased under SynTrogo conditions (**Fig. 6j-m**). This discrepancy from the expectations based on previous studies is indeed intriguing.

As the reviewer suggested, technical noise in electrophysiological recordings could contribute to this discrepancy, although we interpret the enlarged PSDs as representing a preparatory state that provides structural scaffolding for future AMPA receptor recruitment and functional maturation. Supporting this interpretation, our EM analyses revealed that the PSD area increase was primarily significant in SA+ spines under SynTrogo conditions (**Fig. 5m**), highlighting the potential role of SA+ spines in AMPAR-associated postsynaptic remodeling.

To further investigate this hypothesis, we performed immunostaining for GluA1 (an AMPAR subunit) and synaptopodin (a marker of the spine apparatus), and analyzed their levels and colocalization under SynTrogo conditions both with and without fear conditioning (**Extended Data Fig. 19**). Interestingly, we found that while GluA1 levels initially decreased under SynTrogo compared to the control group, they recovered after fear conditioning. In contrast, synaptopodin levels of SynTrogo group remained consistently elevated regardless of conditioning, compared to those of the control group. Moreover, the number and proportion of spines containing both GluA1 and synaptopodin were reduced without conditioning in the SynTrogo group, but these values recovered to levels comparable to the control after conditioning.

These results support the notion that SynTrogo-induced pruning initiates postsynaptic

remodeling that may facilitate subsequent AMPAR recruitment during experience-dependent plasticity, ultimately enhancing synaptic plasticity. We have clarified and elaborated on these points in the revised manuscript (lines 425-435).

Comment 10: Not only showing the enhancement of fear-conditioning memories but also demonstrating the enhanced capability for extinction was a great experiment conducted by the authors. This shows that the remaining neuronal circuit, resistant to artificially enhanced phagocytic activity, has higher adaptability.

However, this raises the question of what these excess synapses were originally for. If the system can function adaptively well without these synapses, why were they prepared in the first place? It would be worthwhile to discuss the adaptive advantage of having excess, non-essential synapses in naïve animals.

We thank the reviewer for this interesting and thoughtful comment. During development, neurons initially form excess synapses, which are later refined and stabilized through experience-dependent pruning, a critical mechanism underlying synaptic remodeling and circuit optimization. Although our study was conducted in adult mice, our findings that SynTrogo-mediated pruning enhances memory performance and extinction suggest that intrinsic programs for synaptic pruning-mediated plasticity and remodeling may persist in the adult brain.

This observation raises an important question as suggested by the reviewer: why does the brain maintain an excess of synapses in adulthood if they can be pruned without impairing, and even enhancing, memory function? One possible explanation is that maintaining an excess of synapses provides a buffer against sudden synaptic loss due to injury, inflammation, or other insults, thereby supporting circuit resilience and facilitating adaptive remodeling under challenging conditions. It is also possible that the evolutionary trajectory of the brain favors adaptability over strict optimization, ensuring that circuits remain flexible and responsive to dynamic environments.

While our study demonstrates that SynTrogo can enhance synaptic plasticity and functional reorganization in adult hippocampal circuits within specific experimental contexts, further studies will be necessary to determine whether these findings are generalizable to other brain regions and environments. We have incorporated these points into the revised Discussion section (lines 443-449, 457-463).

Comment 11: It would also be important to discuss the number of CA3 pyramidal cells that express igk-GFP-TM. If all CA3 cells express this protein and all synapses are eliminated, then there would no longer be a functional hippocampal circuit. The authors could provide an estimate of the relative ratio of infected cells along with the relative ratio of total endogenous synapses that have been eliminated. The outcome of the behavioral experiments may depend crucially on this ratio. It may well be that the surviving synapse ratio just happened to be

optimal for encoding and extinction of fear memories. Please discuss the possibility of the serendipity of the virus infection ratio that resulted in a 'favorable' outcome.

We thank the reviewer for this critical and insightful comment. To address this issue, we performed additional experiments under the same conditions used for the behavioral and EM analyses and quantified the transduction efficiency of the GFP ligand in CA3 pyramidal neurons and the receptor in CA1 astrocytes. We found that approximately 73.4% (control) and 80.3% (SynTrogo) of CA3 neurons expressed the ligand, and 78.3% (control) and 84.4% (SynTrogo) of CA1 astrocytes expressed both RFP and the receptor.

Despite these high transduction efficiencies, our EM analyses revealed only ~27% reduction in synapse density. Immunostaining confirmed similar reductions in excitatory synapses, both within astrocytic territories (~21% reduction) and across the entire CA1 region (~24% reduction). These findings indicate that SynTrogo does not uniformly eliminate all synapses, even within highly transduced cell populations. They also suggest that factors beyond simple expression levels likely influence the extent of synaptic pruning and its functional consequences.

For example, previous studies have shown that approximately 57% of hippocampal synapses are in close proximity to astrocytic processes (PMID: 10436047), suggesting that not all synapses are equally accessible for astrocyte-mediated elimination. In addition, dynamic changes in astrocyte-synapse proximity during neuronal activity (PMID: 25042585) may further modulate SynTrogo efficiency. Our EM and immunostaining results also suggest that SynTrogo cannot eliminate all synapses, even when both ligands and receptors are highly expressed. This resistance to elimination may relate to the synaptic strength and structural stability of individual connections.

In summary, while transduction efficiency may contribute to the observed outcomes, we believe that additional factors—such as synaptic accessibility, activity-dependent modulation, the intrinsic capacity of astrocytes to eliminate synapses, and the structural stability of synaptic connections—play more decisive roles. We have clarified and discussed these points in the revised manuscript (lines 450–456).

Comment 12: I assume that not all synapses that presynaptically express igk-GFP-TM are eliminated before fear conditioning. I also assume that endogenous phagocytosis may be enhanced in control animals after fear conditioning. It may be worthwhile to observe whether there is an increase in the phagocytosis of igk-GFP-TM after fear conditioning in control animals, where there is no expression of igk-alphaGFP-TM in CA1 astrocytes, and in those with its expression. I do not believe the authors have conducted such experiments. Therefore, it would be valuable to discuss the possibility of using this system to evaluate (1) whether there is endogenous phagocytic activity during memory paradigms, and (2) whether the enhancement of the elimination of remaining synapses in SynTrogo animals can also be observed.

We thank the reviewer for this insightful and constructive suggestion. To assess whether fear conditioning influences astrocytic uptake of the GFP ligand, we analyzed astrocytic GFP levels with and without memory formation. Interestingly, we observed a robust increase in GFP uptake one day after fear conditioning in the SynTrogo group, whereas no significant change was detected in the control group, which lacked receptor expression in astrocytes (**Supporting Figure 2**). These results suggest that ligand uptake can be enhanced in an activity-dependent manner when the receptor is present, potentially reflecting the dynamic regulation of neuron–astrocyte interactions reported in previous studies (PMID: 25042585).

However, the absence of a detectable increase in the control group does not necessarily indicate a lack of endogenous phagocytic activity. It is possible that our readout lacked sufficient sensitivity to detect subtle levels of endogenous ligand uptake. Previous studies have reported activity-dependent synapse elimination by astrocytes in the hippocampus (PMID: 33361813), suggesting that astrocytic phagocytosis can occur under physiological conditions and may be modulated by behavioral experience.

As the reviewer insightfully noted, future studies using *in vivo* two-photon imaging and receptor-independent reporters will be essential to directly evaluate whether endogenous astrocytic phagocytosis is enhanced during learning and whether residual synapses in SynTrogo conditions are selectively eliminated in an activity-dependent manner. We have discussed these possibilities in the revised manuscript (lines 440–442).

Supporting Figure 2. Activity-dependent increase in astrocytic ligand uptake in SynTrogo

Comment 13: In the Discussion (page 9, line 250), the authors write "selective elimination of synapses." I do not recall whether the authors demonstrated that trogocytosis occurs selectively for eliminating synaptic components. My impression was that both the ligand and receptor are expressed broadly on neurons and astrocytes, respectively, and that phagocytosis could occur anywhere on the cell. The use of the term "selective" may be misleading.

We appreciate and agree with the reviewer's critical comment. Initially, we used the term "selective" to describe how our system can define the synaptic connections to be eliminated at the circuit level (e.g., the CA3–CA1 circuit). However, as the reviewer correctly pointed out, our system expresses both the ligand and receptor broadly across the neuronal and astrocytic

membranes and relies heavily on the natural proximity between neurons and astrocytes. Therefore, astrocytes could, in principle, trogocytose any part of the neuronal membrane they contact.

Importantly, in our newly updated EM analysis (**Extended Data Fig. 14**), we found that astrocytic processes preferentially interacted with presynaptic boutons rather than axon shafts in the control group, and this preference was even more pronounced in the SynTrogo group. This suggests that while there is a preference for synapses, it is not truly “selective” in the strict sense. We have described these findings (lines 227-232) and removed the term “selective” from the revised manuscript to avoid confusion.

Additionally, the term "selective" may also mislead readers into assuming that the SynTrogo system was targeted to selectively eliminate memory-unrelated synapses. This was not the case. The igk-GFP-TM was expressed randomly on CA3 pyramidal cells using the CamKII promoter with AAV. The expression was not driven, for example, by an activity-dependent promoter such as *cfos*. Since there was no selective expression, the term "selective" may be confusing.

We thank the reviewer for this additional critical comment. As the reviewer correctly pointed out, in our current study, the GFP ligand was expressed randomly across CA3 pyramidal neurons using the CaMKII α promoter delivered by AAV, rather than being restricted to specific neuronal populations through an activity-dependent promoter like *cfos*. Therefore, the expression pattern was not selective, and the term “selective” may indeed be misleading in this context. To truly achieve selectivity for specific neuronal or synaptic populations, we agree that developing a next-generation SynTrogo system that can target defined subsets of neurons or activity-tagged synapses would be essential, as we described in the revised manuscript (lines 471-474).

Comment 14: I think I may have understood why the memory performance was enhanced in animals where a portion of random synapses was eliminated by artificial trogocytosis. The AAV injection, three weeks prior, would have significantly reduced the number of synapses by the time of the behavioral experiments. This is an unusual situation, which could be similar to a pathological condition such as focal stroke in some respects. The neuronal circuit would attempt to adapt to this situation through homeostatic synaptic scaling, and the overall meta-plastic state may be enhanced to adapt to this emergency. It is known that neuronal circuit plasticity increases sometime after a stroke, and rehabilitation during this window of opportunity can result in favorable outcomes or reorganization of the remaining neuronal circuit, leading to functional recovery. The behavioral experiments were conducted specifically during this period of enhanced plasticity. Perhaps a discussion like this, comparing it to focal stroke pathology, may provide additional insights.

We thank the reviewer for this interesting and insightful comment. We fully agree that homeostatic synaptic scaling mechanisms may contribute to the observed enhancements in

synaptic plasticity and memory performance following SynTrogo. As noted by the reviewer, previous studies have shown that synaptic elimination induced by brain injuries such as stroke can trigger compensatory mechanisms that strengthen the remaining synapses and promote circuit remodeling (PMID: 16634041, 19888284). Although our study primarily focused on SynTrogo-mediated synapse elimination and subsequent remodeling in normal mice, we acknowledge that similar processes may occur after brain injuries, where a transient window of enhanced plasticity facilitates functional recovery. We have incorporated a brief discussion of this possibility in the revised manuscript (lines 490–495). Understanding these mechanisms may offer valuable insights into how synaptic loss in various brain disorders could be leveraged for therapeutic strategies.

Reviewer #3 (Remarks to the Author):

We express our gratitude to the Reviewer #3 for positive evaluation and constructive comments. Based on these valuable suggestions, we have implemented the following changes and further improved the manuscript.

In this study, Kim et al engineer a method to target specific cells for partial digestion by another cell, by displaying a GFP domain on the target cell surface and a anti-GFP nanobody with a RFP intracellular domain on the other cell surface. They show this works with HEK293 and HeLa digester cells, name the system synthetic trogocytosis (SynTrogo), then apply it to CA3 neuronal targets and CA1 astrocyte digesters in vivo. They perform detailed EM and ephys studies that show ingestion of presynaptic components by the CA1 astrocytes and reduced presynaptic bouton density with enlargement of the remaining boutons in axons that express the SynTrogo ligand, compared to axons in uninjected control (the control group was actually infected, but had no GFP nanobody for SynTrogo) mice (if I interpret the controls correctly). They also find structural changes in postsynaptic structures at baseline compared to control. Interestingly, they observe enhanced LTP after theta-burst stimulation in CA3-CA1 connections and faster learning in mice subjected to CA3 axon - CA1 astrocytic SynTrogo.

Synthetic trogocytosis is a conceptually novel method for controlling cell structure and function and is likely to have various applications in development, tissue engineering, and neurobiology. Thus I am very enthusiastic about this study as a candidate for publication in Nature Communications. There are just some clarifications needed that I expect the authors can address in a revision.

Important information necessary for reproducing the results, understanding the strength of the effects, and understanding the limitations of the method are missing, and should be provided as follows:

1. In Figures 2-6 and related extended figures, the time of analysis relative to injection and the age of the mice should be stated.

We thank the reviewer for this helpful suggestion. In response, we have included details regarding the time of analysis and the age of the mice in the Methods section of the revised manuscript (lines 615, 626-629).

2. The percent of neurons that express SynTrogo ligand and the percent of astrocytes that express SynTrogo receptor in the ephys and behavioral experiments (currently Figs 4-6) should be quantified and presented. Since presumably < 100% of CA3 neurons express SynTrogo ligand and <100% of CA1 astrocytes express SynTrogo receptor, this information is necessary to interpret the significance of the observed changes in synaptic function measured in the CA1 neurons. The numbers should be discernable from the EM reconstructions of Figs 3-5 if these

were performed under the same conditions, but this would require volumetric reconstruction of axons and astrocytes by EM which may be difficult. Alternatively, the investigators can redo the experiment of Fig 2 under the conditions used for Figures 5 and 6 with a NeuN stain in CA3 and a GFAP stain in CA1 so they can estimate transduction efficiency in CA3 neurons and CA1 astrocytes.

We thank the reviewer for this helpful comment. In response, we performed additional experiments under the same conditions used for Figures 5 and 6 and conducted immunostaining for NeuN and GFAP. As a result, we found that the transduction efficiencies were $74.06 \pm 5.02\%$ (control) and $80.34 \pm 7.28\%$ (SynTrogo) for CA3 neurons, and $78.3\% \pm 4.68$ (control) and $84.37 \pm 3.14\%$ (SynTrogo) for CA1 astrocytes. These data are now included in **Extended Data Fig. 11a-d** and are described in the revised manuscript (lines 202–204).

3. Long-term effects of the system are unclear. The experiment to investigate effects on cell viability is too short, with just 4h of coincubation of SynTrogo ligand and receptor cells (Extended Data Fig 4). This should be extended to 1 and 3 days.

We thank the reviewer for this critical comment. In response, we extended our cell viability assay up to 3 days, and found no significant difference between SynTrogo and non-SynTrogo groups (**Extended Data Fig. 4d-e**). In addition, our 18-hour live-cell imaging (**Supplementary Video 4**) demonstrated that both donor and receiver cells exhibited normal proliferation during SynTrogo conditions. Taken together, these results indicate that SynTrogo does not cause any detrimental effects on cell physiology.

4. Likewise, will the targeted axons eventually get completely removed? If so, will the CA3 neurons lacking axons die? Thus investigating the effects of SynTrogo on the CA3 neuron morphology and survival at one or more timepoints at least 1 month after the latest timepoint shown here (which I assume to be 3 weeks as that was mentioned for the ephys) would be useful. I might suggest 2 and 3 months after viral transduction.

We appreciate the reviewer for this helpful comment. In response, we assessed the number of CA3 neurons at both 3 weeks and 2 months after viral transduction using NeuN immunostaining and found no significant differences in neuronal number, indicating that long-term SynTrogo expression did not affect cell viability (**Extended Data Fig. 11d (right), 12a-b**). To investigate whether axons might be removed by SynTrogo, we co-expressed the GFP ligand along with a non-tagged fluorescent protein in CA3 neurons to visualize axonal morphology. We observed no evidence of axonal loss even 2 months after viral injection (**Extended Data Fig. 12c-d**). Additionally, newly added results from live-cell imaging of SynTrogo between cultured neurons and astrocytes revealed that neurite structures remained intact throughout the course of SynTrogo (**Extended Data Fig. 8e**). Taken together, these results indicate that, under our experimental conditions, astrocytic SynTrogo of CA3 neuronal axons does not induce axonal removal or neuronal death. We described these results in the

revised manuscript (lines 120-123, 205-211).

In addition, there are a few revisions needed to make the conclusions more understandable or more robust.

5. It appears the EM observations of Trogo CA3 axonal and CA1 dendritic structural changes are comparing Trogo groups to non-transduced groups, based on the mention of “groups” in the methods. However this needs to be confirmed and mentioned explicitly in each figure and in the text. Right now there is a consistent lack of mention of what the observed changes are being compared to, for example, “SynTrogo significantly reduced the number of ligand-labeled axonal boutons and their coupled dendritic spines, “mitochondria volume of presynaptic boutons increased”, “the spine volume and PSD area significantly increased” etc. That is, it is important to confirm if the comparisons are always between hippocampi with or without the SynTrogo treatment (ideally same mouse, unilaterally injected and using the contralateral hippocampus as control), or between axons/dendrites in apposition to a SynTrogo receptor-expressing astrocyte vs axons/dendrites not in apparent contact.

We thank the reviewer for this helpful comment. The control group for in vivo experiments consisted of mice expressing GFP ligands in CA3 neurons and membrane-targeted RFP in CA1 astrocytes. In response, we have clarified in the revised manuscript what the SynTrogo group was compared to in each relevant figure and text section.

6. Related to this, in Fig 5, besides presenting the changes in bouton number and size in GFP (ligand)-expressing axons relative to uninjected hippocampi, the authors should also determine whether there are any differences in bouton number and size in non-expressing axons relative to uninjected hippocampi. Given the changes in EPSC frequency observed in postsynaptic non-genetically altered CA1 neurons, one wonders if even non-genetically altered CA3 axons in injected hippocampi might experience some feedback regulation that alters their presynaptic structure. These comparisons should be achievable using the existing EM dataset.

We appreciate the reviewer for this critical comment. In response, we conducted additional analyses using our existing EM dataset. In **Fig. 3j and 5b**, we analyzed both synapse density and bouton volume across three groups: GFP(+) axons, GFP(-) axons in the SynTrogo group, and control axons. Synaptic density was significantly reduced in GFP(+) axons compared to controls, while GFP(-) axons showed no significant difference from controls (**Fig. 3j**). Similarly, bouton volume was significantly increased in GFP(+) axons compared to controls, with no significant difference observed between GFP(-) axons and controls (**Fig. 5b**). These results indicate that synaptic pruning and subsequent remodeling in our system are primarily restricted to ligand-expressing axons, suggesting that feedback regulation from the postsynaptic side is unlikely to be a major contributor in this context. We have incorporated

these points in the revised manuscript (lines 270–277, 295-298).

7. I would suggest a reorganization of Figures 4-6 as currently the way they present the data make the conclusions difficult to evaluate. The main problem is I do not think one can robustly relate EM data to ephys data, or ephys data to behavior, without detailed manipulations that prove the connection, but each of these figures presents a correlation to imply the functional relationship without proving it. Meanwhile data that would be easier to understand together are divided between different figures. Specifically, there is EM data in Figs 4-5 and ephys data in Figs 4-6. In addition Fig 4 as currently structured does not stand well as a main figure on its own, as it does not generate a robust conclusion, while it analyzes the same dataset as the EM panels in Fig 5. The authors can consider grouping the EM panels from Figs 4-5 together into Fig 4 only, and grouping ephys data from Figs 4-6 into Fig 5 only. The text of the results section would then also have to be reorganized to present EM, ephys, and behavioral findings in that order.

We thank the reviewer for the thoughtful suggestion. In response, we have reorganized the figures and corresponding text to present EM, electrophysiology, and behavioral data more coherently.

8. “We found a decrease in the proportion of SER(-) spines, but no change in the SER(+) spine percentage” sounds like an internally illogical sentence. To fix this, it should be rewritten as “We found a decrease in the proportion of SA(-) SER(-) spines, but no change in the SA(-) SER(+) spine percentage, due to an increase in the percentage of SA(+) spines”

We thank the reviewer for meticulous feedback. In response, we have revised the description as suggested (lines 313-315).

Reviewer #4 (Remarks to the Author):

We sincerely appreciate Reviewer #4's positive evaluation of our work and valuable feedback. In response to these comments, we have carefully revised and improved the manuscript as detailed below.

In this manuscript, Kim and co-authors engineered a system to induce trogocytosis and study its downstream effects. To achieve this, a selected cell bearing anti-GFP (receptor) recognizes surface-GFP (ligand) expressed by a target cell, leading to their interaction and the engulfment of small particles from the ligand-expressing cell by the receptor-expressing cell. The authors first extensively characterized this system in vitro, demonstrating that it requires actin polymerization and that the ingested material consists primarily of membrane. This tool was subsequently applied in vivo to investigate the functional consequences of axonal trogocytosis. Using viral injections, the authors ectopically expressed the receptor in CA1 astrocytes and the ligand in CA3 neurons in the hippocampus, inducing synthetic interactions and axonal nibbling by astrocytes. Staining and colocalization analyses suggested that synaptic material was engulfed. Correlative light and electron microscopy revealed that axons in contact with astrocytes contained fewer synapses, and electrophysiology data indicated reduced mEPSC frequency. Based on these findings, the authors concluded that axonal trogocytosis leads to synapse elimination. Additionally, astrocyte-contacted axons appeared to contain larger synapses, leading to the hypothesis that synaptic elimination strengthens a subset of remaining synapses. This hypothesis was further supported by electrophysiology, which revealed increased sEPSC frequency and enhanced LTP, suggesting increased synaptic connectivity. Behavioral experiments also indicated enhanced memory.

The generation of a tool to precisely and specifically induce trogocytosis is undoubtedly valuable for future research across multiple fields. This study is elegant and thorough, addressing a key question in the neuron-glia field: Does presynaptic trogocytosis lead to synapse elimination? What are its functional consequences? However, I believe that some conclusions are not strongly supported by the current data, and I have recommendations to strengthen the manuscript.

Main Concerns and Recommendations

Major 1: While several results point toward synapse elimination via trogocytosis, it cannot be confidently stated that this is the definitive outcome:

1. Colocalization analysis of astrocytes with synaptic markers in fixed tissue, assessed using light microscopy, is unreliable for determining engulfment due to poor axial resolution. Many colocalized puncta may not actually reside within astrocytes. The observed increase in colocalized puncta could result from increased synaptic density outside of astrocytes, as suggested in Fig. 2c. Has this possibility been explored? Additionally, presenting Imaris-

rendered colocalized puncta while omitting external signal may be misleading. It would be beneficial to include original images alongside the renderings.

We appreciate the reviewer's important comment. In response, we have now included original fluorescence images with all external signals that clearly show internalized GFP ligands and synaptic molecules within astrocytic territories (**Fig. 2d-f**). Although **Fig. 2c** shows enrichment of neuron-derived GFP ligands around astrocytes, this does not indicate an increase in synaptic density in that area. In fact, as shown in the newly added results in **Fig. 2j**, we observed a decrease in the number of excitatory synapses within SynTrogo-positive astrocytes. Furthermore, our EM analysis (**Extended Data Fig. 15f-g**) confirmed that the synaptic components from axons were physically internalized within astrocytes. Taken together, these data support our conclusion that the observed ligand and synaptic marker signals in astrocytic regions represent genuine engulfment rather than an artifact of increased external synapse density.

2. The presence of synaptic material within astrocytes does not necessarily imply that the originating synapse was eliminated.

We thank the reviewer's critical comment. Based on multiple complementary approaches – including in vitro/in vivo fluorescence imaging, EM analysis, and electrophysiology – we conclude that SynTrogo can eliminate synapses in the target neural circuit by co-engulfing neuronal membranes and associated synaptic materials. However, as the reviewer rightly noted, we have not yet directly visualized the process of in vivo synaptic pruning per se in our current experimental setting. This would indeed represent an important next step for future investigations, which we have now acknowledged in the revised Discussion (lines 440-442).

3. The reduced synapse count in axons contacting astrocytes could be due to trogocytosis, but it is also possible that astrocytes preferentially contact or trogocytose axons with fewer synapses. This raises the question of axonal selectivity, discussed below. Collectively, the data suggest that trogocytosis leads to synapse elimination, but I would tone down the conclusions and address the points above in the discussion.

We thank the reviewer for raising this important point. In response, we carefully analyzed EM data from the control group to determine whether astrocytes preferentially contact axons with fewer synapses. However, we did not observe any such preference (**Extended Data Fig. 14d-f**), indicating that astrocytes do not selectively target axons with fewer synapses under basal conditions (lines 234–238).

Major 2: A significant portion of the manuscript focuses on the downstream effects of induced trogocytosis, particularly on synaptic function. However, further investigation into astrocyte-neuron interactions would provide critical insight into the process. Specifically, it remains unclear whether astrocytes target select axons and whether they preferentially trogocytose

certain regions:

1. What axons are being targeted by astrocytes? Is targeting purely a matter of proximity, or do astrocytes preferentially interact with axons exhibiting specific activity patterns or lower synaptic counts? If SynTrogo has been active for some time (unclear from the results section), what proportion of CA1 axons would be expected to undergo trogocytosis by the time of analysis? Addressing this question would clarify the dynamics of the process. Additionally, analyzing only axons in contact with astrocytes may be misleading if these represent new targets rather than the consequence of trogocytosis. A general change in synaptic density across the entire CA1 axonal population, rather than just in astrocyte-contacted axons, might be expected.

We appreciate the reviewer for raising this important point. Regarding axon targeting specificity, we believe that proximity is the main determinant of axon targeting by astrocytes under our current experimental conditions. Our EM analysis of the control group showed no significant correlation between axon-astrocyte interaction and synapse density under basal conditions (**Extended Data Fig. 14f**). Given that approximately 57% of hippocampal synapses are in close proximity to astrocytic processes (PMID: 10436047), our result showing a 27% reduction in synapse number (**Fig. 3j**) suggests that some level of selectivity may emerge during SynTrogo-induced synapse elimination. At the synapse level, it is also possible that astrocytes might preferentially interact with a subpopulation of synapses. Supporting this idea, previous studies have shown that astrocytic processes dynamically change over time and can contact synapses in an activity-dependent manner (PMID: 25042585).

Regarding the temporal dynamics of SynTrogo activation, in our current experimental setting – based on virus injection and subsequent protein expression – we could not determine the precise duration of SynTrogo activity or quantify the proportion of CA1 axons undergoing trogocytosis by the time of analysis. We agree that addressing this question would require future studies using an inducible SynTrogo system combined with optogenetic or chemogenetic modulation of synaptic activity, as well as two-photon imaging of live animals, to clarify these dynamics in a more controlled and temporally defined manner (lines 471-474).

Regarding the potential bias introduced by analyzing only astrocyte-contacted axons, we acknowledge that focusing on these axons alone could be misleading if they are new targets rather than the consequence of ongoing trogocytosis. To address this concern, we conducted additional analyses and showed that within astrocytic territories, excitatory synaptic density was significantly reduced, whereas inhibitory synaptic density remained unchanged (**Fig. 2j**). Importantly, when analyzing the entire CA1 axonal population, we also observed an overall decrease in excitatory synapse density (**Fig. 2k**). Notably, EM analyses revealed that GFP-negative CA3 axons in the SynTrogo group did not exhibit significant changes in bouton number or volume compared to the control group (**Fig. 3j and 5b**), suggesting that synaptic pruning and subsequent remodeling are primarily restricted to GFP-positive axons targeted by

SynTrogo. These findings imply that the overall decrease in excitatory synapses in the CA1 region likely reflects local effects in GFP-positive axons rather than widespread structural changes in all axons. We agree that further functional studies are needed to investigate these mechanisms at the circuit level.

2. What part of the axon is being trogocytosed—shaft or bouton? Comparing the shaft/bouton ratio of ongoing trogocytosis with the overall shaft/bouton ratio of axons would clarify whether specific structures are targeted. If boutons are preferentially engulfed, does this depend on astrocytic process proximity?

We thank the reviewer for this important comment. In response, we analyzed the interface ratio between astrocytic processes and either axonal boutons or shafts in the control group. We found that the bouton-astrocyte interface ratio was significantly higher compared to the shaft-astrocyte interface ratio (**Extended Data Fig. 14a,c**). This difference was even more pronounced in the SynTrogo group (**Extended Data Fig. 14b-c**), indicating that astrocytes preferentially target the membranes of boutons rather than shafts. We have described these findings in the Results section of the revised manuscript (lines 227–232).

3. What is being engulfed? Is it primarily membrane (as shown in vitro), cytoplasm with vesicles (as suggested in the text), or mitochondria (as suggested by figures)? A characterization of vesicle content similar to the in vitro analysis would shed light on the underlying cellular mechanisms.

We thank the reviewer for this important question. As the reviewer pointed out, our EM analyses revealed that the primary component being engulfed through SynTrogo is the GFP ligand-embedded plasma membrane. This process is accompanied by the co-uptake of adjacent cytoplasmic components, including vesicle-like structures and, more rarely, mitochondria. Regarding the vesicle structures in the membrane inclusion bodies engulfed by astrocytes, the size of these vesicles (~35 nm in diameter) is consistent with that of synaptic vesicles (PMID: 9221783), supporting the idea that they likely represent synaptic vesicles. Our in vivo EM and fluorescence analyses consistently demonstrated the presence of synaptic vesicles and components within the internalized compartments (**Fig. 2f-k, Fig. 3e, Extended Data Fig. 8d, 9, 15**). These results suggest that SynTrogo-mediated engulfment primarily involves plasma membrane and adjacent synaptic vesicles, while other organelles like mitochondria are rarely internalized.

Major 3: The authors propose that selective synapse elimination strengthens remaining synapses through resource reallocation. While this is an attractive hypothesis, the data do not allow for strong conclusions:

1. SynTrogo axons in contact with astrocytes exhibit larger synapses than controls. The authors conclude that synapse enlarge as a result from synapse elimination. However, as synapse elimination has not been unequivocally proven, this interpretation should be cautious. Instead

of enlargement of synapse, it is possible that axons with larger synapses might be preferentially contacted, or if elimination occurred, the largest synapses may have been spared, shifting the average size. A bouton size distribution analysis would help clarify this point.

We thank the reviewer for this insightful comment. In response, we analyzed the correlation between bouton surface area and the axon-astrocyte interface (AAI) ratio (**Extended Data Fig. 14d-e**). We found no significant correlation between bouton size and AAI ratio in either the control or SynTrogo group, indicating that astrocytes do not preferentially contact boutons based on their size. To further test the resource reallocation hypothesis, we performed a bouton volume distribution analysis (**Fig. 5c**). Interestingly, the SynTrogo group displayed an increased frequency of larger boutons compared to the control group, including some sizes not observed in the control. This suggests that SynTrogo pruning leads to the survival or growth of a subpopulation of large boutons that might benefit from redistributed synaptic resources after the elimination of neighboring synapses.

Taken together, these findings support the notion that bouton enlargement in the SynTrogo group is not due to pre-existing size preference by astrocytes but rather likely arises from synaptic remodeling following the elimination of a subset of synapses. We have clarified these points in the revised manuscript (lines 234–238, 299-301).

2. If boutons are preferentially targeted, how are the eliminated synapses selected? Why are some synapses spared?

We appreciate the reviewer's critical question. One possible explanation for the selective elimination of a subpopulation of synapses is that only approximately 57% of synapses in the hippocampus are in close proximity to astrocytic processes, based on a previous study (PMID: 10436047). Another possibility is that smaller synapses may be more vulnerable to SynTrogo-mediated elimination, whereas larger synapses might be relatively resistant due to higher levels of synaptic adhesion molecules that strongly connect pre- and postsynaptic membranes. Further studies are needed to rigorously test these hypotheses. We have included these points in the revised manuscript (lines 407-411, 450–456).

3. The resource allocation hypothesis is appealing, but how would resources be redistributed if microglia eliminate entire synaptic boutons? This highlights the importance of a more detailed analysis of astrocytic inclusion content.

We appreciate the reviewer for this insightful question. As pointed out, previous studies have shown that the size of phagocytosed material by microglia can be significantly larger than that of astrocytes (PMID: 32637606, 28642575). Therefore, it is plausible to imagine scenarios in which microglia eliminate larger portions of synaptic structures, ultimately leading to the removal of entire synaptic boutons. In such cases, the amount of synaptic components available for redistribution might be too limited to support effective synaptic remodeling. A detailed comparison between SynTrogo-mediated engulfment by astrocytes and microglia would be

valuable for addressing this point in future studies, specifically examining how extensively each cell type engulfs synaptic structures, the persistence of the process, the proportion of synaptic components contained within the engulfed material, and the availability of residual resources for subsequent remodeling. We have described this point in the Discussion section of the revised manuscript (lines 478–483).

Minor Concerns and Recommendations

Minor 1. Scattered electrophysiology experiments throughout the manuscript (Fig. 2, Fig. 4, Fig. 5, Fig. 6) make it difficult to follow. Consolidating these into a single figure focused on synaptic function would improve clarity.

We thank the reviewer for this helpful suggestion. In response, we have consolidated the electrophysiology experiments into a single figure (**Fig. 4**), except for the LTP and AMPA/NMDA current ratio experiments, which have been grouped together with the behavioral data.

Minor 2. In the CLEM section, the correlation between light microscopy and EM data should be shown. Additional examples of trogocytosis with EM depth sequences and 3D rendering would strengthen the argument that axonal structures are being trogocytosed rather than merely enwrapped.

We thank the reviewer for this helpful suggestion. In the revised manuscript, we have now provided additional examples (**Extended Data Fig. 15d-g**) that demonstrate the correlation between fluorescence microscopy and EM image and support our conclusion that axonal structures are being trogocytosed rather than merely enwrapped.

Minor 3. For all 3D Imaris renderings, corresponding original images should be shown, as presenting only renderings can be misleading.

As suggested by the reviewer, we have now included original images corresponding to all 3D Imaris renderings (**Fig. 2d,f and Extended Data Fig. 9b**).

Minor 4. References to original research should be prioritized over reviews wherever possible, as reviews do not allow readers to assess primary data.

As suggested, we have replaced review articles with original research references where possible in the revised manuscript.

Minor 5. The results and discussion sections could be expanded. The rationale for engineering the synthetic interaction is not well explained, and methodology descriptions are sometimes insufficient. The discussion should address the major concerns raised above and consider whether astrocytic trogocytosis occurs physiologically in vivo.

We thank the reviewer for this helpful comment. In response, we have expanded both the

Results and Discussion sections to include additional rationale for engineering the synthetic interaction (lines 83–90), as well as more detailed descriptions of the methodologies employed. In the revised Discussion, we also addressed the reviewer’s major concerns regarding the interpretation of our data and considered whether astrocytic trogocytosis may occur physiologically in vivo. These changes have been incorporated into the revised manuscript (lines 400–406).

Minor 6. Supplementary Fig. 2a lacks a legend—what does iRFP label?

The iRFP used in **Extended Data Fig. 2a** is a non-tagged infrared fluorescent protein that was expressed to fill the cell volume. We have added an explanation in the legend to clarify this point.

Minor 7. The control group needs clearer explanation. Including it in the schematic in Fig. 2 would help.

As suggested by the reviewer, we have now included information about the injected viruses for the control group in the schematic in **Fig. 2a**.

Minor 8. The use of "trog" or "trogocytosis" labeling can be confusing, as trogocytosis occurs in the control group. Using "synthetic trogocytosis" (synTrog) or "induced trogocytosis" (i-Trog) would provide clarity.

We thank the reviewer for this helpful suggestion. To improve clarity and avoid confusion with endogenous trogocytosis, we consistently used the term “SynTrog” (synthetic trogocytosis) throughout the revised manuscript. In the graphs, due to space limitations, “SynT” was used as an abbreviated label to denote SynTrog. We have ensured that this abbreviation is used consistently and is clearly defined in the figure legends to prevent misunderstanding.

Minor 9. L113: What does Supplementary Video 5 illustrate? Expanding on this would help.

Supplementary Video 5 illustrates the cellular behaviors observed during SynTrog of ligand-expressing HEK293 cells by receptor-expressing astrocytes. We have further elaborated on these observations in the Results section (lines 126-129).

Minor 10. L121: Fig. 7b may contain a labeling error—should "igκ-αGFP-TM-iRFP" be "igκ-αGFP-TM-RFP"?

We confirm that the correct construct used in **Extended Data Fig. 7** was indeed igκ-αGFP-TM-iRFP. Because the PKH26 dye emits red fluorescence, we used the infrared fluorescent protein (iRFP) to avoid spectral overlap with the dye.

Minor 11. L129: "No discernible ligand was detected in astrocytes"—does this suggest that astrocytes do not trogocytose axons under physiological conditions?

We thank the reviewer for this critical comment. Under cultured conditions, we barely observed uptake of neuronal membranes by astrocytes in the control group; however, in brain slices, we

consistently detected small but noticeable amounts of GFP ligands within astrocytic somata in the control group (**Fig. 2d,f and Extended Data Fig. 9b**), suggesting that astrocytes can indeed trogocytose axons under physiological conditions. Importantly, the advantage of our synthetic platform is that it allows us to define the target sites for elimination by expressing and localizing the GFP ligand; thus, SynTrogo could potentially be applied to various synapses that are not naturally eliminated by astrocytes. We have clarified this point in the Results section of the revised manuscript (lines 179-180, 400-406, 469-471).

Minor 12. L141: Most GFP signal does not overlap with astrocytes, which seems inconsistent with high-magnification images showing colocalization.

We thank the reviewer for this helpful comment. In the lower-magnification images presented in **Fig. 2b**, we showed that GFP signals largely overlapped with receptor signals within astrocytic territories. Using GFAP staining, we further confirmed that the enriched GFP signals were indeed located in regions occupied by astrocytes, although many GFP signals appeared outside the GFAP-positive areas. This is because GFAP predominantly labels the cytoskeletal structures of astrocytes and does not fully represent the entire cytoplasmic territory. To avoid any confusion, we replaced the GFP-only images with high-magnification merged images showing clear colocalization of GFP and receptor signals in **Fig. 2c**.

Minor 13. L160: How were membrane thickness measurements performed, and how does this indicate association? More methodological detail is needed, and the number of analyzed axons should be increased.

We thank the reviewer for raising this important point. In response, we have provided a more detailed description of the method used to measure membrane thickness in the revised manuscript (lines 239-243, 842-849). As shown in the newly updated graph (**Extended Data Fig.15b**), membranes at sites of astrocytic contact were significantly thicker compared to those in the control group. We interpret this increased membrane thickness, along with a narrowed intermembrane space, as indicative of a tighter molecular association or adhesion between astrocytic and axonal membranes, rather than membrane fusion. Additionally, as suggested by the reviewer, we increased the number of analyzed axons to strengthen the reliability of our conclusions.

Minor14. L172: Are these components presynaptic? Providing examples and a z-stack series would clarify the origin of engulfed material.

The component shown in the serial EM images of **Fig. 3e** are indeed presynaptic in origin. In the Z-stack series labeled “Degradation”, we observe a mitochondrion within the engulfed compartment that undergoes partial degradation inside the astrocyte. 3D reconstruction of the axon containing this engulfed material further demonstrated that it forms synapses onto dendritic spines, characterized by clustered synaptic vesicles and an adjacent postsynaptic density (PSD), consistent with typical presynaptic axonal morphology (**Supporting Fig. 3**). In

our revised manuscript, we included additional examples that clearly demonstrate the axonal origin of the engulfed materials (**Extended Data Fig. 15f-g**).

Supporting Figure 3. 3D-reconstructed EM images revealing the presynaptic origin of astrocyte-engulfed material.

REVIEWER COMMENTS

Reviewer #2 (Remarks to the Author):

The authors have devised a novel strategy to artificially enhance structural remodeling of synapses by augmenting the phagocytic capabilities of astrocytes. My main concern was whether this strategy can truly be regarded as the introduction of a synthetic 'eat-me' signal. The interaction between GFP and the GFP nanobody is unrelated to endogenous 'eat-me' signals. The authors have appropriately reconsidered how to describe this experimental paradigm.

Other concerns previously raised have also been sincerely and adequately addressed, including the addition of new experiments where feasible.

The authors' approach to introducing a synthetic 'find-me' signal does not specifically induce phagocytosis of circuits associated with a particular learning paradigm. As I noted in my previous review, I still believe that the enhancement of seemingly random phagocytic activity may have promoted a meta-plastic state in the brain, thereby facilitating learning in the mice—akin to the enhanced plasticity observed after stroke. The authors have acknowledged this possibility and incorporated it into the Discussion. I believe this explanation best accounts for the observed effects in the authors' experiments. While emphasizing this interpretation further would be welcome, I respect the authors' perspective and find their discussion of this point satisfactory.

Overall, the manuscript has been thoroughly revised, and I have no further comments.

We sincerely appreciate Reviewer #2's recognition of our efforts to revise the manuscript and to address the raised concerns. We are grateful for the reviewer's supportive evaluation and helpful perspective, which have significantly improved the clarity and quality of our study.

Reviewer #3 (Remarks to the Author):

The authors have addressed all my questions well and have greatly improved the study, and I believe they have also addressed the concerns of the other reviewers as reasonably as possible. Thus I would recommend publication of the current manuscript.

We sincerely thank the Reviewer #3's recognition that our responses have adequately addressed the previous concerns. The reviewer's encouraging comments are highly motivating and have been very helpful in finalizing this work.

Reviewer #4 (Remarks to the Author):

We sincerely thank Reviewer #4 for the detailed and constructive evaluation of our revised manuscript, including the thoughtful assessment of our responses to Reviewer #1's comments. The reviewer's feedback has been very helpful in improving the rigor, clarity, and balance of our data presentation and interpretations, which substantially strengthened the overall quality of our study. All corresponding revisions in the manuscript have been highlighted in yellow for clarity.

I thank the authors for their efforts in addressing the reviewer's previous comments and for the additional experimental work. SynTrogo appears to be a potentially powerful tool, and the in vitro characterization is notably thorough. However, several critical issues raised in the initial review remain insufficiently addressed, and some of the newly added data raise further concerns. In particular, the manuscript would benefit from a more objective presentation of the results—especially regarding the in vivo data, where the authors occasionally make overreaching claims. The manuscript advances the idea that SynTrogo enables selective synapse removal and subsequent strengthening of the remaining synapses, yet I believe the current dataset does not fully support these conclusions. Several interpretations would benefit from more cautious phrasing, and some additional data or clarification is needed. Below, I outline what I consider essential points to address prior to publication.

Major 1: Claim that syntrogo induces synapse engulfment and elimination without axonal degradation

1) Assessment of long-term axonal fate:

The long-term integrity of SynTrogo-expressing axons following sustained interaction with astrocytes expressing anti-GFP remains unclear. If I understand correctly, the added Extended Data Fig. 12 examines SynTrogo axons in animals lacking receptor expression on astrocytes. If this is the case, it should be explicitly clarified in the methods.

We thank the reviewer for pointing out this important clarification. We agree that the figure in the previous version could have been misleading. In the **Extended Data Fig. 12** experiment, astrocytes were indeed expressing the receptor, and under these conditions, we carefully monitored axonal morphology for up to two months. We did not observe any signs of axonal damage or fragmentation, suggesting that SynTrogo does not lead to axonal degeneration over time. To avoid any confusion, we have included the receptor signal in astrocytes in the revised **Extended Data Fig. 12**.

One could expect that syntrogo axons in proximity to astrocytes will be nibbled on as long as they stay close to astrocytes bearing the anti-GFP, leading to axonal fragmentation. If this is not the case, the authors should propose an explanation for what prevent syntrogo axons from being entirely munched by astrocyte until disappearance.

Moreover, If SynTrogo clusters at specific sites along the axon (as suggested by the punctate appearance of GFP signal), this may promote localized elimination.

We appreciate the reviewer for this critical point. Our results indicate that SynTrogo removes limited portions of the plasma membrane and adjacent cytosolic components, rather than inducing extensive axonal fragmentation. Because ligand-receptor complexes are co-internalized during the process (**Supplementary Video 2 and 4**), the available pool of surface

ligand–receptor pairs becomes progressively depleted, which likely constrains continued SynTrogo. This self-limiting nature may allow axons to maintain their overall integrity and prevent degeneration. Consistently, live-cell imaging revealed no disruption of membrane integrity or signs of cell death (**Supplementary Video 4 and Extended Data Fig. 4**), supporting that SynTrogo does not compromise axonal structural stability. Regarding this point, we have revised the manuscript accordingly (lines 123-127, 225-228).

As the reviewer noted, the punctate GFP signals along axons (**Extended Data Fig. 14**) suggest that SynTrogo occurs in a spatially confined manner, leading to localized elimination rather than widespread axonal loss. Regarding this point, we have revised the text to clearly describe these observations (lines 250-252).

2) Interpretation of colocalization analyses:

Colocalization analyses using confocal to prove internalization is problematic. Overlay of puncta with astrocytes does not prove that these puncta are within astrocytes, because the axial resolution of confocal microscopy is poor. A punctum sitting above or below the cell in the z dimension will produce colocalized signal in confocal, mistakenly giving the impression that the punctum is within the cell. This is a common issue in fluorescence microscopy, and the issue was clearly exemplified using CLEM in Weinhard et al. 2018.

I think the authors should tone-down their interpretations and avoid claiming synapses are internalized based on these experiments. Instead, it can only be stated that synaptic material appears colocalized with astrocytes, which suggest engulfment.

We thank the reviewer for this valuable comment. As suggested, we have toned down our interpretation of confocal images regarding the internalization of synaptic molecules, and now describe them as colocalization events suggestive of engulfment (lines 191-198, 205-210).

3) Concerns regarding CLEM methodology:

The criteria used to select SynTrogo versus control axons for CLEM appear inconsistent. According to the authors' response, SynTrogo GFP+ axons were selected based on contact with astrocytes, while control GFP+ axons were selected based on proximity, and no information was provided for SynTrogo GFP- axons. This introduces a confound, as contact status could influence synapse density independently of SynTrogo expression. A consistent selection criterion—especially with respect to astrocyte proximity—is essential.

In addition, the length of axon analyzed in each EM volume should be clearly reported, as short segments could introduce significant variability. Ideally, a table listing all analyzed axons, their contact status, length, and annotated synapses (for both GFP+ and GFP- axons) should be provided. This would improve transparency and allow more rigorous evaluation of synapse density data (e.g., in Extended Data Fig. 16).

We thank the reviewer for this important comment and fully agree with the concern. In the SynTrogo condition, membranes of astrocytes and neuronal axons exhibited tight associations, whereas this was not the case in the control. To minimize potential bias, axons were selected based on their close apposition to astrocytes for both control and SynTrogo (GFP+ and GFP- axons) groups, and we have clarified the selection criteria for axons in the Methods section (lines 892-897, 909-912). As requested, we have also added a new table in **Extended Data Fig. 17** that lists all analyzed axons, including their contact status, segment length, number of annotated synapses, and synaptic density. Importantly, this table shows that all analyzed axons

were in close apposition to astrocytes, thereby ensuring consistent criteria for comparison across groups. Our direct measurements indicate that the gap between axonal and astrocytic membranes was consistently ≤ 100 nm.

4) Interpretation of synapse density differences:

That SynTrogo axons contacted by astrocytes contain fewer synapses than control axons is consistent with pruning, but not conclusive. It remains possible that astrocytes preferentially contact axons with fewer synapses. While the authors attempt to rule this out using the AAI index, the method and interpretation remain unclear. A more informative control might compare axons contacted by astrocytes, with or without SynTrogo, in the same spatial context.

We thank the reviewer for this insightful comment. In response to this comment and in alignment with comment 3, we analyzed axons that were in close apposition to astrocytes (≤ 100 nm) in the same spatial context. Under the SynTrogo condition, GFP(+) axons displayed fewer synapses compared to neighboring GFP(-) axons, suggesting that SynTrogo may contribute to synaptic elimination (**Fig. 3j**). Notably, the synapse density of GFP(-) axons in SynTrogo group is comparable to that of axons in the control group.

Nevertheless, we fully agree with the reviewer that these findings are not conclusive and that alternative explanations cannot be entirely excluded. In line with this concern, we have carefully revised the text accordingly to tone down our interpretation throughout the manuscript.

5) Quantification of bouton vs. shaft engagement:

The astrocyte-axon interface ratio used to assess bouton vs shaft preference lacks methodological detail. Moreover, surface contact area does not necessarily correlate with engulfment frequency. Given the emphasis on engulfment events (e.g., pinching and closure), it would be more appropriate to quantify how frequently such events occur on boutons vs shafts. Comparing this to the relative surface area of boutons and shafts could help determine whether synaptic structures are disproportionately targeted. That said, if trogocytosis is uniformly induced and astrocytes happen to reside near synapses more than shafts, this could create an apparent bias that is not mechanistically meaningful. The term “preferential pruning” should be used with caution.

We thank the reviewer for this helpful comment. We agree that our description of the astrocyte-axon interface (AAI) analysis required further clarification. We have now provided additional methodological details in the Methods section regarding how the AAI ratio was quantified (lines 892-904).

We also acknowledge the reviewer’s valid point that surface contact area does not necessarily equate to engulfment frequency. As the reviewer notes, SynTrogo can in principle occur wherever astrocytes are closely apposed to neuronal membranes. Indeed, our in vitro observation showed that astrocytes can trogocytose axons, dendrites, and soma in culture (**Extended Data Fig. 8a-c**). Consistent with this, our analysis of pinching/closure events at boutons vs shafts in the SynTrogo group using 3D-EM reconstructions revealed such events at both compartments (**Extended Fig. 15j**). In line with the reviewer’s suggestion, we have revised the manuscript and now describe our findings more cautiously (lines 246-250, 281-293).

6) Lack of correlation between confocal and EM images:

The manuscript lacks direct correlation between light and EM images of GFP+ inclusions. For a claim as central as synaptic internalization, it is important to provide examples where confocal images showing GFP+ inclusions can be directly correlated to closed inclusions in EM. Despite the high density of GFP+ structures observed in anti-GFP astrocytes, no such examples are shown with synaptic vesicles clearly enclosed. Furthermore, many EM examples show engulfment of mitochondria, yet not synaptic vesicles.

If mitochondria are frequently present in closed inclusions, this raises questions about whether cytosolic content is being ingested, which contrasts with earlier claims that membrane components are primarily transferred. A more systematic report of how many closed inclusions contain synaptic vesicles and/or mitochondria (out of the total observed) would greatly improve clarity. Confocal–EM correlations, presented as z-stacks, would also strengthen the interpretation.

We thank the reviewer for this important comment. In line with the suggestion, we have added CLEM data to demonstrate the relationship between GFP signals and ultrastructural features. Specifically, we present examples of GFP signals correlated with pinching structures (**Extended Data Fig. 14a**) and with closed inclusions that contain synaptic vesicles and mitochondria (**Extended Data Fig. 15h,i**). In addition, we provide z-stacks of confocal–EM correlations in **Extended Data Fig. 15l**, and **Supplementary Video 6 and 7** to illustrate our workflow and representative examples.

To improve clarity, we have also quantified the composition of closed inclusions: 62.5% (5 out of 8) of examined inclusions contained synaptic vesicles and/or mitochondria (**Extended Data Fig. 15k**). While our earlier data described predominant uptake of membrane-associated components, we note that cytosolic components adjacent to the ligand can also be engulfed (**Extended Data Fig. 7**). Thus, the presence of presynaptic mitochondria and synaptic vesicles within inclusions is consistent with our mechanistic interpretation of SynTrogo. This observation is also in line with previous reports of amoebic trogocytosis, in which cytoplasmic material, including mitochondria, can be internalized (PMID: 24717428). We have described this point in the revised manuscript (lines 287-293).

7) Evidence for complete synapse elimination remains indirect:

Even if synaptic material is ingested, this does not establish that full synaptic structures are eliminated. While the data suggest SynTrogo may promote synaptic pruning, no single experiment provides definitive evidence. To strengthen this interpretation, live imaging of synaptic elimination in the presence of astrocyte–SynTrogo interaction would be highly informative. In the absence of such evidence, the conclusions regarding pruning should be more cautiously framed.

We agree that ingestion of synaptic material does not by itself demonstrate complete synapse elimination, and that our current dataset with fixed samples does not provide direct evidence of entire synaptic structures being pruned. In line with this concern, we have revised the text accordingly throughout the manuscript, and explicitly state that live imaging of synapse elimination would further strengthen this claim and plan to explore this in future studies (lines 487-489).

Major 2: Claim that syntrogo induces synaptic pruning which strengthen remaining synapses

1) Dynamic language such as “progressive stages,” “enlargement,” or “thickening” is used to describe observations from fixed tissues:

L275: "astrocytes display progressive stages of engulfment"

L295: "significant enlargement of GFP-labeled boutons"

L241: “This increased membrane thickness”

Such terms may imply temporal progression, which cannot be concluded from static EM images. More neutral descriptions would be appropriate.

We thank the reviewer for pointing out this issue. In response, we have revised these descriptions using more neutral language throughout the manuscript to accurately reflect morphological observations without suggesting dynamics.

2) Interpretation of higher bouton size:

That SynTrogo axons contain larger boutons than controls may suggest strengthening of remaining synapses, but this remains speculative. It is also important to confirm whether the compared groups were equitably selected. If SynTrogo axons were chosen based on confirmed contact with astrocytes, and controls were selected only by proximity, this would introduce bias. I encourage the authors to clarify their selection criteria and to rephrase speculative conclusions.

We thank the reviewer for this helpful comment. As mentioned in our earlier response to comment 3, for both control and SynTrogo (GFP+ and GFP- axons) groups, axons were selected based on their close apposition to astrocytes (≤ 100 nm) to minimize potential bias.

Regarding bouton size, as shown in the distribution of presynaptic bouton volumes in **Fig. 5c**, we observed very large boutons in SynTrogo axons that were rarely, if ever, detected in controls. In addition, the proportion of small boutons was markedly reduced in the SynTrogo group compared with controls, indicating an overall shift in bouton size distribution. This observation originally led us to describe these changes in terms of “strengthening”. However, in line with the reviewer’s concern, we now present the findings more cautiously as an increase in bouton size, which together with electrophysiological results may suggest—but does not prove—structural and functional strengthening of the remaining synapses (lines 335-351).

Minor comments:

We thank the reviewer for the many helpful comments. In some cases, the line references did not fully align with the corresponding text or figures, making it not entirely clear which part was being referred to. Nevertheless, we carefully considered the content of each comment and addressed them to the best of our understanding in the revised manuscript.

Figure 2: Please indicate which AAV serotype was used for infection.

We have now specified the AAV serotypes used for in vivo studies in the revised **Fig. 2a**, **Extended Data Fig. 9a**, and **Extended Data Fig. 12a**, and in the Methods section.

Figure 2: Image processing appears inconsistent between SynTrogo and control conditions—e.g., SynTrogo images appear more smoothed. Please confirm uniform processing.

We thank the reviewer and apologize for the confusion. All images were processed identically, but the magnified image in **Fig. 2f** appeared degraded during PDF conversion. We have now corrected the figure with intact images to accurately reflect the uniformly processed data.

Extended data Figure 9: Since Synaptophysin-TagBFP was co-infected with GFP-TM, why would synaptophysin not co-localize with GFP?

We thank the reviewer for this thoughtful question. One possibility is that even when the GFP ligand and synaptophysin are co-internalized, subsequent vesicle trafficking may separate synaptophysin-containing membranes from GFP ligand-containing membranes, resulting in inclusions that retain synaptophysin without detectable GFP. In addition, synaptophysin(+) GFP(-) puncta were observed in the control group, suggesting that synaptophysin may also be internalized through SynTrogo-independent processes. We have clarified this point in the revised text (lines 198–204).

Figure 2/ extended data Figure 10: The quantification methods for colocalized puncta differ between Extended Data Fig. 10 and Fig. 2, making direct comparison difficult. A standardized approach or clearer justification for the differences would be helpful.

We thank the reviewer for this valuable comment. We confirmed that the analyses in **Fig. 2g** and **Extended Data Fig. 10c,f** were performed using the same quantification method. However, **Fig. 2e** and **Extended Data Fig. 10b,e** were initially analyzed using different approaches. In line with the reviewer's suggestion, we have now re-analyzed these datasets with the same standardized method.

L105: "astrocytes can promote natural synaptic pruning": Please provide a reference

We carefully checked the manuscript but could not locate the exact phrase "astrocytes can promote natural synaptic pruning." A similar sentence in the text reads: "these observations indicate that SynTrogo may amplify naturally occurring processes rather than inducing entirely artificial phenomena." This statement was written as an interpretation of our own results, and thus no external reference can be provided. We kindly ask for the reviewer's understanding on this point.

L135: "Cell types": This should be corrected to "cell lines" to more accurately describe the experimental system shown.

We thank the reviewer for this careful observation. In this case, however, the experimental system included primary neurons and astrocytes as well as fibroblast, microglial, and cancer cell lines. Therefore, we believe that the term “cell types” is more appropriate than “cell lines” to accurately describe the system used.

L175: "in the absence of synaptic engulfment": Consider rephrasing

After carefully checking the manuscript, we could not identify the phrase “in the absence of synaptic engulfment” in our manuscript. As this phrase does not appear in the text, we were not able to rephrase it as suggested.

L180: "[cartoon showing engulfment]": The cartoon implies definitive engulfment of synaptic material, which is not yet convincingly demonstrated; suggest modifying it to reflect uncertainty.

We thank the reviewer for this comment. It was not entirely clear which cartoon was being referred to, but we assume the reviewer meant the schematics in **Fig. 6n** and **Extended Data Fig. 15m**. In response to this helpful suggestion, we have revised the manuscript (lines 305-307) and the legends for these schematics to explicitly indicate that they represent a putative model, thereby avoiding any implication of definitive engulfment. We hope this clarification resolves the concern.

L198: "S100B used to label astrocytes": Why was GFAP, which provided better signal quality, replaced with S100B?

We did not replace GFAP with S100 β ; rather, we used both markers depending on the experimental purpose. S100 β staining was employed to visualize astrocytic soma and to assess whether GFP ligand and synaptic molecules were colocalized within astrocytic territories. In contrast, GFAP staining was used to evaluate astrocytic selectivity of receptor expression and to test the possibility that SynTrogo could induce astrocytic reactivity.

L199: "SynTrogo-treated animals showed dense GFP signal in astrocytes": What is the origin of this dense GFP signal in SynTrogo astrocytes? If due to engulfment, why is it not degraded quickly as shown in extended data?

We thank the reviewer for this important question. When GFP ligand is taken up by astrocytes, it can pass through multiple steps of vesicle trafficking before reaching lysosomes. Thus, at a given time point, GFP may reside in different types of vesicular compartments (**Extended Data Fig. 6a-b**). As shown in **Extended Data Fig. 6f-g**, GFP ligand signals decrease upon fusion with lysosomes, but the signal is not completely lost. Furthermore, EM analysis indicated that inclusions showing intact GFP signals could be observed even at closure and degradation status (**Extended Data Fig. 15k**). Consistently, CLEM analysis revealed GFP signals in axons that remained alive while interacting with receptor-positive astrocytes (**Supporting Fig. 1**). Together, these results provide an explanation for why dense GFP signals can still be observed within astrocytic territories. To avoid any misleading interpretation, we have revised the description in the manuscript accordingly (lines 299-305).

Supporting Figure 1. Representative confocal-EM correlation images showing astrocyte and GFP+ axons in the SynTrogo group.

L200: "GFP signal was significantly higher in SynTrogo animals": A potential control to make sure higher GFP signal colocalized with astrocytes in synTrogo is not fortuitous due to higher signal in the field of view is to compare the amount of non-colocalized GFP signal between control and syntrogo. Expectedly, the non-colocalized (ie external) signal should be lower or at least the same between the two conditions.

We thank the reviewer for this constructive suggestion. To address this point, we quantified GFP signal intensity both inside and outside astrocytic regions in control and SynTrogo groups. Consistent with the reviewer's expectation, SynTrogo animals exhibited higher GFP intensity within astrocytic territories, whereas the GFP intensity outside astrocytes was comparable to that in controls. We have included these results in **Fig. 2d** and clarified this point in the revised manuscript (lines 182–184).

L201: "[figure showing colocalization]": Show separated channels for clarity on the proportion of signal truly inside vs. outside astrocytes.

L203: "vGlut1-RFP": If RFP is used for colocalization analysis, please include this channel in the figure panel.

L205: "Colocalization quantification of GFP and vGlut1-RFP in astrocytic soma": I don't see clear colocalization in the images. Clarify how the quantification was done—was it restricted to the soma such as in extended data figure 10?

We thank the reviewer for these three comments. While the line references were not fully clear, we assume the reviewer was referring to **Fig. 2** and **Extended Data Fig. 9**, which contain colocalization of GFP ligand and synaptic molecules. To address these points, we have revised the figure panels to include separated channels for each signal. In our study, only **Extended Data Fig. 9e** includes colocalization quantification, and this analysis was performed specifically in the astrocytic soma. We have clarified this point in the figure legends.

L208: "[figure panels with dotted lines showing astrocytic processes]": The yellow outline in the control appears offset. Also, image smoothing looks more pronounced in SynTrogo, which could bias the impression of colocalization.

We thank the reviewer for pointing this out. We believe the comment refers to **Fig. 2f**. The position of the yellow outline in the control panel has been corrected. In addition, we found

that the control image appeared degraded during PDF conversion, which may have exaggerated the impression of image smoothing. We have replaced it with the intact original image in the revised figure.

L210: "Fig. ED10a": This part of the figure and caption could benefit from clearer explanation. Why does the GFP signal appear more clustered in SynTrogo? Were images processed differently? This experiment might be important enough to include in the main figure.

Extended Data Fig. 10a shows a representative image of GFP ligand and LaG18-conjugated receptor with lower binding affinity for GFP, expressed in the brain. However, we believe the reviewer may have been referring to **Extended Data Fig. 9a**. In this experiment, GFP signals appeared more clustered in SynTrogo compared to the control, which is consistent with SynTrogo events in this region. We confirmed that both control and SynTrogo images were processed in the same manner.

L214: "[Imaris reconstructions vs original confocal]": Some GFP puncta overlaid with astrocytes in the raw images are missing in the Imaris analysis. This could affect quantification and reduce the apparent overlap between GFP and synaptic markers.

In our analysis, some GFP puncta that appeared in raw confocal images were not retained in the Imaris reconstructions. This was because astrocytic soma were masked using S100 β signals, and fluorescence intensity thresholding was applied equally to both control and SynTrogo groups. In this process, puncta positioned above or below the astrocytic surface were removed during masking, and very weak signals were excluded as noise. As a result, the reconstructed images contained fewer puncta than the raw confocal images. Importantly, the same thresholding and masking criteria were applied across groups to ensure fair comparison. We have clarified this point in the revised figure legends (**Extended Data Fig. 9b**).

L215: "[unlabeled color channels in image]": Please add a legend to specify which markers and channels are shown in this panel.

We thank the reviewer for this helpful comment, which we believe refers to **Fig. 2j**. We have now added the relevant marker and channel information to the figure to improve clarity.

L217: "Density of synaptic vesicle signal": Clarify what this refers to—is this the entire field of view, or only areas that colocalize with astrocytes?

Although the exact phrase at L217 does not appear in the manuscript, we assume the reviewer was referring to **Fig. 2j,k**. As indicated in the text and figure legends, **Fig. 2j** shows quantification within astrocytic territories, whereas **Fig. 2k** presents data from the CA1 stratum radiatum layer. These clarifications are already provided in the manuscript and figures.

L230: "astrocytes strongly favor contact with SynTrogo-infected axons": This phrasing may be too strong. Suggest rewording to indicate a trend or preference rather than a definitive conclusion.

The specific phrase “astrocytes strongly favor contact with SynTrogo-infected axons” does not appear in our manuscript. However, we agree with the reviewer’s concern regarding overly strong phrasing in some descriptions. Accordingly, we have carefully revised similar statements to use more neutral language and avoid definitive conclusions.

L240: "ratio of engulfed to total GFP+ puncta": Please define what this ratio represents, how it's calculated, and what units (if any) are used.

We did not perform an analysis based on the “ratio of engulfed to total GFP+ puncta” in our study. We therefore could not provide a definition or calculation for this parameter.

L248: "[Methods describing sample size not shown in figure]": How many astrocytes were analyzed per group? How many closed inclusions were identified, and how many contained synaptic vesicles?

In line with the suggestion, we have now included the number of analyzed astrocytes, the number of closed inclusions, and how many of them contained synaptic vesicles and/or mitochondria in the revised **Extended Data Fig. 15k**.

L250: "control = AAV expressing GFP alone": Please clarify in each section what “control” refers to, especially if control axons are also GFP-positive—it can otherwise be misleading in figures and legends.

We thank the reviewer for this helpful comment. In most experiments, the control condition refers to astrocytes expressing membrane-anchored RFP (Lck-RFP) instead of the receptor, unless otherwise specified. To avoid any misunderstanding, we have clarified the definition of “control” in each relevant section and updated the corresponding text in the Results and figure legends accordingly.

L265: "[confocal images with varying smoothness]": The GFP channel appears filtered or smoothed more in SynTrogo than control images, which could affect interpretation—please standardize image processing.

We thank the reviewer for raising this concern. All images were processed identically for both control and SynTrogo groups. If the comment refers to **Fig. 2f**, as mentioned earlier, the apparent difference was due to image degradation during PDF conversion. We have now replaced the panel with the intact original image in the revised figure.

L270: "astrocytes show no preference in contact": This conclusion would require comparing contacted vs non-contacted axons. As phrased, it may be overreaching.

We thank the reviewer for this comment. The statement the reviewer referred to does not appear in our manuscript. Instead, at L233 in our previous manuscript we wrote: “We found that astrocytes showed no preference for axons based on the surface areas of boutons and shafts in either the control or SynTrogo group (**Extended Data Fig. 14d–e**).” This analysis was intended to assess whether the axon–astrocyte interface ratio correlates with bouton or shaft surface area, and we found no significant correlation. To avoid misunderstanding, we have rephrased this sentence in the revised text as “We found no significant correlation between bouton/shaft surface area and AAI ratio in either the control or SynTrogo group” (lines 253-254).

L273: "Closed inclusions show vesicular content including mitochondria": Please include z-stacks to confirm these are truly closed inclusions. Also, are mitochondria transfer events consistent with earlier claims that only membrane-associated components are transferred?

In line with the suggestion, we have added z-stack images to the revised figure (**Extended Data Fig. 15l**) to confirm that it represents a closed inclusion. As shown previously (**Extended**

Data Fig. 7c), we showed that not only membrane-associated components but also cytoplasmic components adjacent to the GFP ligand can be co-engulfed. In addition, **Extended Data Fig. 8d** demonstrates that synaptic molecules can be internalized into astrocytes together with GFP ligand, and in vivo we also observed GFP and synaptic molecules colocalized within astrocytic soma, supporting the interpretation that cytoplasmic components can be transferred. Furthermore, earlier work on trogocytosis by amoeba (PMID: 24717428) reported that cytoplasmic components and mitochondria can be engulfed during the process, which is consistent with our findings.

L275: "astrocytes display progressive stages of engulfment": Since EM is static, terms like "pinching stage" or "emergence" are not appropriate. Consider rephrasing to describe morphological differences rather than temporal stages.

As recommended, we have revised the wording to describe morphological differences observed in EM images rather than implying temporal stages (please see Results, 'Cellular mechanisms of SynTrogo underlying reduced synaptic connectivity').

Reviewer# 4 comments on the responses to Reviewer# 1's concerns

While the initial version of the manuscript introduced an intriguing system to induce axonal trogocytosis, the revised version raises serious concerns about the rigor of the data presentation and the strength of the conclusions. Many of the major issues identified in the initial review remain unresolved, and in several cases, new data add further confusion rather than clarity. The syntrogo construct may indeed provide a useful platform to study axonal trogocytosis. However, the current evidence does not support the assertion that syntrogo induces "synaptic pruning" or functions as a presynaptic "eat-me" signal. At most, the results suggest that axonal trogocytosis might contribute to synapse elimination under certain conditions. The conclusions should be revised accordingly to reflect the limitations of the current data. Furthermore, the term "syntrogo" may be misleading, as it implies specificity for synaptic material that is not clearly demonstrated.

Major Point 1: Claims regarding presynaptic "eat-me" / "find-me" signaling

The revised manuscript attempts to support synaptic internalization using time-lapse imaging (Extended Data Fig. 8d,e). However, these data are not convincing:

- The inset image in ED Fig. 8d does not correspond to the time series.

We carefully re-examined the data and confirmed that the inset image in **Extended Data Fig. 8d,e** does correspond to the presented time series. The time is indicated in hr:min format, and to avoid any potential confusion, we have clarified this explicitly in the revised figure.

- The astrocyte channel is absent from the time-lapse sequence.

In the original inset image, the astrocyte channel was omitted to more clearly highlight GFP and synaptic molecules. In line with the reviewer's point, we have now included the astrocyte channel in the revised figure.

- The budding structures are not followed long enough to confirm separation from the axon.

We thank the reviewer for this comment. The budding structures were presented to illustrate the process by which GFP ligand and synaptic molecules are taken up together by astrocytes. As the reviewer noted, these budding events may not by themselves prove complete separation from the axon. However, as shown in the larger overview image on the left, several GFP(+) synaptophysin(+) puncta were observed at a distance from the axon fibers, which is consistent with their full separation. To improve clarity, we highlighted such puncta with arrows and provided magnified views of each ROI in the revised **Extended Data Fig.8d**.

In ED Fig. 8e, the GFP signal (labeled as synaptophysin) shows a pattern inconsistent with the cytoplasmic tagBFP, raising the possibility that the signal originates from astrocytic inclusions rather than axonal compartments. Without the astrocyte channel, this cannot be resolved.

We thank the reviewer for pointing out this issue and apologize for the labeling error in Extended Data Fig. 8e. In this experiment, cytoplasmic TagBFP was expressed in neurons, not Synaptophysin-TagBFP as incorrectly indicated in the panel. The key point of this result is that non-tagged cytoplasmic TagBFP expressed in control neurons was not appreciably internalized together with the GFP ligand. This contrasts with synaptic vesicle proteins such as synaptophysin, which were frequently co-engulfed along with the ligand. We have corrected

the labeling in the revised figure and clarified the interpretation in the text (lines 170-173). In addition, as suggested by the reviewer, we have included the astrocyte channel in the figure to provide a clearer representation of the uptake process.

Even if some synaptic markers are internalized, this does not constitute evidence that astrocytes selectively remove synapses. Additional analyses, such as CLEM, would be required to support the hypothesis, but still would not definitely make the point. Notably, the evidence presented tends to argue against selective engulfment of synaptic material.

We agree with the reviewer's comment. Regarding the selectivity of synapse removal and the engulfment of synaptic material, we had already revised the relevant expressions in the first round of revision to avoid implying selectivity. We have carefully re-checked the current manuscript and confirmed that no such statements remain.

Terminological adjustments—replacing “eat-me” with “find-me”—do not resolve the issue. There is insufficient evidence that syntrogo acts as a “find-me” signal in the classical sense, as trogocytosis appears restricted to local interactions with adjacent astrocytes. These terms, associated with endogenous signaling processes, should be avoided unless robustly supported.

We thank the reviewer for this important comment. We agree that the GFP ligand is not part of an endogenous signaling process. To avoid confusion, we have revised the discussion to refer to it as a “synthetic find-me” signal, emphasizing that it functions independently of endogenous processes and is likely to act based on astrocyte–neuron proximity rather than classical signaling (lines 421-425).

The concern regarding the distinction between adhesion and trogocytosis is acknowledged in the rebuttal, but the comparison between LaG18/LaG5 and the GFP ligand remains problematic, as it appears the colocalization analyses were conducted differently.

We thank the reviewer for raising this concern. As pointed out, **Extended Data Fig. 10b,e** had originally been analyzed differently from **Fig. 2e**. We have now re-analyzed these panels using the same method (v/v, %) and included the updated results in the revised figure.

Major Point 2: Lack of correlation between confocal and EM data

The manuscript still does not present any clear or convincing correlation between confocal images showing RFP+ astrocytes with GFP+ inclusions and EM image sequences depicting closed inclusions containing synaptic vesicles. This raises serious doubts about whether the CLEM data provide any real evidence for synaptic engulfment, and the methodology—as well as the general lack of rigor—remains concerning.

Most EM examples:

- Do not show synaptic vesicles. It is unclear what was quantified in ED Fig. 15c, as no corresponding EM image is shown.

We thank the reviewer for this important comment. In the previous version of the manuscript, we provided EM examples showing synaptic vesicles (**previous Fig. 3e and Extended Data Fig. 15d, f, g**). To address the reviewer's concern regarding the quantification shown in the earlier **Extended Data Fig. 15c**, we have now added representative EM images for each group in the revised figure (current **Extended Data Fig. 15c**) to clarify what was quantified.

-Depict inclusions containing cytoplasm with mitochondria—features inconsistent with the in vitro characterization of engulfed material.

As mentioned above, during SynTrogo, cytoplasmic components can also be taken up together with membrane-associated material. Therefore, the presence of cytoplasmic contents, including mitochondria and vesicular structures, in EM inclusions is consistent with our in vitro characterization (**Extended Data Fig. 7 and 8d**), as well as with previous reports of amoebic trogocytosis showing uptake of cytoplasmic material and mitochondria (PMID: 24717428).

- Are not convincingly shown to be closed structures, since the z-series does not track the compartments from beginning to end.

In line with the suggestion, we have added the complete z-series images to the revised **Extended Data Fig. 15I** to show the compartments from beginning to end. In addition, we have included a **Supplementary Video 6 and 7** to further aid readers' understanding of our analysis method.

- Are not shown to derive from axons expressing the GFP ligand; no correlation is established between GFP+ axons and the inclusions presented.

To directly address the issue, we have added CLEM images and complete z-series (**Extended Data Fig. 15I**) that demonstrate the inclusions are derived from GFP+ axons. In addition, **Supplementary Videos 6 and 7** now show a spatial correlation between GFP ligand signals and corresponding EM structures. As an example, an inclusion derived from a GFP+ axon is shown in **Supplementary Video 7** (00:21) and in the **Supporting Fig. 1** (shown above). These additions clarify the correspondence between GFP+ axons and the EM inclusions presented.

Furthermore, the interpretation that these EM structures represent “ongoing engulfment” is questionable, as the apparent “pinching” might be the result of physical constraints rather than genuine intermediate stages of trogocytosis. ED Fig. 9 is particularly problematic: several GFP+ puncta that appear to colocalize with astrocytes seem to have been excluded from the 3D-rendered mask, with no explanation provided.

We thank the reviewer for this thoughtful comment. As mentioned above, we have revised the text to avoid wording that implies temporal progression in the EM data and we now describe these observations in morphological terms (please see Results, ‘Cellular mechanisms of SynTrogo underlying reduced synaptic connectivity’). In **Extended Data Fig. 9**, we have clarified in the revised figure legends why certain GFP+ puncta visible in raw confocal images were not retained in the 3D-rendered mask – specifically, signals located above or below the astrocytic surface or excluded by fluorescence intensity thresholding during reconstruction. These clarifications explain why some signals were not included in the final rendering.

Lastly, the presence of synaptophysin+ but GFP– inclusions in astrocytes is not reconciled, despite both markers being co-expressed in the system.

We thank the reviewer for raising this important point. As mentioned in our response above, one possible explanation is that, even when synaptophysin and the GFP ligand are co-internalized, subsequent vesicle trafficking may separate synaptophysin-containing membranes from GFP-containing membranes, leading to inclusions that retain synaptophysin without detectable GFP. Another possibility is that synaptophysin can be internalized through

ligand-independent mechanisms, as synaptophysin(+) GFP(-) puncta were also observed in the control condition where GFP uptake was minimal. We have clarified this interpretation in the revised manuscript (lines 198-204).

Major Point 3: Experimental grouping and selection criteria

There is still no clear description of what distinguishes the experimental groups (“control,” “syntrogo GFP-,” “syntrogo GFP+”), how axons were selected for analysis, or how their proximity to astrocytes was determined. This information remains absent from the rebuttal, figure legends, revised results, and methods.

We thank the reviewer for raising this important point. To address this concern, we have added **Extended Data Fig. 17** summarizing the features of analyzed axons and included a detailed description of each experimental group, along with the criteria for axon selection and astrocyte proximity, in the Methods section (lines 892–896, 909-912). Specifically, consistent with our earlier response to Reviewer #4 (comment 3), axons in both control (GFP+) and SynTrogo (GFP+ and GFP-) groups were selected based on close apposition to astrocytes (≤ 100 nm) to minimize potential bias.

Minor Point:

Figure 1i currently lacks a legend and should be annotated accordingly.

We thank the reviewer for this comment. Upon careful checking, we confirmed that a legend is present in **Fig. 1i**. To improve visibility, we have highlighted the legend in yellow in the revised figure.

REVIEWER COMMENTS

Reviewer #4 (Remarks to the Author):

We sincerely thank Reviewer #4 for recognizing the conceptual novelty of our approach and for the constructive and insightful evaluation of our revised manuscript. We deeply appreciate the reviewer's guidance in refining our terminology, tempering mechanistic interpretations, and ensuring that our conclusions remain closely aligned with the imaging evidence. These suggestions substantially improved the clarity, rigor, and balance of the study. All corresponding revisions in the manuscript have been highlighted in yellow.

I would like to thank the authors for their thorough and thoughtful responses, for addressing the issues previously raised, and for providing additional material. It is clear that this manuscript represents a considerable amount of work, and the approaches employed are both elegant and technically demanding. The concept of artificially inducing trogocytosis is highly innovative—particularly in the context of potential synaptic targeting—and the *in vitro* characterization is well executed.

However, the *in vivo* evidence remains unconvincing. As noted in the previous revision round, the colocalization of VGlut1 puncta with astrocytes does not constitute proof of synaptic elimination, as the resolution of confocal microscopy is insufficient to confirm internalization, especially along the z-axis. For this reason, identification and further examination of colocalized puncta by Correlative Light and Electron Microscopy (CLEM) were crucial, and I commend the authors for undertaking this technically challenging approach.

Despite these efforts, and despite the numerous GFP- and synapse-positive puncta shown to colocalize with astrocytes in light microscopy, no double-membrane inclusion in the EM dataset could be correlated with a colocalized GFP punctum. Likewise, no examples of fully closed inclusions containing synaptic vesicles were provided. Together, these observations argue against synaptic elimination and instead suggest that the apparent vGlut/GFP puncta colocalized with astrocytes may represent imaging artefacts arising from limited confocal resolution. As the authors mention, it remains possible that, by the time of imaging (three weeks after Syntrogo expression), all GFP–receptor pairs had already been depleted, precluding observation of active elimination at that stage.

We appreciate the reviewer's careful evaluation and the clear articulation of the key issues concerning the interpretation of our *in vivo* data. The reviewer points out that colocalization of vGlut1 and GFP puncta signals within astrocytic areas cannot conclusively establish internalization of synaptic material, particularly given the limited z-resolution of light microscopy. This is precisely why we undertook extensive CLEM analyses, and we are grateful that the reviewer recognizes this effort.

We agree that our CLEM dataset does not demonstrate fully isolated, double-membrane inclusions – i.e., two clearly separated membranes with an intermembrane space – that contain synaptic vesicles and can be correlated with a GFP+ punctum in confocal images. Nevertheless, we were able to correlate a GFP+ punctum in confocal images with a partially closed inclusion that, in matched EM dataset, contained synaptic vesicles and exhibited membrane apposition consistent with enclosure (**Fig. 5i and Extended Data Fig. 14f**). Additional analysis of the eight inclusions identified in **Extended Data Fig. 14g** revealed that they are surrounded by

membranes with thickness values of ~ 28 nm, closely matching those observed at neuron–astrocyte contact sites (**Fig. 5a-b**). This supports the interpretation that these structures consist of two tightly apposed opposing membranes rather than a single continuous surface. While this does not constitute evidence of complete engulfment, it supports a model in which ligand–receptor coupling drives substantial membrane deformation and partial enclosure of axonal material.

Importantly, the observation of partially enclosed structures is not unique to SynTrogo nor indicative of a technical limitation. Prior EM studies of glial cell-mediated synaptic pruning report similar findings. For example, Buchanan et al. (PNAS 2022) showed that engulfed dendritic spines or boutons in oligodendrocyte precursor cells frequently remained continuous with their parent neurons across serial sections, rather than forming fully severed double-membrane vesicles. Consistently, other studies (Lee et al., Nature 2020; Morizawa et al. Nat. Neurosci. 2022) demonstrated that astrocytes and Bergmann glia partially surround and invaginate neuronal compartments that remain connected to their parental fibers. These reports parallel the partially enclosed configurations we observe and highlight morphological features that are compatible with a “grab-and-nibble”–like interaction between glial cells and neuronal elements, as discussed in prior studies of glia-mediated synaptic pruning.

As an independent line of investigation, our ongoing analysis of volume-EM datasets from P14 mouse brain – a developmental stage characterized by robust astrocytic phagocytic activity – similarly indicates that structures appearing fully enclosed in single sections typically remain continuous with axons when traced across serial slices (**Supporting Fig. 1**). Together, these results suggest that partial enclosure may represent a common feature of glial engagement with neuronal material, rather than a SynTrogo-specific characteristic.

As noted by the reviewer, the limited z-resolution of confocal microscopy makes it difficult to determine whether synaptic puncta colocalized with astrocytic volumes are truly internalized, and our current EM dataset does not allow definitive attribution of these structures to completed synaptic elimination. In light of these limitations, we have carefully rephrased the descriptions of our light and electron microscopy results in the Results section (“SynTrogo of CA3 presynaptic axons by CA1 astrocytes” and “Cellular mechanisms of SynTrogo underlying reduced synaptic connectivity”) to avoid implying synaptic engulfment *in vivo*, and we now explicitly acknowledge in the Discussion (lines 531-537) that our present *in vivo* data do not fully support a synaptic elimination mechanism.

Supporting Figure 1. Examples of axon–astrocyte interfaces in early postnatal brain showing apparent enclosure in single sections but continuity across serial EM slices.

Rather than showing closed inclusions, the data predominantly display extensive apposition of GFP+ axons and RFP+ astrocytes, with axons traversing astrocytic territories and occasionally wrapped or fenestrated by astrocytic processes (see Figure 3, Extended Figures 14 and 15, and Videos 6 and 7). This extensive coverage is likely a consequence of GFP/anti-GFP binding. Such physical interaction could plausibly reduce axonal accessibility for synapse formation and thereby contribute to the observed decrease in synapse number, redistribution of presynaptic components, enlargement of remaining synapses, and associated electrophysiological and behavioral phenotypes.

We agree with the reviewer that the extensive bouton–astrocyte apposition driven by GFP/ α GFP binding could plausibly limit axonal access to postsynaptic partners, contributing to reduced synapse number and the observed structural and functional changes. We now explicitly discuss this alternative mechanism in the Discussion and clarify that SynTrogo-associated synaptic changes and impaired synapse formation are not mutually exclusive mechanisms, both of which could contribute to the observed reduction in synapse number (lines 531–544). Determining the relative contribution of each process under SynTrogo conditions *in vivo* will require further investigation.

In light of these points, if the manuscript were to be considered for publication, I would recommend a substantial revision of the abstract, text, and figure legends to ensure the results are presented more cautiously and without implying that GFP/anti-GFP coupling induces synaptic engulfment. While synaptic elimination cannot be definitively excluded as a possible mechanism preceding the imaging time point, it is not demonstrated convincingly in the current data. Additionally, descriptive terms such as “pinching,” “closure,” “degradation,” or “lysis” should be used with care, as these processes remain speculative. For example, an axon passing through an astrocyte and contacting a phagocytic compartment (Extended Data Figure 15) cannot be confidently described as undergoing degradation.

We fully acknowledge the reviewer’s concerns regarding the risk of over-interpreting our *in vivo* data as evidence for synaptic engulfment or elimination. In accordance with this guidance, we have undertaken substantial revisions across the Abstract, Results, Discussion, and figure legends to ensure that all interpretations remain appropriately cautious and evidence-based.

While our cultured-cell experiments clearly demonstrate a trogocytosis-like process – where receptor-expressing cells internalize portions of ligand-expressing cells – we now explicitly restrict this description to the *in vitro* context. This approach allows the manuscript to maintain conceptual coherence while avoiding overstatement of the *in vivo* implications. Correspondingly, descriptions of *in vivo* observations have been carefully rephrased to avoid mechanistic claims that are not directly supported by the data.

Following the reviewer’s guidance, we have removed or replaced terms such as pinching, closure, degradation, and lysis with non-interpretive, morphology-focused descriptors (e.g., interdigitated membrane profiles, approximation of opposing membranes, partially enclosed configurations). These revisions ensure that the text does not imply processes – such as completed engulfment or degradation – that cannot be conclusively demonstrated by our current analyses.

We are grateful to the reviewer for these constructive insights, which have substantially improved the clarity, precision, and interpretive balance of the manuscript.

Furthermore, the quantification strategy should be clarified. It would be important to explicitly describe what n represents in each analysis and to ensure that trends remain consistent when using biological replicates ($n = \text{animals}$) rather than individual cells or synapses. Given that $n = 3$ animals, it is possible that the overall trend could be disproportionately influenced by a single individual. Demonstrating that the same effect holds true when averaging at the animal level would strengthen the robustness of the conclusions.

We thank the reviewer for raising this important point regarding the quantification strategy and the necessity of clarifying what n represents across analyses. In accordance with this recommendation, we have revised corresponding figure legends and methods to explicitly indicate the definition of n for each dataset.

Although statistical comparisons were originally performed using all measured units across control and SynTrogo groups, we fully agree that evaluating the results at the level of biological replicates is essential for assessing robustness. Following the reviewer's suggestion, we reanalyzed each dataset by averaging measurements within each animal and have now included these per-animal values in the accompanying Source Data file (an example is provided in **Supporting Fig. 2**).

Overall, the trends observed when using animal-level averages were largely consistent with the conclusions reported in the main text. However, as anticipated given the relatively small number of animals ($N = 3$ per group in several analyses) and the inherent inter-animal variability, certain comparisons did not reach statistical significance when analyzed strictly at the animal level. We provide a summarized information of these analyses in the Source Data file and have described the observed variability in the revised manuscript (lines 526-527).

We believe these additional analyses and clarifications enhance the transparency and robustness of our quantification strategy, and we appreciate the reviewer's guidance on this point.

Finally, reorganization of the figures could greatly improve readability. Some data currently placed in the Extended Data could be incorporated into the main figures, and the overall number of figures could be reduced to enhance clarity and narrative flow.

We appreciated the reviewer's thoughtful suggestion to reorganize the figures to improve clarity and narrative flow. In response, we carefully re-evaluated the placement of several datasets – particularly those related to the *in vitro* characterization of SynTrogo and the ultrastructural analyses – and relocated selected results from **Extended Data Figs. 2, 14, and 15** in the previous manuscript into the main figures.

We believe these revisions enhance the readability of the manuscript and more effectively guide the reader through the mechanistic and structural findings associated with SynTrogo.

4c				
	Per axon		Per mouse	
	Ctrl	SynT	Ctrl	SynT
Mean	11.3	30.22	10.25	33.16
S.E.M	1.378	2.388	1.175	3.478
Number of values	35	32	3	3
Ctrl vs SynT (Unpaired t-test)	<0.0001		0.0034	
	17.16	23.777019	11.81	29.01
	6.63	39.204545	7.95	30.4
	2.08	25.658915	11	40.07
	5.28	46.304348		
	2.61	41.127542		
	5.3	52.962963		
	8.43	10.877404		
	6.25	23.370577		
	1.8	28.113553		
	3.95	2.3636364		
	14.62	41.362126		
	4.19	46.40884		
	20.61	21.955556		
	14.25	21.789561		
	24.64	23.85		

Supporting Figure 2. Example of individual unit values and corresponding animal-level averages in the Source Data file.

REVIEWER COMMENTS

Reviewer #4 (Remarks to the Author):

We sincerely thank Reviewer #4 for the careful and constructive feedback provided over multiple rounds of revision. The reviewer's detailed suggestions have been invaluable in strengthening our manuscript. We have carefully reviewed the reviewer's final comments and prepared our responses accordingly, as detailed below.

I appreciate the substantial efforts the authors have made over several rounds of revision to refine the manuscript and more carefully articulate the limitations of the *in vivo* data. At this final stage, my comments are focused on ensuring that terminology and figure presentation consistently and unambiguously reflect what is directly supported by the evidence.

Given the current level of scrutiny in the field of glial-mediated pruning, clear distinction between demonstrated astrocyte–axon interactions and more speculative interpretations regarding synaptic elimination will be particularly helpful for readers. Addressing the minor points below will help prevent potential misinterpretation and further strengthen the clarity and impact of the manuscript upon publication.

1. Terminology: The term “SynTrogo” remains ambiguous. Since “Syn” implies “synaptic,” it suggests “synaptic trogocytosis”—a framing no longer supported by the *in vivo* data. I recommend a term that explicitly reflects induced trogocytosis (e.g., “i-Trogo”, or anything of the like) to avoid reader confusion.

We sincerely thank the reviewer for raising this important point regarding terminology. We agree that the abbreviation “SynTrogo” could potentially be misinterpreted as referring to “synaptic trogocytosis,” particularly in the context of our *in vivo* experiments. However, our intention has consistently been to denote Synthetic Trogocytosis, as defined at first mention in the Abstract and Results sections in the manuscript. To further minimize any possibility of confusion, we have revised the Results section title to “Development and characterization of Synthetic Trogocytosis (SynTrogo)”, and have introduced the full name in the opening sentence of the Discussion to explicitly link the abbreviation to its full term and ensure clarity and consistency in terminology throughout the manuscript.

As suggested, we carefully considered alternative names such as “i-Trogo” (induced trogocytosis). However, we think that the term “induced” may imply precise temporal control of trogocytosis, which is not an inherent feature of the current platform. For conceptual accuracy and consistency, we therefore retain the term SynTrogo while reinforcing its definition and usage in the text.

2. Correlation vs. Causation: Passages such as L344 (“Remodelling of remaining synapses under reduced synaptic connectivity”) and L439–445 (“Following this synaptic reduction, the remaining presynaptic and postsynaptic compartments exhibited coordinated structural and functional remodelling...”, “Our findings indicate that a reduction in synaptic connectivity alone can reshape the synaptic landscape...”) still imply a causal relationship. Please explicitly state that remodelling is associated with reduced connectivity, rather than necessarily resulting from it.

We fully acknowledge the reviewer's important point that certain descriptions in the manuscript could be interpreted as implying a causal relationship between synaptic reduction and subsequent remodeling. We agree that our data show an association rather than direct causation. In response, we have revised the relevant passages to avoid causal phrasing and to more accurately reflect the observational nature of our findings. The revised text now reads as follows:

L355: Remodeling of remaining synapses under reduced synaptic connectivity

→ Remodeling of remaining synapses *associated with* reduced synaptic connectivity

L449-450: Following this synaptic reduction, the remaining presynaptic and postsynaptic compartments exhibited coordinated structural and functional remodeling

→ *Under* reduced synaptic connectivity, the remaining presynaptic and postsynaptic compartments exhibited coordinated structural and functional remodeling

L454-457: Our findings indicate that a reduction in synaptic connectivity alone can reshape the synaptic landscape, and that neurons actively reorganize remaining synaptic connections following such reduction, providing insights into the cellular mechanisms that may contribute to homeostatic synaptic scaling.

→ Our findings indicate that reduced synaptic connectivity *is accompanied by* broader changes in synaptic organization and properties of remaining connections, providing insight into cellular processes *potentially relevant to* homeostatic synaptic scaling.

L494-496: An important finding of this study is that SynTrogo-mediated synaptic reduction led to substantial remodeling of both targeted presynaptic and coupled postsynaptic regions.

→ An important finding of this study is that *the reduction in synaptic connectivity under SynTrogo conditions was associated with* remodeling of both targeted presynaptic and coupled postsynaptic regions.

3. Distinguishing Colocalization from Engulfment: The wording in L191 (and anywhere else relevant) should be adjusted for technical accuracy. The phrase "accumulation of ligand puncta within soma area..." should be revised into "increased puncta colocalized with astrocytes". It is necessary to state in the results section that light microscopy cannot definitively prove engulfment due to resolution limits, and that these observations are suggestive of engulfment but require the higher-resolution validation provided by the CLEM experiments.

We thank the reviewer for this important suggestion. We agree that it is critical to distinguish colocalization from definitive engulfment in light-microscopy analyses. Accordingly, we have revised the relevant descriptions to replace wording implying internalization with terminology reflecting signal colocalization. Specifically, we have modified the following statements:

L190-191: High-magnification images revealed an accumulation of ligand puncta within soma area of receptor-expressing astrocytes relative to the control group.

→ High-magnification images revealed *significantly increased ligand signal colocalized with* receptor-expressing astrocytes relative to the control group.

L195-196: excitatory presynaptic molecules (VGluT1) were significantly enriched within astrocytic soma areas under SynTrogo conditions

→ excitatory presynaptic molecules (VGluT1) *showed increased signal intensity* within astrocytic soma areas under SynTrogo conditions

L200-201: suggesting that astrocytic enrichment of presynaptic components is positively associated with ...

→ suggesting that astrocytic *colocalization* of presynaptic components is positively associated with ...

L203-204: astrocytes displayed puncta containing both ligand and synaptophysin

→ *astrocytic regions* displayed puncta *in which ligand and synaptophysin signals were colocalized*

L205-206: when the GFP ligand and synaptophysin enter the same astrocytic territory

→ when the GFP ligand and synaptophysin *are present within* the same astrocytic territory

L215-216: Application of LaG18- or LaG5-conjugated receptors ... did not cause significant increase in synaptic molecule signals in astrocytes

→ ... did not cause a significant increase in *synaptic marker signal intensity within astrocytic territories*

L218: the enrichment of synaptic molecules in astrocytes

→ *the increased colocalization* of synaptic molecules with astrocytes

In addition, we have added a clarifying statement in the Results section (L210-213) as follows: *Given the resolution limits of confocal light microscopy, these observations cannot definitively establish engulfment. Although they may be suggestive of internalization, definitive ultrastructural validation requires higher-resolution imaging analyses (e.g. correlative light and electron microscopy imaging).*

4. Presentation of “Inclusions” and Figure Organization (Figs 4, 5, & 7): CLEM data show that confocal puncta correspond to axons closely apposed to or wrapped by astrocytic processes, rather than discrete, double-membrane synaptic inclusions. To clarify this:

- The CLEM data should be presented explicitly as validation of the light-microscopy colocalization analysis (e.g., by swapping or merging Figures 4 and 5).
- In the current Figure 5, no bona fide inclusions (double-membrane structures) are shown. It is therefore unclear what structures are quantified in Fig. 5g and 5h. Please avoid the term “inclusion” for enwrapped axons that are not severed. Please label structures in Fig 5d/f accordingly.

We appreciate the reviewer’s concern that our presentation could be interpreted as implying discrete, double-membrane synaptic inclusions. We agree that it is important to clearly distinguish closely apposed or enwrapped axonal profiles from bona fide engulfed inclusions. **Figure 4** was designed to define the ultrastructural characteristics of axon–astrocyte interfaces corresponding to the colocalization of the ligand and astrocytes observed by light microscopy. **Figure 5** then focused specifically on bouton-associated ultrastructural features within these interaction sites. This organization preserves a coherent progression from interface characterization to synapse-level analysis.

To address the reviewer's concern, we have clarified in the revised manuscript that the CLEM data provide ultrastructural context and validation for the light-microscopy colocalization analysis, rather than direct evidence of synaptic pruning. We have also removed the term "inclusion" throughout the manuscript and revised the legend in **Fig. 5d,f** to describe these structures as astrocyte-enwrapped axonal profiles that remain continuous with parent axons. Furthermore, we now explicitly state that the quantified structures in **Fig. 5g-h** correspond to axonal profiles appearing as enclosed structures within single 2D EM sections, and do not necessarily represent severed or fully internalized synaptic fragments. Together, these revisions clarify the structural interpretation of the CLEM data while preserving the logical organization of **Figures 4 and 5**.

- Clarify how continuous axons could be "individualized" for diameter measurements in Fig 5g/h. Since I assume this refers to enwrapped continuous axons and not round, double membrane inclusions, I do not think diameter quantification is appropriate, and these measurements are probably redundant to the AAI previously measured. Consider removing these panels if they duplicate the contact quantification in Fig 4.

We thank the reviewer for raising this important point. The structures analyzed in **Fig. 5g-h** correspond to astrocyte-enwrapped axonal profiles observed in single 2D EM sections. Three-dimensional reconstructions confirmed that these profiles remain continuous with their parent axons and do not represent severed or fully internalized fragments. For quantitative analysis, we selected single 2D sections from z-stack series in which axonal cross-sections appeared enclosed by astrocytic processes (as illustrated in **Fig. 5e**). Within these sections, we measured the maximal cross-sectional span of these profiles, acknowledging that the structures are not perfectly circular and reflect 2D cross-sectional views rather than fully isolated structures.

This analysis is distinct from the axon-astrocyte interface (AAI) in **Fig. 4**, which quantifies the extent of membrane apposition along interfaces. In contrast, **Fig. 5g-h** quantify the cross-sectional dimension and density of axonal profiles appearing enclosed by astrocytic processes in single 2D EM sections at bouton-associated interaction sites. The manuscript text and figure legends have been revised to explicitly describe the nature of the quantified structures and the method of measurement.

- Move panels 5j, k, and l to Figure 7, as they characterize synaptic morphology. Note: Fig 5k lacks a y-axis label.

We thank the reviewer for this suggestion. **Fig. 5j-l** were included to quantify synaptic density and spine volume distribution in control and SynTrogo groups, rather than to characterize overall synaptic morphology. Figure 5 integrates ultrastructural interaction analysis (**Fig. 5d-h**) with synaptic measurements derived from ligand-expressing axons contacting receptor-expressing astrocytes within the same spatial context. Specifically, **Fig. 5j-l** quantify synapse number per unit axonal length and spine volume distribution under SynTrogo conditions, linking astrocyte-axon interaction sites to changes in synaptic density.

We have corrected **Fig. 5k** to include the y-axis label, now clearly indicated as "Synapse number / μm ".

5. While the EM data could be compatible with a “grab-and-nibble” model, they do not constitute direct evidence of elimination. Explicitly stating that colocalized puncta represent proximity or wrapping, rather than severed fragments, aligns with recent literature questioning colocalization-based pruning claims.

We appreciate and agree with the reviewer’s point. In response, we have revised the relevant sections (CLEM Results: L288, 290-291, 326-331) to explicitly state that the observed colocalized puncta represent close apposition or wrapping of axonal segments by astrocytic processes, rather than severed synaptic fragments. We further clarify that these structures remain continuous with parent axons in 3D reconstructions and do not constitute direct evidence of synaptic elimination.

6. Figure 8n: The schematic depicts synaptic elimination, which is not demonstrated. This should be revised to reflect astrocyte–axon interaction and remodelling without implying pruning.

We appreciate the reviewer’s concern regarding the schematic in **Fig. 8n**. As the current data do not directly demonstrate synaptic elimination, we have removed this schematic to avoid potential overinterpretation.

Overall, I believe this work presents a valuable and technically impressive tool, and the authors have made substantial efforts to revise the manuscript in response to prior concerns. The remaining points above are intended to further align the language, figures, and schematics with the strength and limitations of the in vivo data, and to avoid potential misinterpretation by readers. Addressing these minor clarifications would, in my view, strengthen the manuscript’s impact and ensure that its conclusions are conveyed as precisely and transparently as possible.